# Universality of AdaGrad Stepsizes for Stochastic Optimization: Inexact Oracle, Acceleration and Variance Reduction

**Anton Rodomanov**
CISPA*
anton.rodomanov@cispa.de

**Xiaowen Jiang**
Saarland University and CISPA*
xiaowen.jiang@cispa.de

**Sebastian Stich**
CISPA*
stich@cispa.de

## Abstract

We present adaptive gradient methods (both basic and accelerated) for solving convex composite optimization problems in which the main part is approximately smooth (a.k.a. $(\delta, L)$-smooth) and can be accessed only via a (potentially biased) stochastic gradient oracle. This setting covers many interesting examples including Hölder smooth problems and various inexact computations of the stochastic gradient. Our methods use AdaGrad stepsizes and are adaptive in the sense that they do not require knowing any problem-dependent constants except an estimate of the diameter of the feasible set but nevertheless achieve the best possible convergence rates as if they knew the corresponding constants. We demonstrate that AdaGrad stepsizes work in a variety of situations by proving, in a unified manner, three types of new results. First, we establish efficiency guarantees for our methods in the classical setting where the oracle's variance is uniformly bounded. We then show that, under more refined assumptions on the variance, the same methods without any modifications enjoy implicit variance reduction properties allowing us to express their complexity estimates in terms of the variance only at the minimizer. Finally, we show how to incorporate explicit SVRG-type variance reduction into our methods and obtain even faster algorithms. In all three cases, we present both basic and accelerated algorithms achieving state-of-the-art complexity bounds. As a direct corollary of our results, we obtain universal stochastic gradient methods for Hölder smooth problems which can be used in all situations.

## 1  Introduction

**Motivation.** Gradient methods are among the most popular and efficient optimization algorithms for solving machine learning problems. To achieve the best convergence speed for these algorithms, their stepsizes needs to be chosen properly. While there exist various theoretical recommendations, dictated by the convergence analysis, on how to select stepsizes based on various problem-dependent parameters, they are usually impractical because the corresponding constants may be unknown or their worst-case estimates might be too pessimistic. Furthermore, every applied problem usually belongs to multiple problem classes at the same time, and it is not always evident in advance which of them better suits the concrete problem instance one works with. For classical optimization algorithms, this problem is typically resolved by using a line search. This is a simple yet powerful mechanism which automatically chooses the best stepsize by checking at each iteration a certain condition involving the objective value, its gradient, etc.

However, the line-search approach is usually unsuitable for problems of stochastic optimization, where gradients are observed with random noise (unless some extra assumptions are made, see [57]). For these problems, it is common instead to apply so-called adaptive methods which set up their

---

*CISPA Helmholtz Center for Information Security, Saarbrücken, Germany

stepsizes by simply accumulating on-the-fly certain information about observed stochastic gradients. The first such an algorithm, AdaGrad [17, 39], was obtained from theoretical considerations but quickly inspired several other heuristic methods like RMSProp [56] and Adam [32] that are now at the forefront of training machine learning models.

Excellent practical performance of adaptive methods on various applied problems naturally sparked a lot of theoretical interest in these algorithms. An important observation was done by Levy, Yurtsever, and Cevher [34] who showed that AdaGrad possesses a certain universality property, in the sense that it works for several problem classes simultaneously. Specifically, they showed that AdaGrad converges both for nonsmooth problems with bounded gradient and also for smooth problems with Lipschitz gradient, without needing to know neither the corresponding Lipschitz constants, nor the oracle's variance but enjoying the rates which are characteristic for algorithms which have the knowledge of these constants. They also presented an accelerated version of AdaGrad with similar properties. An independent version of the accelerated AdaGrad including diagonal scaling was proposed by Deng, Cheng, and Lan [12]. Further improvements and generalization of these ideas were considered in [18, 28, 30].

Nonsmooth and smooth problems are the extremes of the more general Hölder class of problems. The fact that AdaGrad methods simultaneously work for these two extreme cases does not seem to be a coincidence and suggests that these algorithms should work more generally for any problem with intermediate level of smoothness. Some further confirmations to this were recently provided in [48] although in a rather restricted setting of deterministic problems and only for the basic AdaGrad method. The stochastic case and acceleration were constituting an open problem which was recently resolved in [49] for a slightly modified AdaGrad stepsize (see (4)).

All the previously discussed results were proved only for the classical stochastic optimization setting where the variance of stochastic gradients is assumed to be uniformly bounded. In a recent work, Attia and Koren [2] showed that the basic AdaGrad method for smooth problems works under the more general assumption when the variance is bounded by a constant plus a multiple of the squared gradient norm. On a related note, it was also shown recently that AdaGrad stepsizes can be used inside gradient methods with SVRG-type variance-reduction. The first such an algorithm was proposed in [16]. The accelerated SVRG method enjoying optimal worst-case oracle complexity for smooth finite-sum optimization problems was later presented in [36].

**Contributions.** In this work, we further extend the results mentioned above by demonstrating that AdaGrad stepsizes are even more universal than was shown previously in the literature. Specifically, we consider the composite optimization problem where the main part is approximately smooth (a.k.a. $(\delta, L)$-smooth) and can be accessed only via a (potentially biased) stochastic gradient oracle. This setting is more general than typically considered in the literature on adaptive methods and covers many interesting examples, including smooth, nonsmooth and, more generally, Hölder smooth problems, problems in which the objective function is given itself as another optimization problem whose solution can be computed only approximately, etc.

Our contributions can be summarized as follows:

1. We start, in Section 3, with identifying the key property of AdaGrad stepsizes, which allows us to apply these stepsizes, in a unified manner, in a variety of situations we consider later. We present our two mains algorithms, UniSgd and UniFastSgd which are the classical stochastic gradient method (SGD) and its accelerated version, respectively, equipped with AdaGrad stepsizes.

2. We then establish, in Section 4, efficiency guarantees for these methods in the classical setting where the oracle's variance is assumed to be uniformly bounded.

3. In Section 5, we complement these results by showing that, under additional assumptions that the variance is itself approximately smooth w.r.t. the objective function, the same UniSgd and UniFastSgd without any modifications enjoy implicit variance reduction properties allowing us to express their complexity estimates in terms of the variance only at the minimizer.

4. Under the additional assumption that one can periodically compute the full (inexact) gradient of the objective function, we show, in Section 6, how to incorporate explicit SVRG-type variance reduction into our methods, obtaining new UniSvrg and UniFastSvrg algorithms which enjoy even faster convergence rates by completely eliminating the variance.

Our results are summarized in Table 1 (in the BigO-notation). In all the situations, we present both basic and accelerated algorithms whose only essential parameter is an estimate $D$ of the diameter

Table 1: Summary of main results for solving problem (1) with our methods. "Convergence rate" is expressed in terms of the expected function residual at iteration $k$ (or $t$, depending on the method). "SO complexity" denotes the cumulative stochastic-oracle complexity of the method since its start and up to iteration $k$ (or $t$), which is defined as the number of queries to the stochastic oracle $\widehat{g}$; for SVRG methods, we assume that querying the (inexact) full-gradient oracle $\bar{g}$ is $n$ times more expensive than $\widehat{g}$, and define the SO complexity as $N_{\widehat{g}} + n N_{\bar{g}}$, where $N_{\widehat{g}}$ and $N_{\bar{g}}$ are the number of queries to $\widehat{g}$ and $\bar{g}$, respectively. The second and third columns should be understood in terms of the BigO-notation which we omit for brevity.

| Method | Convergence rate | SO complexity | Assumptions | Reference |
|---|---|---|---|---|
| UniSgd (Alg. 1) | $\frac{L_f D^2}{k} + \frac{\sigma D}{\sqrt{k}} + \delta_f$ | $k$ | 1, 2, 3 | Thm. 4 |
| | $\frac{(L_f + L_{\widehat{g}}) D^2}{k} + \frac{\sigma_* D}{\sqrt{k}} + \delta_f + \delta_{\widehat{g}}$ | | 1, 2, 6 | Thm. 7 |
| UniFastSgd (Alg. 2) | $\frac{L_f D^2}{k^2} + \frac{\sigma D}{\sqrt{k}} + k\delta_f$ | $k$ | 1, 2, 3 | Thm. 5 |
| | $\frac{L_f D^2}{k^2} + \frac{L_{\widehat{g}} D^2}{k} + \frac{\sigma_* D}{\sqrt{k}} + k\delta_f + \delta_{\widehat{g}}$ | | 1, 2, 6 | Thm. 8 |
| UniSvrg (Alg. 3) | $\frac{(L_f + L_{\widehat{g}}) D^2}{2^t} + \delta_f + \delta_{\widehat{g}}$ | $2^t + n \log t$ | 1, 2, 6, 9 | Thm. 10 |
| UniFastSvrg (Alg. 4) | $\frac{(L_f + L_{\widehat{g}}) D^2}{n(t - \log\log n)^2} + t(\delta_f + \delta_{\widehat{g}})$ | $nt$ | 1, 2, 6 | Thm. 11 |

of the feasible set; the methods automatically adapt to all other problem-dependent constants. In a number of special cases, our algorithms achieve known state-of-the-art complexity bounds, but not restricted to those special cases. In Section 7, we illustrate the significance of our results by demonstrating that complexities for our methods on stochastic optimization problems with Hölder smooth components can be obtained as simple corollaries from our main results.

## 2 Preliminaries

**Notation.** We work in the space $\mathbb{R}^d$ equipped with the standard inner product $\langle \cdot, \cdot \rangle$ and a certain Euclidean norm: $\|x\| := \langle Bx, x \rangle^{1/2}$, where $B$ is a fixed positive definite matrix. The dual norm is defined in the standard way: $\|s\|_* := \max_{\|x\|=1} \langle s, x \rangle = \langle s, B^{-1}s \rangle^{1/2}$.

For a convex function $\psi : \mathbb{R}^d \to \mathbb{R} \cup \{+\infty\}$, its (effective) domain is the following set: $\operatorname{dom}\psi := \{x \in \mathbb{R}^d : \psi(x) < +\infty\}$. By $\partial\psi(x)$, we denote the subdifferential of $\psi$ at a point $x \in \operatorname{dom}\psi$; the specific subgradients are typically denoted by $\nabla\psi(x)$.

A convex function $f : \mathbb{R}^d \to \mathbb{R}$ is called $(\nu, H)$-Hölder smooth for some $\nu \in [0, 1]$ and $H \geq 0$ iff $\|\nabla f(x) - \nabla f(y)\|_* \leq H\|x - y\|^\nu$ for all $x, y \in \mathbb{R}^d$ and all $\nabla f(x) \in \partial f(x)$, $\nabla f(y) \in \partial f(y)$. Apart from the special case of $\nu = 0$, such a function $f$ is differentiable at every point, i.e., $\partial f(x)$ is a singleton. A $(1, L)$-Hölder smooth function is usually called $L$-smooth.

For a convex function $\psi : \mathbb{R}^d \to \mathbb{R} \cup \{+\infty\}$, point $x \in \mathbb{R}^d$, vector $g \in \mathbb{R}^d$, and coefficient $M \geq 0$, by $\operatorname{Prox}_\psi(x, g, M) := \operatorname{argmin}_{y \in \operatorname{dom}\psi}\{\langle g, y \rangle + \psi(y) + \frac{M}{2}\|y - x\|^2\}$, we denote the proximal mapping. When $M = 0$, we allow the solution to be chosen arbitrarily.

For a convex function $f : \mathbb{R}^d \to \mathbb{R}$, points $x, y \in \mathbb{R}^d$ and $\nabla f(x) \in \partial f(x)$, we denote the Bregman distance by $\beta_f^{\nabla f(x)}(x, y) := f(y) - f(x) - \langle \nabla f(x), y - x \rangle \ (\geq 0)$. When the specific subgradient $\nabla f(x)$ is clear from the context, we use the simplified notation $\beta_f(x, y)$.

The positive part of $t \in \mathbb{R}$ is $[t]_+ := \max\{t, 0\}$. For $\tau > 0$, we also use $\log_+ \tau := \max\{1, \log \tau\}$.

**Problem Formulation.** In this paper, we consider the composite optimization problem

$$F^* := \min_{x \in \operatorname{dom}\psi} \left[ F(x) := f(x) + \psi(x) \right], \tag{1}$$

where $f : \mathbb{R}^d \to \mathbb{R}$ is a convex function, and $\psi : \mathbb{R}^d \to \mathbb{R} \cup \{+\infty\}$ is a proper closed convex function which is assumed to be sufficiently simple in the sense that the proximal mapping $\operatorname{Prox}_\psi$ can be easily computed. We assume that this problem has a solution which we denote by $x^*$.

To quantify the smoothness level of the objective function, we use the following assumption:

---

**Algorithm 1** $(\bar{x}_N, x_N, M_N) \cong \mathrm{UniSgd}_{\widehat{g}, \psi}(x_0, M_0, N; D)$

---

**Input:** Oracle $\widehat{g}$, comp. part $\psi$, point $x_0 \in \mathrm{dom}\,\psi$, coefficient $M_0$, iteration limit $N$, diameter $D$.
 1: $g_0 \cong \widehat{g}(x_0)$.
 2: **for** $k = 0, \ldots, N-1$ **do**
 3:      $x_{k+1} = \mathrm{Prox}_\psi(x_k, g_k, M_k), \ g_{k+1} \cong \widehat{g}(x_{k+1})$.
 4:      $M_{k+1} = M_+(M_k, D^2, x_k, x_{k+1}, g_k, g_{k+1})$           $\triangleright$ e.g. $\overset{(3)}{=} \sqrt{M_k^2 + \frac{1}{D^2}\|g_{k+1} - g_k\|_*^2}$.
 5: **return** $(\bar{x}_N, x_N, M_N)$, where $\bar{x}_N := \frac{1}{N}\sum_{i=1}^{N} x_i$.

---

**Assumption 1.** *The function $f$ in problem* (1) *is approximately smooth: there exist constants $L_f, \delta_f \geq 0$ and $\bar{f}\colon \mathbb{R}^d \to \mathbb{R}$, $\bar{g}\colon \mathbb{R}^d \to \mathbb{R}^d$ such that, for any $x, y \in \mathbb{R}^d$, $\beta_{f,\bar{f},\bar{g}}(x,y) := f(y) - \bar{f}(x) - \langle \bar{g}(x), y - x \rangle$ satisfies the following inequality:* $0 \leq \beta_{f,\bar{f},\bar{g}}(x,y) \leq \frac{L_f}{2}\|x - y\|^2 + \delta_f$.

Assumption 1 is well-known in the literature under the name $(\delta, L)$-*oracle* and was originally introduced in [15]. It covers many interesting examples. For instance, if $f$ is $L$-smooth, then Assumption 1 is satisfied with $\bar{f} = f$, $\bar{g} = \nabla f$, $\delta_f = 0$ and $L_f = L$. More generally, if the function $f$ is $(\nu, H_f(\nu))$-Hölder smooth, then Assumption 1 is satisfied with $\bar{f} = f$, $\bar{g} = \nabla f$ (arbitrary selection of subgradients), any $\delta_f > 0$ and $L_f := \left[\frac{1-\nu}{2(1+\nu)\delta_f}\right]^{\frac{1-\nu}{1+\nu}}[H_f(\nu)]^{\frac{2}{1+\nu}}$ (see Theorem 13). If $f$ can be uniformly approximated by an $L$-smooth function $\phi$, i.e., $\phi(x) \leq f(x) \leq \phi(x) + \delta$, then Assumption 1 is satisfied with $\bar{f} = \phi$, $\bar{g} = \nabla\phi$ and $\delta_f = \delta$. If $f$ represents another auxiliary optimization problem with a strongly concave objective, e.g., $f(x) = \max_u \Psi(x, u)$, whose solution $\bar{u}(x)$ can only be found with accuracy $\delta$, then $f$ satisfies Assumption 1 with $\bar{f}(x) = \Psi(x, \bar{u}(x))$, $\bar{g}(x) = \nabla_u \Psi(x, \bar{u}(x))$ and $\delta_f = \delta$. For more details and other interesting examples, we refer the reader to [15].

In what follows, we assume that we have access to an unbiased stochastic oracle $\widehat{g}$ for $\bar{g}$. Formally, this is a pair $\widehat{g} = (g, \xi)$ consisting of a random variable $\xi$ and a mapping $g\colon \mathbb{R}^d \times \mathrm{Im}\,\xi \to \mathbb{R}^d$ (with $\mathrm{Im}\,\xi$ being the image of $\xi$). When queried at a point $x$, the oracle automatically generates an independent copy $\xi$ of its randomness and then returns $\widehat{g}_x = g(x, \xi)$ (notation: $\widehat{g}_x \cong \widehat{g}(x)$). We call $g$ and $\xi$ the function component and the random variable component of $\widehat{g}$, respectively. At this point, we only assume that our stochastic oracle $\widehat{g}$ is un unbiased estimator of $\bar{g}$, and later make various assumptions on its variance.

Another important assumption on problem (1), that we need in our analysis, is the boundedness of the feasible set $\mathrm{dom}\,\psi$.

**Assumption 2.** *There exists $D > 0$ such that $\|x - y\| \leq D$ for any $x, y \in \mathrm{dom}\,\psi$.*

Assumption 2 is rather standard in the literature on adaptive methods for stochastic convex optimization (see [16, 18, 30, 34, 36, 49]) and can always be ensured with $D = 2R_0$ whenever one has the knowledge of an upper bound $R_0$ on the distance from the initial point $x_0$ to the solution $x^*$. To that end, it suffices to rewrite the problem (1) in the following equivalent form: $\min_{x \in \mathrm{dom}\,\psi_D}[f(x) + \psi_D(x)]$, where $\psi_D$ is the sum of $\psi$ and the indicator function of the ball $B_0 := \{x \in \mathbb{R}^d : \|x - x_0\| \leq R_0\}$. Note that this transformation keeps the function $\psi_D$ reasonably simple as its proximal mapping can be computed via that of $\psi$ by solving a certain one-dimensional nonlinear equation, which can be done very efficiently by Newton's method (at no extra queries to the stochastic oracle); in some special cases, the corresponding nonlinear equation can even be solved analytically, e.g., when $\psi = 0$, the proximal mapping of $\psi_D$ is simply the projection on $B_0$.

Throughout this paper, we refer to $D$ from Assumption 2 as the diameter of the feasible set, and assume that its value is known to us. This will be the only essential parameter in our methods.

## 3 Main Algorithms and Stepsize Update Rules

We now present our two main algorithms for solving problem (1): UniSgd (Algorithm 1), and its accelerated version, UniFastSgd (Algorithm 2). Except the specific choice of the stepsize coefficients $M_k$, both algorithms are rather standard: the first one is the classical SGD method, and the second one is the classical accelerated gradient method for stochastic optimization [33], also known as the Method of Similar Triangles (see, e.g., Section 6.1.3 in [46]).

Both methods are expressed in terms of a certain abstract stepsize update rule $M_+(\cdot)$ defined as follows. Given the current stepsize coefficient $M \geq 0$, constant $\Omega > 0$ (the scaled squared diameter),

**Algorithm 2** $\mathrm{UniFastSgd}_{\widehat{g},\psi}(x_0; D)$

---

**Input:** Stochastic oracle $\widehat{g}$, composite part $\psi$, point $x_0 \in \mathrm{dom}\,\psi$, diameter $D$.
1: $v_0 = x_0$, $M_0 = A_0 = 0$.
2: **for** $k = 0, 1, \ldots$ **do**
3:      $a_{k+1} = \frac{1}{2}(k+1)$, $A_{k+1} = A_k + a_{k+1}$.
4:      $y_k = \frac{A_k}{A_{k+1}}x_k + \frac{a_{k+1}}{A_{k+1}}v_k$, $g_{y_k} \cong \widehat{g}(y_k)$.
5:      $v_{k+1} = \mathrm{Prox}_\psi(v_k, g_{y_k}, \frac{M_k}{a_{k+1}})$.
6:      $x_{k+1} = \frac{A_k}{A_{k+1}}x_k + \frac{a_{k+1}}{A_{k+1}}v_{k+1}$, $g_{x_{k+1}} \cong \widehat{g}(x_{k+1})$.
7:      $M_{k+1} = \frac{a_{k+1}^2}{A_{k+1}}M_+\left(\frac{A_{k+1}}{a_{k+1}^2}M_k, \frac{a_{k+1}^2}{A_{k+1}^2}D^2, y_k, x_{k+1}, g_{y_k}, g_{x_{k+1}}\right)$    $\triangleright \mathrm{e.g.,}\overset{(3)}{=}\sqrt{M_k^2 + \frac{a_{k+1}^2}{D^2}\|g_{x_{k+1}} - g_{y_k}\|_*^2}$.

---

current point $x \in \mathrm{dom}\,\psi$ with the stochastic gradient $\widehat{g}_x \cong \widehat{g}(x)$, next iterate $\widehat{x}_+ = x_+(\widehat{g}_x) \in \mathrm{dom}\,\psi$ (which is the result of the deterministic function applied to $\widehat{g}_x$), and the corresponding stochastic gradient $\widehat{g}_{x_+} \cong \widehat{g}(\widehat{x}_+)$, the update rule computes $\widehat{M}_+ = M_+(M, \Omega, x, \widehat{x}_+, \widehat{g}_x, \widehat{g}_{x_+})$ (deterministic function of its arguments) such that $\widehat{M}_+ \geq M$ and the following inequality holds for any $\overline{M} > c_2 L_f$:

$$
\begin{aligned}
&\mathbb{E}[\widehat{\Delta}(\widehat{M}_+) + (\widehat{M}_+ - M)\Omega + \beta_{f,\bar{f},\bar{g}}(\widehat{x}_+, x)] \\
&\qquad \leq \frac{c_1}{\overline{M} - c_2 L_f}\mathbb{E}[\mathrm{Var}_{\widehat{g}}(\widehat{x}_+) + \mathrm{Var}_{\widehat{g}}(x)] + c_3\delta_f + c_4\,\mathbb{E}\{[\min\{\widehat{M}_+, \overline{M}\} - M]_+\Omega\},
\end{aligned}
\tag{2}
$$

where $\widehat{\Delta}(\widehat{M}_+) := \beta_{f,\bar{f},\bar{g}}(x, \widehat{x}_+) + \langle\bar{g}(x) - \widehat{g}_x, \widehat{x}_+ - x\rangle - \frac{\widehat{M}_+}{2}\|\widehat{x}_+ - x\|^2$, $c_1, c_2, c_3, c_4 > 0$ are some absolute constants, and $\mathrm{Var}_{\widehat{g}}(x) := \mathbb{E}_\xi[\|g(x,\xi) - \bar{g}(x)\|_*^2]$ is the variance of $\widehat{g}$. The expectations in (2) are taken w.r.t. the randomness $(\xi, \xi_+)$ coming from $\widehat{g}_x \equiv g(x, \xi)$, $\widehat{g}_{x_+} \equiv g(\widehat{x}_+, \xi_+)$.

The main example is the following AdaGrad rule:

$$
\boxed{\widehat{M}_+ = \sqrt{M^2 + \frac{1}{\Omega}\|\widehat{g}_{x_+} - \widehat{g}_x\|_*^2}.}
\tag{3}
$$

For this rule, we have $c_1 = \frac{5}{2}$, $c_2 = 4$, $c_3 = 6$, $c_4 = 2$ (see Lemma 20). Another interesting example recently suggested in [49] is $\widehat{M}_+$ found from the equation

$$
(\widehat{M}_+ - M)\Omega = \left[\langle\widehat{g}_{x_+} - \widehat{g}_x, \widehat{x}_+ - x\rangle - \frac{\widehat{M}_+}{2}\|\widehat{x}_+ - x\|^2\right]_+.
\tag{4}
$$

This equation admits a unique solution which can be easily written down in closed form (see Lemma E.1 in [49]). For this rule, we have $c_1 = 1$, $c_2 = 2$, $c_3 = 6$, $c_4 = 2$ (see Lemma 21).

Inequality (2) is the only property we need from the stepsize update rule to establish all forthcoming results. This inequality is exactly what is typically used inside the convergence proofs for stochastic gradient methods with predefined stepsizes $M_k \equiv \overline{M}$ (in which case $M = \widehat{M}_+ = \overline{M}$), where $\overline{M}$ depends on problem-dependent constants. The key property of AdaGrad stepsizes (either (3) or (4)) is that they ensure the same inequality but now $\overline{M}$ is the virtual stepsize existing only in the theoretical analysis. The price for this is the extra error term $[\min\{\widehat{M}_+, \overline{M}\} - M]_+\Omega$ appearing in the right-hand side of (2). The crucial property of this error term is that it is telescopic, $\sum_{i=0}^k[\min\{M_{i+1}, \overline{M}\} - M_i]_+\Omega = [\min\{M_{k+1}, \overline{M}\} - M_0]_+\Omega$ (see Lemma 18) and therefore its total cumulative impact is always bounded by the controllable constant $\overline{M}\Omega$. Although a number of other works on theoretical analysis of AdaGrad methods for smooth optimization use some similar ideas about the virtual stepsize (e.g., [30, 34, 36]), this is the first time one has abstracted away all the technical details and identified the specific inequality (2) responsible for the universality of AdaGrad.

## 4 Uniformly Bounded Variance

In this section, we assume that the variance of our stochastic oracle is uniformly bounded.

**Assumption 3.** *For the stochastic oracle $\widehat{g}$, we have $\sigma^2 := \sup_{x \in \mathrm{dom}\,\psi}\mathrm{Var}_{\widehat{g}}(x) < +\infty$, where $\mathrm{Var}_{\widehat{g}}(x) := \mathbb{E}_\xi[\|g(x,\xi) - \bar{g}(x)\|_*^2]$.*

Under this assumption, we can establish the following efficiency estimates for our UniSgd and UniFastSgd methods (the proofs are deferred to Appendix C).

**Theorem 4.** *Let Algorithm 1 with $M_0 = 0$ be applied to problem* (1) *under Assumptions 1–3. Then, for the point $\bar{x}_N$ generated by the algorithm, we have*

$$\mathbb{E}[F(\bar{x}_N)] - F^* \leq \frac{c_2 c_4 L_f D^2}{N} + 2\sigma D \sqrt{\frac{2c_1 c_4}{N}} + c_3 \delta_f.$$

**Theorem 5.** *Let Algorithm 2 be applied to problem* (1) *under Assumptions 1–3. Then, for any $k \geq 1$,*

$$\mathbb{E}[F(x_k)] - F^* \leq \frac{4c_2 c_4 L_f D^2}{k(k+1)} + 4\sigma D \sqrt{\frac{2c_1 c_4}{3k}} + \frac{c_3}{3}(k+2)\delta_f.$$

We see that, in contrast to UniSgd, the accelerated algorithm UniFastSgd is not robust to the oracle's errors: it accumulates them with time at the rate of $O(k\delta)$. This is not surprising since the same phenomenon also occurs in the classical accelerated gradient method, even when the oracle is deterministic and the algorithm has the knowledge about all constants (see [15]).

The complexity results from Theorems 4 and 5 are similar to those from [13]. However, it is important that our methods are adaptive and do not require knowing the constants $L_f$ and $\sigma$.

In the specific case when $\delta_f = 0$, we recover the same convergence rates as in [30, 34], although our methods work for the more general composite optimization problem and, in contrast to [34], do not require that $\nabla f(x^*) = 0$.

## 5 Implicit Variance Reduction

The assumption of uniformly bounded variance may not hold for some problems, or the corresponding constant $\sigma^2$ might be quite large, which is why there has recently been a growing interest in various alternative variance bound assumptions [5, 22, 24, 29, 42, 54, 59]. One interesting option is expressing complexity bounds via the variance at the minimizer, $\sigma_*^2 := \mathrm{Var}_{\widehat{g}}(x^*)$, assuming that the stochastic oracle $\widehat{g}$ satisfies some extra smoothness conditions. Let us show that, for our Algorithms 1 and 2, we can also establish such bounds, moreover, this can be done *without any modifications to the algorithms*.

In this section, we study problem (1) under Assumptions 1 and 2 and also under the following additional smoothness assumption on the variance:

**Assumption 6.** *There exist $\delta_{\widehat{g}}, L_{\widehat{g}} \geq 0$ such that $\mathrm{Var}_{\widehat{g}}(x, y) \leq 2L_{\widehat{g}}[\beta_{f, \bar{f}, \bar{g}}(x, y) + \delta_{\widehat{g}}]$ for any $x, y \in \mathbb{R}^d$, where $\mathrm{Var}_{\widehat{g}}(x, y) := \mathbb{E}_\xi[\|[g(x, \xi) - g(y, \xi)] - [\bar{g}(x) - \bar{g}(y)]\|_*^2]$.*

Note that $\mathrm{Var}_{\widehat{g}}(x, y)$ is the usual variance of the estimator $g(x, \xi) - g(y, \xi)$ which uses the same randomness $\xi$ for both arguments. Hence, $\mathrm{Var}_{\widehat{g}}(x, y) \leq \mathbb{E}[\|g(x, \xi) - g(y, \xi)\|_*^2]$ for any $x, y$. Furthermore, if $\widehat{g}_b$ is the mini-batch version of $\widehat{g}$ of size $b$ (i.e., the average of $b$ i.i.d. samples of $\widehat{g}(x)$ at any point $x$), then $\mathrm{Var}_{\widehat{g}_b}(x, y) = \frac{1}{b} \mathrm{Var}_{\widehat{g}}(x, y)$ for any $x, y$.

For instance, if $f(x) = \mathbb{E}_\xi[f_\xi(x)]$, where each function $f_\xi$ is convex and $(\delta_\xi, L_\xi)$-approximately smooth with components $(\bar{f}_\xi, \bar{g}_\xi)$, then, the stochastic gradient oracle $\widehat{g}$, defined by $g(x, \xi) := \bar{g}_\xi(x)$ satisfies Assumption 6 with $\bar{f}(x) = \mathbb{E}_\xi[\bar{f}_\xi(x)]$, $\bar{g}(x) = \mathbb{E}_\xi[\bar{g}_\xi(x)]$, and $\delta_{\widehat{g}} = \frac{1}{L_{\max}} \mathbb{E}_\xi[L_\xi \delta_\xi]$ ($\leq \mathbb{E}_\xi[\delta_\xi]$), $L_{\widehat{g}} = L_{\max}$, where $L_{\max} := \sup_\xi L_\xi$ (see Lemma 16). Furthermore, if $\widehat{g}_b$ is the mini-batch version of $\widehat{g}$ of size $b$, then $\widehat{g}_b$ satisfies Assumption 6 with the same $\delta_{\widehat{g}_b} = \delta_{\widehat{g}}$ but $L_{\widehat{g}_b} = \frac{1}{b} L_{\widehat{g}} = \frac{1}{b} L_{\max}$ which can be much smaller than $L_{\max}$ when $b$ is large enough.

Under the new assumption on the variance, UniSgd enjoys the following convergence rate (see Appendix D.1 for the proof).

**Theorem 7.** *Let Algorithm 1 with $M_0 = 0$ be applied to problem* (1) *under Assumptions 1, 2 and 6, and let $\sigma_*^2 := \mathrm{Var}_{\widehat{g}}(x^*)$. Then, for the point $\bar{x}_N$ produced by the method, we have*

$$\mathbb{E}[F(\bar{x}_N)] - F^* \leq \frac{c_4(c_2 L_f + 12c_1 L_{\widehat{g}})D^2}{N} + 2\sigma_* D \sqrt{\frac{6c_1 c_4}{N}} + c_3 \delta_f + \frac{4}{3}\delta_{\widehat{g}}.$$

Comparing the above result with Theorem 4, we see that we have essentially replaced the uniform bound $\sigma$ with the more refined one $\sigma_*$ at the cost of replacing $L_f$ with $L_f + L_{\widehat{g}}$ and $\delta_f$ with $\delta_f + \delta_{\widehat{g}}$.

---
**Algorithm 3** $\text{UniSvrg}_{\widehat{g},\bar{g},\psi}(x_0; D)$
---
**Input:** Oracles $\widehat{g}, \bar{g}$, composite part $\psi$, point $x_0 \in \text{dom } \psi$, diameter $D$.
 1: $\tilde{x}_0 = x_0, M_0 = 0$.
 2: **for** $t = 0, 1, \ldots$ **do**
 3:    $(\tilde{x}_{t+1}, x_{t+1}, M_{t+1}) \cong \text{UniSgd}_{\widehat{G}_t, \psi}(x_t, M_t, 2^{t+1}; D)$ with $\widehat{G}_t = \text{SvrgOrac}_{\widehat{g},\bar{g}}(\tilde{x}_t)$.
---

This corresponds to classical results on the usual SGD for which we know all problem dependent-constants. However, our method is universal and works automatically under both assumptions from the previous section and the current one, and therefore enjoys the best among the rates given by Theorems 4 and 7.

For the accelerated algorithm, we have the following result (whose proof is located in Appendix D.2).

**Theorem 8.** *Let Algorithm 2 be applied to problem* (1) *under Assumptions 1, 2 and 6, and let* $\sigma_*^2 := \text{Var}_{\widehat{g}}(x^*)$. *Then, for any* $k \geq 1$, *we have*

$$\mathbb{E}[F(x_k)] - F^* \leq \frac{4c_2 c_4 L_f D^2}{k(k+1)} + \frac{24 c_1 c_4 L_{\widehat{g}} D^2}{k+1} + 4\sigma_* D \sqrt{\frac{2c_1 c_4}{k}} + \frac{c_3}{3}(k+2)\delta_f + \frac{4}{3}\delta_{\widehat{g}}.$$

Comparing our previous complexity bound for $\text{UniFastSgd}$ under the assumption on uniformly bounded variance (Theorem 5) with the bound from Theorem 8, we see that, instead of simply replacing $\sigma$ with $\sigma_*$, $L_f$ with $L_f + L_{\widehat{g}}$ and $\delta_f$ with $\delta_f + \delta_{\widehat{g}}$, which was the case for the basic method, the situation is now not that simple. Specifically, the $L_f$ and $L_{\widehat{g}}$ terms now converge at different rates: $O(\frac{1}{k^2})$ and $O(\frac{1}{k})$, respectively. While this may seem strange at first, this behavior is actually unavoidable, at least in the case when $\delta_f = \delta_{\widehat{g}} = 0$ (see, e.g., Section E in [59]). For the case when $\delta_f = \delta_{\widehat{g}} = 0$, the complexity result from Theorem 8 is similar to the results for the Accelerated SGD algorithm from [59]. However, the latter paper studies a specific setting where $f(x) = \mathbb{E}[f_\xi(x)]$, where each component $f_\xi$ is $L_{\max}$-smooth and then assumes that $f$ is also $L_{\max}$-smooth, instead of working with the constant $L_f$ which can be much smaller than $L_{\max}$. A similar separation of the constants $L_f$ and $L_{\widehat{g}}$, which we do, was recently considered in [24], where the authors obtained some similar rates to our Theorem 8. However, it is important that, unlike the algorithms considered in [24, 59], our $\text{UniFastSgd}$ is universal and does not require knowing any problem-dependent constants except $D$. Furthermore, our results are more general because we allow the oracle to be inexact.

## 6 Explicit Variance Reduction with SVRG

Let us now show that we can also incorporate explicit SVRG-type variance reduction into our methods. In this section, we consider problem (1) under Assumptions 1, 2 and 6. All the proofs are deferred to Appendix E.

In addition to the stochastic oracle $\widehat{g}$, we now assume that we can also compute the (approximate) full-gradient oracle $\bar{g}$. This allows us to define the following auxiliary *SVRG oracle* induced by $\widehat{g}$ with center $\tilde{x} \in \mathbb{R}^d$ (notation $\widehat{G} = \text{SvrgOrac}_{\widehat{g},\bar{g}}(\tilde{x})$) as the oracle with the same random variable component $\xi$ as $\widehat{g}$ and the function component given by $G(x, \xi) = g(x, \xi) - g(\tilde{x}, \xi) + \bar{g}(\tilde{x})$.

Our $\text{UniSvrg}$ method is presented in Algorithm 3. This is the classical epoch-based SVRG algorithm which can be seen as the adaptive version of the SVRG++ method from [1]. A similar scheme was suggested in [16], however, instead of accumulating gradient differences as in (3), their method accumulates gradients and therefore does not work without the additional assumption of $\nabla f(x^*) = 0$ (which may not hold for constrained optimization).

Let us now present the complexity guarantees. To do so, we first need to introduce, one more assumption we need in our analysis.

**Assumption 9.** *The variance of* $\widehat{g}$ *satisfies* $\text{Var}_{\widehat{g}}(x, y) \leq 4L_{\widehat{g}}[\beta_f^{\nabla f(x)}(x, y) + 2\delta_{\widehat{g}}]$ *for any* $x, y \in \mathbb{R}^d$ *and any* $\nabla f(x) \in \partial f(x)$.

Assumption 9 is very similar to Assumption 6. The only difference between them is that the former contains the standard Bregman distance in the right-hand side, while the latter contains its approximation $\beta_{f,\bar{f},\bar{g}}(x, y)$ involving the approximate function value $\bar{f}(x)$ and the approximate gradient $\bar{g}(x)$. Nevertheless, both assumptions are actually satisfied for the main examples we discussed after introducing Assumption 6 (see Lemma 16).

**Algorithm 4** $\text{UniFastSvrg}_{\widehat{g},\bar{g},\psi}(x_0, N; D)$

**Input:** Oracles $\widehat{g}$, $\bar{g}$, composite part $\psi$, point $x_0 \in \text{dom } \psi$, epoch length $N$, diameter $D$.
1: $\tilde{x}_0 = \text{Prox}_\psi(x_0, \bar{g}(x_0), 0)$, $v_0 = x_0$, $M_0 = 0$, $A_0 = \frac{1}{N}$.
2: **for** $t = 0, 1, \dots$ **do**
3: $\quad a_{t+1} = \sqrt{A_t}$, $A_{t+1} = A_t + a_{t+1}$.
4: $\quad (\tilde{x}_{t+1}, v_{t+1}, M_{t+1}) \cong \text{UniTriSvrgEpoch}_{\widehat{g},\bar{g},\psi}(\tilde{x}_t, v_t, M_t, A_t, a_{t+1}, N; D)$.

---

**Algorithm 5** $(\tilde{x}_+, v_+, M_+) \cong \text{UniTriSvrgEpoch}_{\widehat{g},\bar{g},\psi}(\tilde{x}, v_0, M_0, A, a, N; D)$

**Input:** Oracles $\widehat{g}$, $\bar{g}$, comp. part $\psi$, points $\tilde{x}, v_0$, coefficients $M_0, A, a$, epoch length $N$, diameter $D$.
1: $A_+ = A + a$, $x_0 = \frac{A}{A_+}\tilde{x} + \frac{a}{A_+}v_0$, $\widehat{G} = \text{SvrgOrac}_{\widehat{g},\bar{g}}(\tilde{x})$, $G_{x_0} \cong \widehat{G}(x_0)$.
2: **for** $k = 0, \dots, N-1$ **do**
3: $\quad v_{k+1} = \text{Prox}_\psi(v_k, G_{x_k}, \frac{M_k}{a})$.
4: $\quad x_{k+1} = \frac{A}{A_+}\tilde{x} + \frac{a}{A_+}v_{k+1}$, $G_{x_{k+1}} \cong \widehat{G}(x_{k+1})$.
5: $\quad M_{k+1} = \frac{a^2}{A_+}M_+\big(\frac{A_+}{a^2}M_k, \frac{a^2}{A_+^2}D^2, x_k, x_{k+1}, G_{x_k}, G_{x_{k+1}}\big)$ $\qquad \triangleright$ e.g., $\stackrel{(3)}{=} \sqrt{M_k^2 + \frac{a^2}{D^2}\|G_{x_{k+1}} - G_{x_k}\|_*^2}$.
6: **return** $(\bar{x}_N, v_N, M_N)$, where $\bar{x}_N := \frac{1}{N}\sum_{k=1}^N x_k$.

---

**Theorem 10.** *Let* $\text{UniSvrg}$ *(as defined by Algorithm 3) be applied to problem* (1) *under Assumptions 1, 2, 6 and 9. Then, for any $t \geq 1$ and $\bar{c}_3 := \max\{c_3, 1\}$, we have*

$$\mathbb{E}[F(\tilde{x}_t)] - F^* \leq \frac{[(c_2 c_4 + 1)L_f + 48c_1 c_4 L_{\widehat{g}}]D^2}{2^t} + 2\bar{c}_3\delta_f + \frac{8}{3}\delta_{\widehat{g}}.$$

*To construct $\tilde{x}_t$, the algorithm needs to make $O(2^t)$ queries to $\widehat{g}$ and $O(t)$ queries to $\bar{g}$.*

We now present an accelerated version of UniSvrg, see Algorithm 4. As UniSvrg, this method is also epoch-based, and its epoch is very similar to UniFastSgd (Algorithm 4) in the sense that it also iterates similar-triangle steps. However, the triangles in UniTriSvrgEpoch are of the form $(\tilde{x}, v_k, v_{k+1})$, i.e., they always share the common vertex $\tilde{x}$, in contrast to the triangles $(x_k, v_k, v_{k+1})$ in UniFastSgd (in UniTriSvrgEpoch, the role of the average points $y_k$ is played by $x_k$). We note that our UniFastSvrg is essentially the primal version of the VRADA method from [53], but equipped with AdaGrad stepsizes. Alternative accelerated SVRG schemes with AdaGrad stepsizes (3) were recently proposed in [36]; however, they seem to be much more complicated.

The special choice of the initial reference point $\tilde{x}_0$ at Line 1 is rather standard and motivated by the desire to keep the initial function residual appropriately bounded: $F(\tilde{x}_0) - F^* \leq \frac{1}{2}L_f D^2 + \delta_f$; the simplest way to achieve this is to make the full gradient step from any feasible point (see Lemma 34).

**Theorem 11.** *Let* $\text{UniFastSvrg}$ *(Algorithm 4) be applied to problem* (1) *under Assumptions 1, 2 and 6, and let $N \geq 9$. Then, for any $t \geq t_0 := \lceil \log_2 \log_3 N \rceil - 1 \ (\geq 0)$, it holds that*

$$\mathbb{E}[F(\tilde{x}_t)] - F^* \leq \frac{9[(c_2 c_4 + \frac{1}{2})L_f + 6c_1 c_4 L_{\widehat{g}}]D^2}{N(t - t_0 + 1)^2} + (c_3 t + 1)\delta_f + \frac{5}{3}t\delta_{\widehat{g}}.$$

*To construct $\tilde{x}_t$, the algorithm needs to make $O(Nt)$ queries to $\widehat{g}$ and $O(t)$ queries to $\bar{g}$. Assuming that the complexity of querying $\bar{g}$ is $n$ times bigger than that of querying $\widehat{g}$ and choosing $N = \Theta(n)$, we get the total stochastic-oracle complexity of $O(nt)$.*

Note that Theorem 11, unlike Theorem 10, does not require the extra Assumption 9. This suggests that Assumption 9 might be somewhat artificial and could potentially be removed from Theorem 10 as well. However, we do not know how to do it, even in the simplest case when $\delta_f = \delta_{\widehat{g}} = 0$ and the algorithm has the knowledge of the constants $L_f$ and $L_{\widehat{g}}$ from Assumptions 1 and 6.

## 7 Application to Hölder Smooth Problems

To illustrate how powerful our results are, let us quickly consider the specific example of solving the stochastic optimization problem with Hölder smooth components.

*Example* 12. Suppose that the function $f$ in problem (1) is the expectation of other functions, $f(x) = \mathbb{E}_\xi[f_\xi(x)]$, where each function $f_\xi$ is convex and $(\nu, H_\xi(\nu))$-Hölder smooth. Consider the

Table 2: Corollaries of our results for the case when problem (1) has Hölder smooth components, as defined in Example 12. "SO complexity" is the stochastic-oracle complexity for reaching accuracy $\epsilon$ in terms of the expected function residual, defined as in Table 1 but with $\widehat{g} = \widehat{g}_b$, $\bar{g} = \nabla f$, $n = n_b$.

| Method | SO complexity | Reference |
|---|---|---|
| UniSgd (Alg. 1) | $\left(\frac{H_f(\nu)}{\epsilon}\right)^{\frac{2}{1+\nu}} D^2 + \frac{1}{b}\min\left\{\frac{\sigma^2 D^2}{\epsilon^2}, \left(\frac{H_{\max}(\nu)}{\epsilon}\right)^{\frac{2}{1+\nu}} D^2 + \frac{\sigma_*^2 D^2}{\epsilon^2}\right\}$ | Cors. 37, 40 |
| UniFastSgd (Alg. 2) | $\left(\frac{H_f(\nu) D^{1+\nu}}{\epsilon}\right)^{\frac{2}{1+3\nu}} + \frac{1}{b}\min\left\{\frac{\sigma^2 D^2}{\epsilon^2}, \left(\frac{H_{\max}(\nu)}{\epsilon}\right)^{\frac{2}{1+\nu}} D^2 + \frac{\sigma_*^2 D^2}{\epsilon^2}\right\}$ | Cors. 38, 41 |
| UniSvrg (Alg. 3) | $\left[N_\nu(\epsilon) := \left(\frac{H_f(\nu)}{\epsilon}\right)^{\frac{2}{1+\nu}} D^2 + \frac{1}{b}\left(\frac{H_{\max}(\nu)}{\epsilon}\right)^{\frac{2}{1+\nu}} D^2\right] + n_b \log_+ N_\nu(\epsilon)$ | Cor. 43 |
| UniFastSvrg (Alg. 4) | $\left[\frac{n_b^\nu H_f(\nu) D^{1+\nu}}{\epsilon}\right]^{\frac{2}{1+3\nu}} + \left[\frac{n_b^\nu H_{\max}(\nu) D^{1+\nu}}{b^{(1+\nu)/2}\epsilon}\right]^{\frac{2}{1+3\nu}} + n_b \log\log n_b$ | Cor. 44 |

standard mini-batch stochastic gradient oracle $\widehat{g}_b$ of size $b$, defined by $g_b(x, \xi_{[b]}) = \frac{1}{b}\sum_{j=1}^{b} \nabla f_{\xi_j}(x)$, where $\xi_{[b]} := (\xi_1, \ldots, \xi_b)$ with $b$ i.i.d. copies of $\xi$, and $\nabla f_\xi(x) \in \partial f_\xi(x)$ is an arbitrary selection of subgradients for each $\xi$. We define $H_f(\nu)$ as the Hölder constant for the function $f$ and $H_{\max}(\nu) := \sup_\xi H_\xi(\nu)$ as the worst among Hölder constants for each $f_\xi$. Note that we always have $H_f(\nu) \leq \mathbb{E}_\xi[H_\xi(\nu)]$ but $H_f(\nu)$ can, in principle, be much smaller than the right-hand side. Also, define $\sigma^2 := \sup_{x\in\operatorname{dom}\psi} \operatorname{Var}_{\widehat{g}_1}(x) \equiv \sup_{x\in\operatorname{dom}\psi} \mathbb{E}_\xi[\|\nabla f_\xi(x) - \nabla f(x)\|_*^2]$ and $\sigma_*^2 := \operatorname{Var}_{\widehat{g}_1}(x^*) \equiv \mathbb{E}_\xi[\|\nabla f_\xi(x^*) - \nabla f(x^*)\|_*^2]$. We assume that the computation of $\widehat{g}_b$ can be parallelized and the computation of $\nabla f$ is $n_b$ times more expensive than that of $\widehat{g}_b$.

To solve the above problem, we can apply any of the methods we presented before. The resulting oracle complexities (in terms of the BigO-notation) are summarized in Table 2; the precise statements the corresponding results and their proofs are deferred to Appendix F.

Note that our problem is characterized by a large number of parameters, $\nu$, $H_f(\nu)$, $H_{\max}(\nu)$, $\sigma$, $\sigma_*$. For each combination of these parameters, we get a certain complexity guarantee for each of our methods, and it is impossible to say in advance which combination results in the smaller complexity bound. However, it is not important for our methods since none of them needs to know any of these constants to ensure the corresponding bound. This means that our algorithms are *universal*: they automatically figure out the best problem class for a specific problem given to them.

## 8 Experiments

Let us illustrate the performance of our methods in preliminary numerical experiments[2] on solving

$$f^* := \min_{\|x\| \leq R}\left\{f(x) := \frac{1}{n}\sum_{i=1}^{n}[\langle a_i, x\rangle - b_i]_+^q\right\}, \tag{5}$$

where $a_i, b_i \in \mathbb{R}^d$, $q \in [1, 2]$ and $R > 0$.

This test problem covers several interesting applications. Indeed, if $q = 2$, we get the classical Least squares problem. If $q = 1$, this is the well-known Support-Vector Machines (SVM) problem. In both cases, the ball-constraint $\|x\| \leq R$ acts as a regularizer, and problem (5) is, in fact, equivalent to $\min_{x\in\mathbb{R}^d}[f(x) + \frac{\mu}{2}\|x\|^2]$ for a certain $\mu \geq 0$ (this follows, e.g., from the KKT optimality conditions) such that $\mu$ decreases when $R$ increases.

Another interesting application of (5), which we consider in this section, is the *polyhedron feasibility problem*: find $x^* \in \mathbb{R}^d$, $\|x^*\| \leq R$, inside the polyhedron $P = \{x : \langle a_i, x\rangle \leq b_i, i = 1, \ldots, n\}$. Such a point exists iff $f^* = 0$. Note that (5) is a problem with Hölder smooth components of degree $\nu = q - 1$. By varying $q$ in (5), we can therefore check the adaptivity of different methods to the unknown to them Hölder characteristics of the objective function.

The data for our problem is generated randomly. First, we generate $x^*$ uniformly from the sphere of radius $0.95R$ centered at the origin. Then, we generate i.i.d. vectors $a_i$ with components uniformly distributed on $[-1, 1]$. We then make sure that $\langle a_n, x^*\rangle < 0$ by inverting the sign of $a_n$ if necessary. Next, we generate positive reals $s_i$ uniformly in $[0, -0.1c_{\min}]$, where $c_{\min} := \min_i\langle a_i, x^*\rangle < 0$, and set $b_i = \langle a_i, x^*\rangle + s_i$. By construction, $x^*$ is a solution of our problem with $f^* = 0$, and the origin $x_0 = 0$ lies outside the polyhedron since there exists $j$ (corresponding to $c_{\min}$) such that $b_j = c_{\min} + s_j \leq 0.9c_{\min} < 0$.

---

[2]The corresponding source code is available at https://github.com/mlolab/universal-adagrad-experiments.

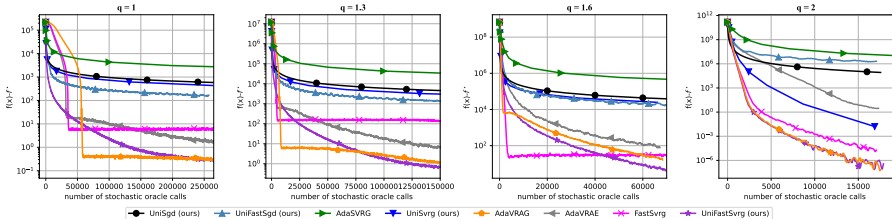

Figure 1: Comparison of different methods on the polyhedron feasibility problem (5).

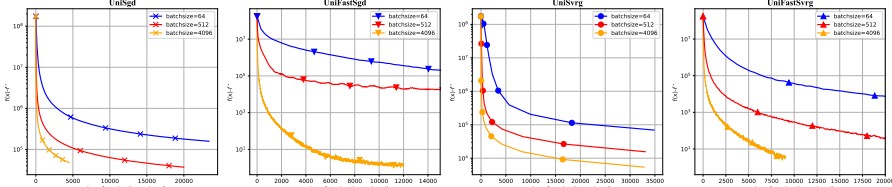

Figure 2: Impact of mini-batch size on performance of our methods.

We compare UniSvrg (Algorithm 3) against AdaSVRG [16] (with parameters $K = 3$ and $\eta = D = 2R$). We next compare UniFastSvrg (Algorithm 4) against AdaVRAE and AdaVRAG [36]. We also compare it with the FastSvrg method with constant stepsize, which is the primal version of the VRADA method from [53]; the stepsize is selected by doing a grid search over $\{10^j : j = -3, \ldots, 4\}$ and choosing the best value in the sense that the algorithm is neither too slow nor has a large error. We report UniSgd (Algorithm 1) and UniFastSgd (Algorithm 2) together with these methods. For UniFastSvrg, contrary to the theoretical recommendation of choosing $\tilde{x}_0$ as the result of the full gradient step, we found it slightly more useful to simply set $\tilde{x}_0 = x_0$. For all our methods, we use the AdaGrad stepsize (3); the other stepsize (4) works very similarly (see Appendix H.2 for a detailed comparison). For all methods, we use the standard mini-batch stochastic oracle of size $b = 256$.

The results are shown in Fig. 1, where we fix $n = 10^4$, $d = 10^3$, $R = 10^6$ and consider different values of $q \in \{1, 1.3, 1.6, 2\}$. We plot the total number of stochastic oracle calls against the function residual. We treat one mini-batch oracle computation as one stochastic oracle call. If we compute the full gradient, we count this as $n/b$ stochastic oracle calls where $n$ is the total number of samples and $b$ denotes the mini-batch size.

We see that, except the AdaSVRG method, all SVRG algorithms typically converge much faster than the usual SGD methods without explicit variance reduction, at least after a few computations of the full gradient. Among the non-accelerated SVRG methods, UniSvrg converges consistently faster than AdaSVRG, while UniFastSvrg performs the best across the accelerated ones. Note that FastSvrg with constant stepsize is not converging when the problem is not Lipschitz smooth ($q < 2$), in contrast to our universal methods.

In Fig. 2, we also illustrate the impact of the mini-batch size $b$ on the convergence of our methods. We consider the same values of $n$, $d$, $R$ as before and fix $q = 1.5$. As we can see, in the idealized situation, when one can implement the mini-batch oracle computations by perfect parallelism, there is a significant speedup in convergence when increasing the mini-batch size, as predicted by our theory.

For additional experiments, including the discussion of implicit variance reduction, see Appendix H.

## 9   Conclusions

In this paper, we showed that AdaGrad stepsizes can be applied, in a unified manner, in a large variety of situations, leading to universal methods suitable for multiple problem classes at the same time. Note that this does not come for free. We still need to know one parameter, the diameter $D$ of the feasible set. While it is not necessary to know this parameter precisely, the cost of underestimating or overestimating it, can be high (all complexity bounds would be multiplied by the ratio between our guess and the true $D$). At the same time, there already exist some parameter-free methods which are based on AdaGrad and aim to solve precisely this problem [6, 11, 25, 31, 41]. It is therefore interesting to consider extensions of our results to these more advanced algorithms. Another interesting direction is, of course, nonconvex problems.

## Acknowledgements

The authors are thankful to the anonymous reviewers for their comments and suggestions. Sebastian Stich acknowledges funding support from the Meta Research Award and the Google Research Award.

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

# A General Auxiliary Results

## A.1 Approximately Smooth Functions

**Theorem 13** (Lemma 2 in [45]). *Let $f \colon \mathbb{R}^d \to \mathbb{R}$ be a convex $(\nu, H)$-Hölder smooth function for some $\nu \in [0, 1]$ and $H \geq 0$. Then, for any $\delta > 0$, any $x, y \in \mathbb{R}^d$ and any $\nabla f(x) \in \partial f(x)$, it holds that $\beta_f^{\nabla f(x)}(x, y) \leq \frac{L}{2} \|x - y\|^2 + \delta$ with $L = [\frac{1-\nu}{2(1+\nu)\delta}]^{\frac{1-\nu}{1+\nu}} H^{\frac{2}{1+\nu}}$ (with the convention that $0^0 = 1$).*

**Theorem 14.** *Let $f \colon \mathbb{R}^d \to \mathbb{R}$ be a $(\delta, L)$-approximately smooth convex function with components $(\bar{f}, \bar{g})$, i.e., for any $x, y \in \mathbb{R}^d$ and $\beta_{f, \bar{f}, \bar{g}}(x, y) := f(y) - \bar{f}(x) - \langle \bar{g}(x), y - x \rangle$, we have*

$$0 \leq \beta_{f, \bar{f}, \bar{g}}(x, y) \leq \frac{L}{2} \|x - y\|^2 + \delta. \tag{6}$$

*Then, for any $x, y \in \mathbb{R}^d$ and any $\nabla f(x) \in \partial f(x)$, the following inequalities hold:*

$$\bar{f}(x) \leq f(x) \leq \bar{f}(x) + \delta, \tag{7}$$

$$\langle \bar{g}(x) - \bar{g}(y), x - y \rangle \leq \beta_{f, \bar{f}, \bar{g}}(x, y) + \beta_{f, \bar{f}, \bar{g}}(y, x) \leq \langle \bar{g}(x) - \bar{g}(y), x - y \rangle + 2\delta, \tag{8}$$

$$\langle \bar{g}(x) - \bar{g}(y), x - y \rangle \leq L \|x - y\|^2 + 2\delta, \tag{9}$$

$$\|\bar{g}(x) - \bar{g}(y)\|_*^2 \leq 2L(\beta_{f, \bar{f}, \bar{g}}(x, y) + \delta), \tag{10}$$

$$\|\nabla f(x) - \bar{g}(y)\|_*^2 \leq 2L(\beta_f^{\nabla f(x)}(x, y) + \delta), \tag{11}$$

$$\|\bar{g}(x) - \bar{g}(y)\|_*^2 \leq L^2 \|x - y\|^2 + 4L\delta, \tag{12}$$

$$\|\bar{g}(x) - \bar{g}(y)\|_*^2 \leq 4L(\beta_f^{\nabla f(x)}(x, y) + 2\delta), \tag{13}$$

$$\beta_f^{\nabla f(x)}(x, y) \leq L \|x - y\|^2 + 2\delta. \tag{14}$$

*Proof.* Inequality (7) follows immediately from (6) by substituting $y = x$.

To prove (8), we rewrite

$$\beta_{f, \bar{f}, \bar{g}}(x, y) + \beta_{f, \bar{f}, \bar{g}}(y, x) = \langle \bar{g}(x) - \bar{g}(y), x - y \rangle + [f(x) - \bar{f}(x)] + [f(y) - \bar{f}(y)],$$

and then apply (7).

Using the first part of (8) and applying (6) twice, we obtain (9).

To prove (10) and (11), let us fix some $\bar{f}_1(x) \in \mathbb{R}$ and $\bar{g}_1(x) \in \mathbb{R}^d$ such that $\beta_{f, \bar{f}_1, \bar{g}_1}(z) := f(z) - \bar{f}_1(x) - \langle \bar{g}_1(x), z - x \rangle \geq 0$ for any $z \in \mathbb{R}^d$. Note that we can choose either $(\bar{f}_1, \bar{g}_1) = (\bar{f}, \bar{g})$ or $(\bar{f}_1, \bar{g}_1) = (f, \nabla f)$. In view of (6), for any $z \in \mathbb{R}^d$, we can write the following inequalities:

$$0 \leq \beta_{f, \bar{f}_1, \bar{g}_1}(z) \leq \bar{f}(y) - \bar{f}_1(x) - \langle \bar{g}_1(x), y - x \rangle + \langle \bar{g}(y) - \bar{g}_1(x), z - y \rangle + \frac{L}{2} \|z - y\|^2 + \delta.$$

Minimizing the right-hand side in $z \in \mathbb{R}^d$ and rearranging, we conclude that

$$\frac{1}{2L} \|\bar{g}(y) - \bar{g}_1(x)\|_*^2 \leq \bar{f}(y) - \bar{f}_1(x) - \langle \bar{g}_1(x), y - x \rangle + \delta \leq \beta_{f, \bar{f}_1, \bar{g}_1}(x, y) + \delta,$$

where the final inequality is due to (7). Substituting now either $(\bar{f}_1, \bar{g}_1) = (\bar{f}, \bar{g})$ or $(\bar{f}_1, \bar{g}_1) = (f, \nabla f)$, we obtain either (10) or (11), respectively.

Inequality (12) follows immediately from (6) and (10).

Inequality (13) follows from (11):

$$\begin{aligned}
\|\bar{g}(x) - \bar{g}(y)\|_*^2 &\leq 2 \|\nabla f(x) - \bar{g}(y)\|_*^2 + 2 \|\bar{g}(x) - \nabla f(x)\|_*^2 \\
&\leq 4L(\beta_f^{\nabla f(x)}(x, y) + \delta) + 4L\delta = 4L(\beta_f^{\nabla f(x)}(x, y) + 2\delta).
\end{aligned}$$

To prove (14), we proceed as follows using first (6), then (7), and then (11):

$$\beta_f^{\nabla f(x)}(x, y) \equiv f(y) - f(x) - \langle \nabla f(x), y - x \rangle$$

$$\leq \bar{f}(x) - f(x) + \langle \bar{g}(x) - \nabla f(x), y - x \rangle + \frac{L}{2} \|y - x\|^2 + \delta$$

$$\leq \langle \bar{g}(x) - \nabla f(x), y - x \rangle + \frac{L}{2} \|y - x\|^2 + \delta$$

$$\leq \sqrt{2L\delta} \|y - x\| + \frac{L}{2} \|y - x\|^2 + \delta$$

$$= \left( \sqrt{\frac{L}{2}} \|y - x\| + \sqrt{\delta} \right)^2 \leq L\|y - x\|^2 + 2\delta,$$

where the final inequality is $(a + b)^2 \leq 2a^2 + 2b^2$, $a, b \geq 0$. $\square$

*Remark* 15. Some of the inequalities from Theorem 14, namely, (7), (10) and (12), were established already in [15]. We nevertheless prefer to present the corresponding proofs since they are rather simple, and we use the associated ideas for proving the other new inequalities.

**Lemma 16.** *Let* $f \colon \mathbb{R}^d \to \mathbb{R}$ *be the function* $f(x) := \mathbb{E}_\xi[f_\xi(x)]$, *where each* $f_\xi \colon \mathbb{R}^d \to \mathbb{R}$ *is convex and* $(\delta_\xi, L_\xi)$-*approximately smooth with components* $(\bar{f}_\xi, \bar{g}_\xi)$. *Further, let* $\hat{g}$ *be the stochastic oracle defined by* $g(x, \xi) := \bar{g}_\xi(x)$, *and let* $\bar{f}(x) := \mathbb{E}_\xi[\bar{f}_\xi(x)]$, $\bar{g}(x) := \mathbb{E}_\xi[\bar{g}_\xi(x)]$. *Then,* $\hat{g}$ *is an unbiased oracle for* $\bar{g}$ *and, for any* $x, y \in \mathbb{R}^d$, $L_{\max} := \sup_\xi L_\xi$ *and* $\bar{\delta} := \frac{1}{L_{\max}} \mathbb{E}_\xi[L_\xi \delta_\xi]$, *it holds that*

$$\mathrm{Var}_{\hat{g}}(x, y) \leq 2L_{\max}[\beta_{f, \bar{f}, \bar{g}}(x, y) + \bar{\delta}]. \tag{15}$$

*Furthermore, for any* $x, y \in \mathbb{R}^d$ *and any* $\nabla f(x) \in \partial f(x)$, *it also holds that*

$$\mathrm{Var}_{\hat{g}}(x, y) \leq 4L_{\max}[\beta_f^{\nabla f(x)}(x, y) + 2\bar{\delta}]. \tag{16}$$

*Proof.* According to our definition of $\bar{g}$, we have $\mathbb{E}_\xi[\bar{g}_\xi(x)] = \bar{g}(x)$ for any $x$, so $\hat{g}$ is indeed an unbiased oracle for $\bar{g}$. Further, for any $x, y \in \mathbb{R}^d$, we can estimate

$$\mathrm{Var}_{\hat{g}}(x, y) \equiv \mathbb{E}_\xi \left[ \| [\bar{g}_\xi(x) - \bar{g}_\xi(y)] - [\bar{g}(x) - \bar{g}(y)] \|_*^2 \right]$$

$$\leq \mathbb{E}_\xi \left[ \|\bar{g}_\xi(x) - \bar{g}_\xi(y)\|_*^2 \right] \leq \mathbb{E}_\xi \left[ 2L_\xi \left( \beta_{f_\xi, \bar{f}_\xi, \bar{g}_\xi}(x, y) + \delta_\xi \right) \right]$$

$$\leq 2L_{\max} \left( \mathbb{E}_\xi[\beta_{f_\xi, \bar{f}_\xi, \bar{g}_\xi}(x, y)] + \bar{\delta} \right) = 2L_{\max}[\beta_{f, \bar{f}, \bar{g}}(x, y) + \bar{\delta}],$$

where $\bar{\delta}$ is as defined in the statement; the second inequality follows from Theorem 14 (inequality (10)), and the final identity is due to the linearity of $\beta_{f, \bar{f}, \bar{g}}(x, y)$ in $(f, \bar{f}, \bar{g})$ and the fact that, by our definitions, $\mathbb{E}_\xi[f_\xi(x)] = f(x)$, $\mathbb{E}_\xi[\bar{f}_\xi(x)] = \bar{f}(x)$, $\mathbb{E}_\xi[\bar{g}_\xi(x)] = \bar{g}(x)$ for any $x$. This proves (15).

The proof of (16) is similar but now we apply (13) instead of (10):

$$\mathrm{Var}_{\hat{g}}(x, y) \leq \mathbb{E}_\xi \left[ \|\bar{g}_\xi(x) - \bar{g}_\xi(y)\|_*^2 \right] \leq \mathbb{E}_\xi \left[ 4L_\xi \left( \beta_{f_\xi}^{\nabla f_\xi(x)}(x, y) + 2\delta_\xi \right) \right]$$

$$\leq 4L_{\max} \left( \mathbb{E}_\xi[\beta_{f_\xi}^{\nabla f_\xi(x)}(x, y)] + 2\bar{\delta} \right) = 4L_{\max}[\beta_f^{\nabla f(x)}(x, y) + 2\bar{\delta}],$$

where we have used the fact that $\partial f(x) = \mathbb{E}_\xi[\partial f_\xi(x)]$ (see Proposition 2.2 in [4]), meaning that, for any $\nabla f(x) \in \partial f(x)$, we can find a selection of $\nabla f_\xi(x) \in \partial f_\xi(x)$ such that $\nabla f(x) = \mathbb{E}_\xi[\nabla f_\xi(x)]$. $\square$

## A.2 Miscellaneous

**Lemma 17.** *Let* $\psi \colon \mathbb{R}^d \to \mathbb{R} \cup \{+\infty\}$ *be a proper closed convex function,* $x \in \mathrm{dom}\,\psi$, $g \in \mathbb{R}^d$, $M \geq 0$, *and let*

$$x_+ := \mathrm{Prox}_\psi(x, g, M).$$

*Then, for any* $y \in \mathrm{dom}\,\psi$, *we have*

$$\langle g, y - x_+ \rangle + \psi(y) + \frac{M}{2} \|x - y\|^2 \geq \psi(x_+) + \frac{M}{2} \|x - x_+\|^2 + \frac{M}{2} \|x_+ - y\|^2.$$

*Proof.* Indeed, by definition, $x_+$ is the minimizer of the function $h: \mathbb{R}^d \to \mathbb{R} \cup \{+\infty\}$ given by $h(y) := \langle g, y \rangle + \psi(y) + \frac{M}{2}\|x - y\|^2$, which is strongly convex with parameter $M$ (or simply convex if $M = 0$). Hence, for any $y \in \operatorname{dom} \psi \,(= \operatorname{dom} h)$, we have $h(y) \geq h(x_+) + \frac{M}{2}\|y - x_+\|^2$, which is exactly the claimed inequality. $\square$

**Lemma 18.** *Let $N \geq 1$ be an integer, $(M_k)_{k=0}^N$ be a nondecreasing nonnegative sequence of reals, and let $\overline{M} \geq 0$. Then,*

$$\sum_{k=0}^{N-1} [\min\{M_{k+1}, \overline{M}\} - M_k]_+ = [\min\{M_N, \overline{M}\} - M_0]_+.$$

*Proof.* It suffices to prove the identity only in the special case when $N = 2$, i.e., to show that $\gamma_0 + \gamma_1 = \Gamma$, where $\gamma_0 := [\min\{M_1, \overline{M}\} - M_0]_+$, $\gamma_1 := [\min\{M_2, \overline{M}\} - M_1]_+$, $\Gamma := [\min\{M_2, \overline{M}\} - M_0]_+$. The general case then easily follows by induction.

To prove the identity, we use our assumption that $M_0 \leq M_1 \leq M_2$ and consider three possible cases. If $M_1 \geq \overline{M}$, then $\gamma_0 + \gamma_1 = [\overline{M} - M_0]_+ + 0 = [\overline{M} - M_0]_+ = \Gamma$. If $M_1 < \overline{M} \leq M_2$, then $\gamma_0 + \gamma_1 = (M_1 - M_0) + (\overline{M} - M_1) = \overline{M} - M_0 = \Gamma$. Finally, if $M_2 < \overline{M}$, then $\gamma_0 + \gamma_1 = (M_1 - M_0) + (M_2 - M_1) = M_2 - M_0 = \Gamma$. $\square$

**Lemma 19.** *Let $\widehat{g}$ be a stochastic oracle in $\mathbb{R}^d$. Then, for any $x, y, z \in \mathbb{R}^d$ and any $\tau > 0$, we have*

$$\operatorname{Var}_{\widehat{g}}(x) \leq (1 + \tau) \operatorname{Var}_{\widehat{g}}(y) + (1 + \tau^{-1}) \operatorname{Var}_{\widehat{g}}(x, y),$$
$$\operatorname{Var}_{\widehat{g}}(x, y) \leq (1 + \tau) \operatorname{Var}_{\widehat{g}}(x, z) + (1 + \tau^{-1}) \operatorname{Var}_{\widehat{g}}(y, z).$$

*Proof.* Both inequalities are direct consequences of the standard inequality $\|s_1 + s_2\|_*^2 \leq (1 + \tau)\|s_1\|_*^2 + (1 + \tau^{-1})\|s_2\|_*^2$ which is valid for any $s_1, s_2 \in \mathbb{R}^d$ and any $\tau > 0$. Indeed, let $g$ and $\xi$ be, respectively, the function and the random variable components of $\widehat{g}$, and let $\Delta(x, \xi) := g(x, \xi) - \mathbb{E}[g(x, \xi)]$ for any $x \in \mathbb{R}^d$. Then, for any $x, y, z \in \mathbb{R}^d$ and $\tau > 0$, we can estimate

$$\begin{aligned}
\operatorname{Var}_{\widehat{g}}(x) &\equiv \mathbb{E}[\|\Delta(x, \xi)\|_*^2] = \mathbb{E}[\|\Delta(y, \xi) + [\Delta(x, \xi) - \Delta(y, \xi)]\|_*^2] \\
&\leq (1 + \tau) \mathbb{E}[\|\Delta(y, \xi)\|_*^2] + (1 + \tau^{-1}) \mathbb{E}[\|\Delta(x, \xi) - \Delta(y, \xi)\|_*^2] \\
&\equiv (1 + \tau) \operatorname{Var}_{\widehat{g}}(y) + (1 + \tau^{-1}) \operatorname{Var}_{\widehat{g}}(x, y).
\end{aligned}$$

Similarly,

$$\begin{aligned}
\operatorname{Var}_{\widehat{g}}(x, y) &\equiv \mathbb{E}\big[\|\Delta(x, \xi) - \Delta(y, \xi)\|_*^2\big] = \mathbb{E}\big[\|[\Delta(x, \xi) - \Delta(z, \xi)] - [\Delta(y, \xi) - \Delta(z, \xi)]\|_*^2\big] \\
&\leq (1 + \tau) \mathbb{E}\big[\|\Delta(x, \xi) - \Delta(z, \xi)\|_*^2\big] + (1 + \tau^{-1}) \mathbb{E}\big[\|\Delta(y, \xi) - \Delta(z, \xi)\|_*^2\big] \\
&\equiv (1 + \tau) \operatorname{Var}_{\widehat{g}}(x, z) + (1 + \tau^{-1}) \operatorname{Var}_{\widehat{g}}(y, z). \qquad\square
\end{aligned}$$

## B  Omitted Proofs for Section 3

**Lemma 20** (AdaGrad stepsize). *Let function $f$ satisfy Assumption 1. Consider the stepsize update rule $\widehat{M}_+ = M_+(M, \Omega, x, \widehat{x}_+, \widehat{g}_x, \widehat{g}_{x_+})$ defined by*

$$\widehat{M}_+ := \sqrt{M^2 + \frac{1}{\Omega}\|\widehat{g}_{x_+} - \widehat{g}_x\|_*^2}.$$

*Then, this stepsize update rules satisfies (2) with $c_1 = \frac{5}{2}$, $c_2 = 4$, $c_3 = 6$, $c_4 = 2$.*

*Proof.* Let $\widehat{\Delta}(\overline{M}) := \beta_{f, \bar{f}, \bar{g}}(x, \widehat{x}_+) + \langle \bar{g}(x) - \widehat{g}_x, \widehat{x}_+ - x \rangle - \frac{\overline{M}}{2}\|\widehat{x}_+ - x\|^2$. From our Assumption 1 and Theorem 14 (inequality (8)), it follows that $\beta_{f, \bar{f}, \bar{g}}(x, \widehat{x}_+) + \beta_{f, \bar{f}, \bar{g}}(\widehat{x}_+, x) \leq \langle \bar{g}(\widehat{x}_+) - \bar{g}(x), \widehat{x}_+ - x \rangle + 2\delta_f$. Hence,

$$\mathbb{E}[\widehat{\Delta}(\widehat{M}_+) + \beta_{f, \bar{f}, \bar{g}}(\widehat{x}_+, x)] \leq \mathbb{E}\Big[\langle \bar{g}(\widehat{x}_+) - \widehat{g}_x, \widehat{x}_+ - x \rangle - \frac{\widehat{M}_+}{2}\|\widehat{x}_+ - x\|^2\Big] + 2\delta_f = \mathbb{E}[\widehat{\Delta}_1(\widehat{M}_+)] + 2\delta_f,$$

where $\widehat{\Delta}_1(\widehat{M}_+) := \langle \widehat{g}_{x_+} - \widehat{g}_x, \widehat{x}_+ - x \rangle - \frac{\widehat{M}_+}{2}\|\widehat{x}_+ - x\|^2$. Hence,

$$\Gamma := \mathbb{E}[\widehat{\Delta}(\widehat{M}_+) + (\widehat{M}_+ - M)\Omega + \beta_{f,\bar{f},\bar{g}}(\widehat{x}_+, x)] \le \mathbb{E}[\widehat{\Delta}_1(\widehat{M}_+) + (\widehat{M}_+ - M)\Omega] + 2\delta_f.$$

From the definition of $\widehat{M}_+$, it follows that $\|\widehat{g}_{x_+} - \widehat{g}_x\|_*^2 = (\widehat{M}_+^2 - M^2)\Omega = (\widehat{M}_+ + M)(\widehat{M}_+ - M)\Omega$. Since $\widehat{M}_+ \ge M$, this means that

$$\frac{1}{2\widehat{M}_+}\|\widehat{g}_{x_+} - \widehat{g}_x\|_*^2 \le (\widehat{M}_+ - M)\Omega \le \frac{1}{\widehat{M}_+}\|\widehat{g}_{x_+} - \widehat{g}_x\|_*^2$$

Let us now upper bound $\widehat{\Gamma} := \widehat{\Delta}_1(\widehat{M}_+) + (\widehat{M}_+ - M)\Omega$. For this, let us fix an arbitrary constant $\overline{M} \ge 0$ and consider two cases. If $\widehat{M}_+ \ge \overline{M}$, we can bound

$$\widehat{\Gamma} \le \widehat{\Delta}_1(\widehat{M}_+) + \frac{1}{\widehat{M}_+}\|\widehat{g}_{x_+} - \widehat{g}_x\|_*^2 \le \widehat{\Delta}_1(\overline{M}) + \frac{1}{\overline{M}}\|\widehat{g}_{x_+} - \widehat{g}_x\|_*^2 =: \widehat{\Gamma}(\overline{M}).$$

If $\widehat{M}_+ \le \overline{M}$, we can bound

$$\widehat{\Gamma} \le \frac{1}{2\widehat{M}_+}\|\widehat{g}_{x_+} - \widehat{g}_x\|_*^2 + (\widehat{M}_+ - M)\Omega \le 2(\widehat{M}_+ - M)\Omega = 2[\min\{\widehat{M}_+, \overline{M}\} - M]_+\Omega.$$

Combining the two cases, we get $\widehat{\Gamma} \le [\widehat{\Gamma}(\overline{M})]_+ + 2[\min\{\widehat{M}_+, \overline{M}\} - M]_+\Omega$. Thus,

$$\Gamma \le \mathbb{E}[\widehat{\Gamma}] + 2\delta_f \le \mathbb{E}\{[\widehat{\Gamma}(\overline{M})]_+\} + 2\mathbb{E}\{[\min\{\widehat{M}_+, \overline{M}\} - M]_+\Omega\} + 2\delta_f.$$

Let us now estimate the first term. Denote $\widehat{S} := \widehat{g}_x - \bar{g}(x)$ and $\widehat{S}_+ := \widehat{g}_{x_+} - \bar{g}(\widehat{x}_+)$. Then,

$$\widehat{\Gamma}(\overline{M}) \equiv \langle \widehat{g}_{x_+} - \widehat{g}_x, \widehat{x}_+ - x \rangle - \frac{\overline{M}}{2}\|\widehat{x}_+ - x\|^2 + \frac{1}{\overline{M}}\|\widehat{g}_{x_+} - \widehat{g}_x\|_*^2$$

$$\le \langle \bar{g}(\widehat{x}_+) - \bar{g}(x), \widehat{x}_+ - x \rangle + \frac{2}{\overline{M}}\|\bar{g}(\widehat{x}_+) - \bar{g}(x)\|_*^2$$

$$+ \langle \widehat{S}_+ - \widehat{S}, \widehat{x}_+ - x \rangle + \frac{2}{\overline{M}}\|\widehat{S}_+ - \widehat{S}\|_*^2 - \frac{\overline{M}}{2}\|\widehat{x}_+ - x\|^2$$

Using now our Assumption 1 and Theorem 14 (inequalities (9) and (12)), we can continue as follows:

$$\widehat{\Gamma}(\overline{M}) \le L_f\|\widehat{x}_+ - x\|^2 + 2\delta_f + \frac{2}{\overline{M}}(L_f^2\|\widehat{x}_+ - x\|^2 + 4L_f\delta_f)$$

$$+ \langle \widehat{S}_+ - \widehat{S}, \widehat{x}_+ - x \rangle + \frac{2}{\overline{M}}\|\widehat{S}_+ - \widehat{S}\|_*^2 - \frac{\overline{M}}{2}\|\widehat{x}_+ - x\|^2$$

$$\le \langle \widehat{S}_+ - \widehat{S}, \widehat{x}_+ - x \rangle + \frac{2}{\overline{M}}\|\widehat{S}_+ - \widehat{S}\|_*^2 - \frac{\overline{M} - 2L_f(1 + \frac{2L_f}{\overline{M}})}{2}\|\widehat{x}_+ - x\|^2 + 2\Big(1 + \frac{4L_f}{\overline{M}}\Big)\delta_f$$

$$\le \Big(\frac{2}{\overline{M}} + \frac{1}{2[\overline{M} - 2L_f(1 + \frac{2L_f}{\overline{M}})]}\Big)\|\widehat{S}_+ - \widehat{S}\|_*^2 + 2\Big(1 + \frac{4L_f}{\overline{M}}\Big)\delta_f.$$

Consequently,

$$\mathbb{E}\{[\widehat{\Gamma}(\overline{M})]_+\} \le \Big(\frac{2}{\overline{M}} + \frac{1}{2[\overline{M} - 2L_f(1 + \frac{2L_f}{\overline{M}})]}\Big)\mathbb{E}[\|\widehat{S}_+ - \widehat{S}\|_*^2] + 2\Big(1 + \frac{4L_f}{\overline{M}}\Big)\delta_f.$$

In particular, for $\overline{M} > 4L_f$, we can estimate $\frac{2}{\overline{M}} + \frac{1}{2[\overline{M} - 2L_f(1 + \frac{2L_f}{\overline{M}})]} \le \frac{2}{\overline{M}} + \frac{1}{2(\overline{M} - 4L_f)} \le \frac{5}{2(\overline{M} - 4L_f)}$. Therefore, for any $\overline{M} > 4L_f$,

$$\mathbb{E}\{[\widehat{\Gamma}(\overline{M})]_+\} \le \frac{5}{2(\overline{M} - 4L_f)}\mathbb{E}[\|\widehat{S}_+ - \widehat{S}\|_*^2] + 4\delta_f = \frac{5}{2(\overline{M} - 4L_f)}\mathbb{E}[\mathrm{Var}_{\widehat{g}}(\widehat{x}_+) + \mathrm{Var}_{\widehat{g}}(x)] + 4\delta_f,$$

where the final identity follows from the fact that $\mathbb{E}[\|\widehat{S}_+ - \widehat{S}\|_*^2] = \mathbb{E}[\|\widehat{S}_+\|_*^2] + \mathbb{E}[\|\widehat{S}\|_*^2] = \mathbb{E}[\mathrm{Var}_{\widehat{g}}(\widehat{x}_+)] + \mathrm{Var}_{\widehat{g}}(x)$ (because $\widehat{S}_+$, conditioned on the randomness $\xi$ defining $\widehat{g}_x \equiv g(x, \xi)$, has zero mean).

Combining everything together, we get

$$\Gamma \le \frac{5}{2(\overline{M} - 4L_f)} \mathbb{E}[\mathrm{Var}_{\widehat{g}}(\widehat{x}_+) + \mathrm{Var}_{\widehat{g}}(x)] + 6\delta_f + 2\mathbb{E}\{[\min\{\widehat{M}_+, \overline{M}\} - M]_+\Omega\}.$$

This is exactly (2) with $c_1 = \frac{5}{2}$, $c_2 = 4$, $c_3 = 6$, $c_4 = 2$. $\qquad\square$

**Lemma 21.** *Let function $f$ satisfy Assumption 1. Consider the stepsize update rule $\widehat{M}_+ = M_+(M, \Omega, x, \widehat{x}_+, \widehat{g}_x, \widehat{g}_{x_+})$ defined as the solution of the following equation:*

$$(\widehat{M}_+ - M)\Omega = [\widehat{\Delta}_1(\widehat{M}_+)]_+, \qquad \widehat{\Delta}_1(\widehat{M}_+) := \langle \widehat{g}_{x_+} - \widehat{g}_x, \widehat{x}_+ - x \rangle - \frac{\widehat{M}_+}{2}\|\widehat{x}_+ - x\|^2.$$

*Then, this stepsize update rules satisfies (2) with $c_1 = 1$, $c_2 = 2$, $c_3 = 6$, $c_4 = 2$.*

*Proof.* Let us define $\widehat{\Delta}(\overline{M}) := \beta_{f,\bar{f},\bar{g}}(x, \widehat{x}_+) + \langle \bar{g}(x) - \widehat{g}_x, \widehat{x}_+ - x \rangle - \frac{\overline{M}}{2}\|\widehat{x}_+ - x\|^2$. Starting as in the proof of Lemma 20, we see that

$$\Gamma := \mathbb{E}[\widehat{\Delta}(\widehat{M}_+) + (\widehat{M}_+ - M)\Omega + \beta_{f,\bar{f},\bar{g}}(\widehat{x}_+, x)] \le \mathbb{E}[\widehat{\Delta}_1(\widehat{M}_+) + (\widehat{M}_+ - M)\Omega] + 2\delta_f,$$

with the same $\widehat{\Delta}_1(\cdot)$ as defined in the statement.

Let us now upper bound $\widehat{\Gamma} := \widehat{\Delta}_1(\widehat{M}_+) + (\widehat{M}_+ - M)\Omega$. For this, let us fix an arbitrary constant $\overline{M} \ge 0$ and consider two cases. If $\widehat{M}_+ \ge \overline{M}$, we can bound, using the monotonicity of $\widehat{\Delta}_1(\cdot)$,

$$\widehat{\Gamma} = \widehat{\Delta}_1(\widehat{M}_+) + [\widehat{\Delta}_1(\widehat{M}_+)]_+ \le \widehat{\Delta}_1(\overline{M}) + [\widehat{\Delta}_1(\overline{M})]_+ \le 2[\widehat{\Delta}_1(\overline{M})]_+.$$

If $\widehat{M}_+ \le \overline{M}$, we can bound

$$\widehat{\Gamma} \le [\widehat{\Delta}_1(\widehat{M}_+)]_+ + (\widehat{M}_+ - M)\Omega = 2(\widehat{M}_+ - M)\Omega = 2[\min\{\widehat{M}_+, \overline{M}\} - M]_+\Omega.$$

Combining the two cases, we get $\widehat{\Gamma} \le 2[\widehat{\Delta}_1(\overline{M})]_+ + 2[\min\{\widehat{M}_+, \overline{M}\} - M]_+\Omega$, and hence

$$\Gamma \le \mathbb{E}[\widehat{\Gamma}] + 2\delta_f \le 2\mathbb{E}\{[\widehat{\Delta}_1(\overline{M})]_+\} + 2\mathbb{E}\{[\min\{\widehat{M}_+, \overline{M}\} - M]_+\Omega\} + 2\delta_f.$$

Let us now estimate the first term. According to our Assumption 1 and Theorem 14 (inequality (9)), we have $\langle \bar{g}(\widehat{x}_+) - \bar{g}(x), \widehat{x}_+ - x \rangle \le L_f\|\widehat{x}_+ - x\|^2 + 2\delta_f$. Hence, denoting $\widehat{S} := \widehat{g}_x - \bar{g}(x)$ and $\widehat{S}_+ := \widehat{g}_{x_+} - \bar{g}(\widehat{x}_+)$, we can estimate, for any $\overline{M} > 2L_f$,

$$\widehat{\Delta}_1(\overline{M}) = \langle \bar{g}(\widehat{x}_+) - \bar{g}(x), \widehat{x}_+ - x \rangle + \langle \widehat{S}_+ - \widehat{S}, \widehat{x}_+ - x \rangle - \frac{\overline{M}}{2}\|\widehat{x}_+ - x\|^2$$

$$\le \langle \widehat{S}_+ - \widehat{S}, \widehat{x}_+ - x \rangle - \frac{\overline{M} - 2L_f}{2}\|\widehat{x}_+ - x\|^2 + 2\delta_f \le \frac{1}{2(\overline{M} - 2L_f)}\|\widehat{S}_+ - \widehat{S}\|_*^2 + 2\delta_f.$$

Hence,

$$\mathbb{E}\{[\widehat{\Delta}_1(\overline{M})]_+\} \le \frac{1}{2(\overline{M} - 2L_f)}\mathbb{E}[\|\widehat{S}_+ - \widehat{S}\|_*^2] + 2\delta_f = \frac{1}{2(\overline{M} - 2L_f)}\mathbb{E}[\mathrm{Var}_{\widehat{g}}(\widehat{x}_+) + \mathrm{Var}_{\widehat{g}}(x)] + 2\delta_f,$$

where the final identity follows from the fact that $\mathbb{E}[\|\widehat{S}_+ - \widehat{S}\|_*^2] = \mathbb{E}[\|\widehat{S}_+\|_*^2] + \mathbb{E}[\|\widehat{S}\|_*^2] = \mathbb{E}[\mathrm{Var}_{\widehat{g}}(\widehat{x}_+)] + \mathrm{Var}_{\widehat{g}}(x)$ (because $\widehat{S}_+$, conditioned on the randomness $\xi$ defining $\widehat{g}_x \equiv g(x, \xi)$, has zero mean).

Thus,

$$\Gamma \le \frac{1}{\overline{M} - 2L_f}\mathbb{E}[\mathrm{Var}_{\widehat{g}}(\widehat{x}_+) + \mathrm{Var}_{\widehat{g}}(x)] + 6\delta_f + 2\mathbb{E}\{[\min\{\widehat{M}_+, \overline{M}\} - M]_+\Omega\},$$

which is exactly (2) with $c_1 = 1$, $c_2 = 2$, $c_3 = 6$, $c_4 = 2$. $\qquad\square$

## C   Omitted Proofs for Section 4

### C.1   Universal SGD

**Lemma 22** (Stochastic Gradient Step). *Consider problem* (1) *under Assumption 1. Let $\widehat{g}$ be an unbiased oracle for $\bar{g}$. Let $x \in \operatorname{dom}\psi$ be a point, $M \geq 0$ be a coefficient, $\widehat{g}_x \cong \widehat{g}(x)$, and let*

$$\widehat{x}_+ = \operatorname{Prox}_\psi(x, \widehat{g}_x, M).$$

*Denote $\widehat{\Delta}(M) := \beta_{f,\bar{f},\bar{g}}(x, \widehat{x}_+) + \langle \bar{g}(x) - \widehat{g}_x, \widehat{x}_+ - x \rangle - \frac{M}{2}\|\widehat{x}_+ - x\|^2$. Then,*

$$\mathbb{E}\Big[F(\widehat{x}_+) - F^* + \frac{M}{2}\|\widehat{x}_+ - x^*\|^2\Big] + \beta_{f,\bar{f},\bar{g}}(x, x^*) \leq \frac{M}{2}\|x - x^*\|^2 + \mathbb{E}[\widehat{\Delta}(M)].$$

*If further Assumption 2 is satisfied, and $\widehat{M}_+ \geq M$ is a random coefficient (possibly dependent on $\widehat{g}_x$), then, we also have*

$$\mathbb{E}\Big[F(\widehat{x}_+) - F^* + \frac{\widehat{M}_+}{2}\|\widehat{x}_+ - x^*\|^2\Big] + \beta_{f,\bar{f},\bar{g}}(x, x^*) \leq \frac{M}{2}\|x - x^*\|^2 + \mathbb{E}\big[\widehat{\Delta}(\widehat{M}_+) + (\widehat{M}_+ - M)D^2\big].$$

*Proof.* From Lemma 17, it follows that

$$\bar{f}(x) + \langle \widehat{g}_x, \widehat{x}_+ - x \rangle + \psi(\widehat{x}_+) + \frac{M}{2}\|\widehat{x}_+ - x^*\|^2 + \frac{M}{2}\|\widehat{x}_+ - x\|^2$$
$$\leq \bar{f}(x) + \langle \widehat{g}_x, x^* - x \rangle + \psi(x^*) + \frac{M}{2}\|x - x^*\|^2.$$

Passing to expectations and rewriting

$$\mathbb{E}[\bar{f}(x) + \langle \widehat{g}_x, x^* - x \rangle + \psi(x^*)] = \bar{f}(x) + \langle \bar{g}(x), x^* - x \rangle + \psi(x^*) = F(x^*) - \beta_{f,\bar{f},\bar{g}}(x, x^*),$$

and

$$\bar{f}(x) + \langle \widehat{g}_x, \widehat{x}_+ - x \rangle + \psi(\widehat{x}_+) = F(\widehat{x}_+) - [f(\widehat{x}_+) - \bar{f}(x) - \langle \widehat{g}_x, \widehat{x}_+ - x \rangle]$$
$$= F(\widehat{x}_+) - [\beta_{f,\bar{f},\bar{g}}(x, \widehat{x}_+) + \langle \bar{g}(x) - \widehat{g}_x, \widehat{x}_+ - x \rangle],$$

we obtain the first of the claimed inequalities.

To prove the second one, we simply add to both sides of the already proved first inequality the expected value of

$$\frac{\widehat{M}_+ - M}{2}\|\widehat{x}_+ - x^*\|^2 + \widehat{\Delta}(M) - \widehat{\Delta}(\widehat{M}_+) = \frac{\widehat{M}_+ - M}{2}\big(\|\widehat{x}_+ - x^*\|^2 + \|\widehat{x}_+ - x\|^2\big)$$

and then bound $\|\widehat{x}_+ - x^*\| \leq D$, $\|\widehat{x}_+ - x\| \leq D$ using our Assumption 2 and the fact that $x, \widehat{x}_+, x^* \in \operatorname{dom}\psi$. $\qquad\square$

**Lemma 23** (Universal Stochastic Gradient Step). *Consider problem* (1) *under Assumptions 1 and 2. Let $\widehat{g}$ be an unbiased oracle for $\bar{g}$. Further, let $x \in \operatorname{dom}\psi$ be a point, $M \geq 0$ be a coefficient, $\widehat{g}_x \cong \widehat{g}(x)$, and let*

$$\widehat{x}_+ = \operatorname{Prox}_\psi(x, \widehat{g}_x, M), \quad \widehat{g}_{x_+} \cong \widehat{g}(\widehat{x}_+), \quad \widehat{M}_+ = M_+(M, D^2, x, \widehat{x}_+, \widehat{g}_x, \widehat{g}_{x_+}).$$

*Then, for any $\overline{M} > c_2 L_f$, it holds that*

$$\mathbb{E}\Big[F(\widehat{x}_+) - F^* + \frac{\widehat{M}_+}{2}\|\widehat{x}_+ - x^*\|^2 + \beta_{f,\bar{f},\bar{g}}(\widehat{x}_+, x)\Big] + \beta_{f,\bar{f},\bar{g}}(x, x^*)$$
$$\leq \frac{M}{2}\|x - x^*\|^2 + \frac{c_1}{\overline{M} - c_2 L_f}\,\mathbb{E}[\operatorname{Var}_{\widehat{g}}(\widehat{x}_+) + \operatorname{Var}_{\widehat{g}}(x)] + c_3\delta_f + c_4\,\mathbb{E}\{[\min\{\widehat{M}_+, \overline{M}\} - M]_+ D^2\}.$$

*Proof.* According to Lemma 22,

$$\mathbb{E}\Big[F(\widehat{x}_+) - F^* + \frac{\widehat{M}_+}{2}\|\widehat{x}_+ - x^*\|^2\Big] + \beta_{f,\bar{f},\bar{g}}(x, x^*) \leq \frac{M}{2}\|x - x^*\|^2 + \mathbb{E}\big[\widehat{\Delta}(\widehat{M}_+) + (\widehat{M}_+ - M)D^2\big],$$

where $\widehat{\Delta}(\widehat{M}_+) := \beta_{f,\bar{f},\bar{g}}(x,\widehat{x}_+) + \langle \bar{g}(x) - \widehat{g}_x, \widehat{x}_+ - x \rangle - \frac{\widehat{M}_+}{2}\|\widehat{x}_+ - x\|^2$. At the same time, according to the main requirement (2) on the stepsize update rule, for any $\overline{M} > c_2 L_f$,

$$\mathbb{E}\big[\widehat{\Delta}(\widehat{M}_+) + (\widehat{M}_+ - M)D^2 + \beta_{f,\bar{f},\bar{g}}(\widehat{x}_+, x)\big]$$
$$\leq \frac{c_1}{\overline{M} - c_2 L_f}\,\mathbb{E}[\mathrm{Var}_{\widehat{g}}(\widehat{x}_+) + \mathrm{Var}_{\widehat{g}}(x)] + c_3 \delta_f + c_4\,\mathbb{E}\big\{[\min\{\widehat{M}_+, \overline{M}\} - M]_+ D^2\big\}.$$

Combining the two displays, we get the claim. $\qquad\square$

**Lemma 24** (Universal SGD: General Guarantee). *Consider problem* (1) *under Assumptions 1 and 2. Let $\widehat{g}$ be an unbiased oracle for $\bar{g}$. Further, let $x \in \mathrm{dom}\,\psi$ be a point, $M \geq 0$ be a coefficient, $N \geq 1$ be an integer, and let*
$$(\bar{x}_N, x_N, M_N) \cong \mathrm{UniSgd}_{\widehat{g},\psi}(x_0, M_0, N; D),$$
*as defined by Algorithm 1, and let $x_0, \ldots, x_N$ be the corresponding points generated inside the algorithm. Then, for any $\overline{M} > c_2 L_f$, it holds that*

$$\mathbb{E}\left[N[F(\bar{x}_N) - F^*] + \frac{M_N}{2}\|x_N - x^*\|^2 + \sum_{k=0}^{N-1}[\beta_{f,\bar{f},\bar{g}}(x_{k+1}, x_k) + \beta_{f,\bar{f},\bar{g}}(x_k, x^*)]\right]$$

$$\leq \frac{M_0}{2}\|x_0 - x^*\|^2 + \frac{c_1}{\overline{M} - c_2 L_f}\sum_{k=0}^{N-1}\mathbb{E}[\mathrm{Var}_{\widehat{g}}(x_{k+1}) + \mathrm{Var}_{\widehat{g}}(x_k)] + c_3 N \delta_f$$

$$+ c_4\,\mathbb{E}\big\{[\min\{M_N, \overline{M}\} - M_0]_+ D^2\big\}.$$

*Proof.* Each iteration $k$ of the algorithm, when conditioned on $x_k$, follows the construction from Lemma 23 (with $x = x_k$, $\widehat{g}_x = g_k$, $M = M_k$, $\widehat{x}_+ = x_{k+1}$, $\widehat{g}_{x_+} = g_{k+1}$, $\widehat{M}_+ = M_{k+1}$). Hence, we can write, after passing to full expectations, that, for each $k = 0, \ldots, N - 1$,

$$\mathbb{E}\left[F(x_{k+1}) - F^* + \frac{M_{k+1}}{2}\|x_{k+1} - x^*\|^2 + \beta_{f,\bar{f},\bar{g}}(x_{k+1}, x_k) + \beta_{f,\bar{f},\bar{g}}(x_k, x^*)\right]$$
$$\leq \mathbb{E}\left[\frac{M_k}{2}\|x_k - x^*\|^2 + \frac{c_1}{\overline{M} - c_2 L_f}[\mathrm{Var}_{\widehat{g}}(x_{k+1}) + \mathrm{Var}_{\widehat{g}}(x_k)] + c_4[\min\{M_{k+1}, \overline{M}\} - M_k]_+ D^2\right] + c_3 \delta_f,$$

where $\overline{M} > 2L_f$ is an arbitrary constant. Telescoping the above inequalities (using Lemma 18) and then bounding $N[F(\bar{x}_N) - F^*] \leq \sum_{k=1}^{N}[F(x_k) - F^*]$ (using the convexity of $F$ and our choice of $\bar{x}_N = \frac{1}{N}\sum_{k=1}^{N} x_k$), we get the claim. $\qquad\square$

**Theorem 4.** *Let Algorithm 1 with $M_0 = 0$ be applied to problem* (1) *under Assumptions 1–3. Then, for the point $\bar{x}_N$ generated by the algorithm, we have*

$$\mathbb{E}[F(\bar{x}_N)] - F^* \leq \frac{c_2 c_4 L_f D^2}{N} + 2\sigma D\sqrt{\frac{2c_1 c_4}{N}} + c_3 \delta_f.$$

*Proof.* Applying Lemma 24, substituting our choice of $M_0 = 0$, estimating $\mathrm{Var}_{\widehat{g}}(\cdot) \leq \sigma^2$ and dropping the nonnegative $\beta_{f,\bar{f},\bar{g}}(\cdot, \cdot)$ terms, we obtain

$$\mathbb{E}[F(\bar{x}_N)] - F^* \leq \frac{1}{N}\left(c_4 \overline{M} D^2 + \frac{2c_1 \sigma^2 N}{\overline{M} - c_2 L_f} + c_3 N \delta_f\right) = \frac{c_4 \overline{M} D^2}{N} + \frac{2c_1 \sigma^2}{\overline{M} - c_2 L_f} + c_3 \delta_f,$$

where $\overline{M} > 2L_f$ is an arbitrary constant. The optimal $\overline{M}$ which minimizes the right-hand side is $\overline{M} = c_2 L_f + \frac{\sigma}{D}\sqrt{\frac{2c_1}{c_4}N}$. Substituting this choice into the above display, we get

$$\mathbb{E}[F(\bar{x}_N)] - F^* \leq \frac{c_4 D^2}{N}\left(c_2 L_f + \frac{\sigma}{D}\sqrt{\frac{2c_1}{c_4}N}\right) + \frac{2c_1 \sigma^2}{\frac{\sigma}{D}\sqrt{\frac{2c_1}{c_4}N}} + c_3 \delta_f$$

$$= \frac{c_2 c_4 L_f D^2}{N} + 2\sigma D\sqrt{\frac{2c_1 c_4}{N}} + c_3 \delta_f. \qquad\square$$

## C.2 Universal Fast SGD

**Lemma 25** (Stochastic Triangle Step). *Consider problem* (1) *under Assumption 1. Let $\widehat{g}$ be an unbiased oracle for $\bar{g}$, let $x, v \in \operatorname{dom} \psi$ be points and $M, A \geq 0$, $a > 0$ be coefficients. Further, for $A_+ := A + a$, let*

$$y = \frac{Ax + av}{A_+}, \quad \widehat{g}_y \cong \widehat{g}(y), \quad \widehat{v}_+ = \operatorname{Prox}_\psi(v, \widehat{g}_y, M/a), \quad \widehat{x}_+ = \frac{Ax + a\widehat{v}_+}{A_+}.$$

*Denote $\widehat{\Delta}(M) := \beta_{f,\bar{f},\bar{g}}(y, \widehat{x}_+) + \langle \bar{g}(y) - \widehat{g}_y, \widehat{x}_+ - y \rangle - \frac{MA_+}{2a^2}\|\widehat{x}_+ - y\|^2$. Then,*

$$\mathbb{E}\left[ A_+[F(\widehat{x}_+) - F^*] + \frac{M}{2}\|\widehat{v}_+ - x^*\|^2 \right] + A\beta_{f,\bar{f},\bar{g}}(y, x) + a\beta_{f,\bar{f},\bar{g}}(y, x^*)$$
$$\leq A[F(x) - F^*] + \frac{M}{2}\|v - x^*\|^2 + A_+ \mathbb{E}[\widehat{\Delta}(M)].$$

*If further Assumption 2 is satisfied, and $\widehat{M}_+ \geq M$ is a random coefficient (possibly dependent on $\widehat{g}_y$), then we also have*

$$\mathbb{E}\left[ A_+[F(\widehat{x}_+) - F^*] + \frac{\widehat{M}_+}{2}\|\widehat{v}_+ - x^*\|^2 \right] + A\beta_{f,\bar{f},\bar{g}}(y, x) + a\beta_{f,\bar{f},\bar{g}}(y, x^*)$$
$$\leq A[F(x) - F^*] + \frac{M}{2}\|v - x^*\|^2 + \mathbb{E}[A_+\widehat{\Delta}(\widehat{M}_+) + (\widehat{M}_+ - M)D^2],$$

*Proof.* Denoting $\theta := A\beta_{f,\bar{f},\bar{g}}(y, x) + a\beta_{f,\bar{f},\bar{g}}(y, x^*)$ and using the fact that $\mathbb{E}[\widehat{g}_y] = \bar{g}(y)$, we can rewrite

$$AF(x) + aF(x^*) + \frac{M}{2}\|v - x^*\|^2$$
$$= A[\bar{f}(y) + \langle \bar{g}(y), x - y \rangle + \beta_{f,\bar{f},\bar{g}}(y, x) + \psi(x)]$$
$$\quad + a[\bar{f}(y) + \langle \bar{g}(y), x^* - y \rangle + \beta_{f,\bar{f},\bar{g}}(y, x^*) + \psi(x^*)] + \frac{M}{2}\|v - x^*\|^2$$
$$= A_+\bar{f}(y) + \langle \bar{g}(y), Ax + ax^* - A_+y \rangle + A\psi(x) + a\psi(x^*) + \frac{M}{2}\|v - x^*\|^2 + \theta$$
$$= \mathbb{E}\left[ A_+\bar{f}(y) + \langle \widehat{g}_y, Ax + ax^* - A_+y \rangle + A\psi(x) + a\psi(x^*) + \frac{M}{2}\|v - x^*\|^2 \right] + \theta.$$

Further, by the definition of $\widehat{v}_+$ and Lemma 17,

$$\langle \widehat{g}_y, x^* - \widehat{v}_+ \rangle + \psi(x^*) + \frac{M}{2a}\|v - x^*\|^2 \geq \psi(\widehat{v}_+) + \frac{M}{2a}\|v - \widehat{v}_+\|^2 + \frac{M}{2a}\|\widehat{v}_+ - x^*\|^2.$$

This means that

$$A_+\bar{f}(y) + \langle \widehat{g}_y, Ax + ax^* - A_+y \rangle + A\psi(x) + a\psi(x^*) + \frac{M}{2}\|v - x^*\|^2$$
$$\geq A_+\bar{f}(y) + \langle \widehat{g}_y, Ax + a\widehat{v}_+ - A_+y \rangle + A\psi(x) + a\psi(\widehat{v}_+) + \frac{M}{2}\|v - \widehat{v}_+\|^2 + \frac{M}{2}\|\widehat{v}_+ - x^*\|^2$$
$$\geq A_+[\bar{f}(y) + \langle \widehat{g}_y, \widehat{x}_+ - y \rangle + \psi(\widehat{x}_+)] + \frac{M}{2}\|v - \widehat{v}_+\|^2 + \frac{M}{2}\|\widehat{v}_+ - x^*\|^2$$
$$= A_+F(\widehat{x}_+) + \frac{M}{2}\|\widehat{v}_+ - x^*\|^2 - A_+\widehat{\Delta}(M),$$

where the second inequality is due to the definition of $\widehat{x}_+$ and the convexity of $\psi$, and

$$\widehat{\Delta}(M) := f(\widehat{x}_+) - \bar{f}(y) - \langle \widehat{g}_y, \widehat{x}_+ - y \rangle - \frac{M}{2A_+}\|v - \widehat{v}_+\|^2$$
$$= \beta_{f,\bar{f},\bar{g}}(y, \widehat{x}_+) + \langle \bar{g}(y) - \widehat{g}_y, \widehat{x}_+ - y \rangle - \frac{MA_+}{2a^2}\|\widehat{x}_+ - y\|^2$$

since $\widehat{x}_+ - y = \frac{a}{A_+}(\widehat{v}_+ - v)$ (by the definitions of $y$ and $\widehat{x}_+$). Substituting the above inequality into the first display and rearranging, we get the first of the claimed inequalities.

To prove the second one, we simply add to both sides of the already proved first inequality the expected value of

$$\frac{\widehat{M}_+ - M}{2}\|\widehat{v}_+ - x^*\|^2 + A_+[\widehat{\Delta}(M) - \widehat{\Delta}(\widehat{M}_+)] = \frac{\widehat{M}_+ - M}{2}\left(\|\widehat{v}_+ - x^*\|^2 + \frac{A_+^2}{a^2}\|\widehat{x}_+ - y\|^2\right)$$

and then bound, using the fact that $\widehat{x}_+ - y = \frac{a}{A_+}(\widehat{v}_+ - v)$ together with our Assumption 2,

$$\|\widehat{v}_+ - x^*\|^2 + \frac{A_+^2}{a^2}\|\widehat{x}_+ - y\|^2 = \|\widehat{v}_+ - x^*\|^2 + \|\widehat{v}_+ - v\|^2 \le 2D^2. \qquad \square$$

**Lemma 26** (Universal Stochastic Triangle Step). *Consider problem* (1) *under Assumptions 1 and 2, and let $\widehat{g}$ be an unbiased oracle for $\bar{g}$. Let $x, v \in \operatorname{dom}\psi$ be points, $M, A \ge 0$, $a > 0$ be coefficients. Further, for $A_+ := A + a$, let*

$$y = \frac{Ax + av}{A_+}, \quad \widehat{g}_y \cong \widehat{g}(y), \quad \widehat{v}_+ = \operatorname{Prox}_\psi(v, \widehat{g}_y, M/a), \quad \widehat{x}_+ = \frac{Ax + a\widehat{v}_+}{A_+},$$

$$\widehat{g}_{x_+} \cong \widehat{g}(\widehat{x}_+), \quad \widehat{M}_+ = \frac{a^2}{A_+}M_+\left(\frac{A_+}{a^2}M, \frac{a^2}{A_+^2}D^2, y, \widehat{x}_+, \widehat{g}_y, \widehat{g}_{x_+}\right).$$

*Then, for any $\overline{M} > c_2 L_f \frac{a^2}{A_+}$, it holds that*

$$\mathbb{E}\left[A_+[F(\widehat{x}_+) - F^*] + \frac{\widehat{M}_+}{2}\|\widehat{v}_+ - x^*\|^2 + A_+\beta_{f,\bar{f},\bar{g}}(\widehat{x}_+, y)\right] + A\beta_{f,\bar{f},\bar{g}}(y, x) + a\beta_{f,\bar{f},\bar{g}}(y, x^*)$$

$$\le A[F(x) - F^*] + \frac{M}{2}\|v - x^*\|^2 + \frac{c_1 a^2}{\overline{M} - c_2 L_f \frac{a^2}{A_+}}\mathbb{E}[\operatorname{Var}_{\widehat{g}}(\widehat{x}_+) + \operatorname{Var}_{\widehat{g}}(y)]$$

$$+ c_3 A_+ \delta_f + c_4 \mathbb{E}\{[\min\{\widehat{M}_+, \overline{M}\} - M]_+ D^2\}.$$

*Proof.* According to Lemma 25 (together with the fact that $\widehat{M}_+ \ge M$ which is guaranteed by the requirement on the stepsize update rule), we have

$$\mathbb{E}\left[A_+[F(\widehat{x}_+) - F^*] + \frac{\widehat{M}_+}{2}\|\widehat{v}_+ - x^*\|^2\right] + A\beta_{f,\bar{f},\bar{g}}(y, x) + a\beta_{f,\bar{f},\bar{g}}(y, x^*)$$

$$\le A[F(x) - F^*] + \frac{M}{2}\|v - x^*\|^2 + \mathbb{E}\left[A_+\widehat{\Delta}(\widehat{M}_+) + (\widehat{M}_+ - M)D^2\right],$$

where $\widehat{\Delta}(\widehat{M}_+) := \beta_{f,\bar{f},\bar{g}}(y, \widehat{x}_+) + \langle \bar{g}(y) - \widehat{g}_y, \widehat{x}_+ - y\rangle - \frac{\widehat{M}_+ A_+}{2a^2}\|\widehat{x}_+ - y\|^2$. Further, according to the main requirement (2) on the stepsize update rule (applied in the variables $M' := \frac{A_+}{a^2}M, \Omega := \frac{a^2}{A_+^2}D^2$, $\widehat{M}'_+ := \frac{A_+}{a^2}\widehat{M}_+, \overline{M}' := \frac{A_+}{a^2}\overline{M}$ for which we have $M'\Omega = M\frac{D^2}{A_+}, \widehat{M}'_+\Omega = \widehat{M}_+\frac{D^2}{A_+}, \overline{M}'\Omega = \overline{M}\frac{D^2}{A_+}$), it holds that

$$\mathbb{E}\left[\widehat{\Delta}(\widehat{M}_+) + (\widehat{M}_+ - M)\frac{D^2}{A_+} + \beta_{f,\bar{f},\bar{g}}(\widehat{x}_+, y)\right]$$

$$\le \frac{c_1}{\frac{A_+}{a^2}\overline{M} - c_2 L_f}\mathbb{E}[\operatorname{Var}_{\widehat{g}}(\widehat{x}_+) + \operatorname{Var}_{\widehat{g}}(y)] + c_3\delta_f + c_4\mathbb{E}\left\{[\min\{\widehat{M}_+, \overline{M}\} - M]_+\frac{D^2}{A_+}\right\},$$

where $\overline{M} > c_2 L_f \frac{a^2}{A_+}$ is an arbitrary constant. Multiplying both sides of the above display by $A_+$ and adding the result to the first display, we obtain the claim. $\qquad \square$

**Lemma 27** (Universal Fast SGD: General Guarantee). *Consider Algorithm 2 applied to problem* (1) *under Assumptions 1 and 2. Then, for any $k \ge 1$ and any $\overline{M} > c_2 L_f$, it holds that*

$$\mathbb{E}\left[A_k[F(x_k) - F^*] + \sum_{i=0}^{k-1}[A_{i+1}\beta_{f,\bar{f},\bar{g}}(x_{i+1}, y_i) + a_{i+1}\beta_{f,\bar{f},\bar{g}}(y_i, x^*)]\right]$$

$$\le c_4\overline{M}D^2 + \frac{c_1}{\overline{M} - c_2 L_f}\sum_{i=0}^{k-1}a_{i+1}^2\mathbb{E}[\operatorname{Var}_{\widehat{g}}(x_{i+1}) + \operatorname{Var}_{\widehat{g}}(y_i)] + c_3\delta_f\sum_{i=1}^{k}A_i,$$

*where* $a_k = \frac{1}{2}k$, $A_k = \frac{1}{4}k(k+1)$, $\sum_{i=1}^{k} a_i^2 = \frac{1}{24}k(k+1)(2k+1)$, $\sum_{i=1}^{k} A_i = \frac{1}{12}k(k+1)(k+2)$
*for each* $k \geq 1$.

*Proof.* Each iteration $k$ of the algorithm, when conditioned on $(x_k, v_k)$, follows the construction from Lemma 26 (with $x = x_k$, $v = v_k$, $M = M_k$, $A = A_k$, $a = a_{k+1}$, $A_+ = A_{k+1}$, $y = y_k$, $\hat{g}_y = g_{y_k}$, $\hat{v}_+ = v_{k+1}$, $\hat{x}_+ = x_{k+1}$, $\hat{g}_{x_+} = g_{x_{k+1}}$, $\widehat{M}_+ = M_{k+1}$), where $A_k$ and $a_k$ are the following coefficients: $a_k = \frac{1}{2}k$, $A_k = \sum_{i=1}^{k} a_i = \frac{1}{4}k(k+1)$. Applying Lemma 26 (dropping the nonnegative $\beta_{f,\bar{f},\bar{g}}(y,x)$ term) and passing to full expectations, we therefore obtain, for each $k \geq 0$,

$$\mathbb{E}\Big[A_{k+1}[F(x_{k+1}) - F^*] + \frac{M_{k+1}}{2}\|v_{k+1} - x^*\|^2 + A_{k+1}\beta_{f,\bar{f},\bar{g}}(x_{k+1}, y_k) + a_{k+1}\beta_{f,\bar{f},\bar{g}}(y_k, x^*)\Big]$$

$$\leq \mathbb{E}\Big[A_k[F(x_k) - F^*] + \frac{M_k}{2}\|v_k - x^*\|^2 + \frac{c_1 a_{k+1}^2}{\overline{M} - c_2 L_f \frac{a_{k+1}^2}{A_{k+1}}}[\mathrm{Var}_{\hat{g}}(x_{k+1}) + \mathrm{Var}_{\hat{g}}(y_k)]\Big]$$

$$+ c_3 A_{k+1}\delta_f + c_4 \mathbb{E}\big\{[\min\{M_{k+1}, \overline{M}\} - M_k]_+ D^2\big\},$$

where $\overline{M}$ is an arbitrary constant such that $\overline{M} > c_2 L_f \frac{a_{k+1}^2}{A_{k+1}}$. Note however that, for our sequences $a_k$ and $A_k$, we have $\frac{a_k^2}{A_k} = \frac{\frac{1}{4}k^2}{\frac{1}{4}k(k+1)} = \frac{k}{k+1} \leq 1$. Therefore, we can replace $\frac{c_1 a_{k+1}^2}{\overline{M} - c_2 L_f \frac{a_{k+1}^2}{A_{k+1}}}$ in the above

display with $\frac{c_1 a_{k+1}^2}{\overline{M} - c_2 L_f}$ under the requirement that $\overline{M} > c_2 L_f$. Doing this and then telescoping the above inequalities (applying Lemma 18), and using the fact that $M_0 = A_0 = 0$, we get the claimed inequality.

It remains to do some standard computations to see that $\sum_{i=1}^{k} a_i^2 \equiv \frac{1}{4}\sum_{i=1}^{k} i^2 = \frac{1}{24}k(k+1)(2k+1)$ and $\sum_{i=1}^{k} A_i \equiv \frac{1}{4}\sum_{i=1}^{k} i(i+1) = \frac{1}{4}(\frac{1}{6}k(k+1)(2k+1) + \frac{1}{2}k(k+1)) = \frac{1}{12}k(k+1)(k+2)$. $\square$

**Theorem 5.** *Let Algorithm 2 be applied to problem* (1) *under Assumptions 1–3. Then, for any* $k \geq 1$,

$$\mathbb{E}[F(x_k)] - F^* \leq \frac{4c_2 c_4 L_f D^2}{k(k+1)} + 4\sigma D\sqrt{\frac{2c_1 c_4}{3k}} + \frac{c_3}{3}(k+2)\delta_f.$$

*Proof.* Let $k \geq 1$ be arbitrary and $F_k := \mathbb{E}[F(x_k)] - F^*$. Applying Lemma 27, dropping the nonnegative $\beta_{f,\bar{f},\bar{g}}(\cdot, \cdot)$ terms and bounding $\mathrm{Var}_{\hat{g}}(\cdot) \leq \sigma^2$, we obtain, for an arbitrary constant $\overline{M} > c_2 L_f$,

$$F_k \leq \frac{1}{A_k}\Big(c_4 \overline{M}D^2 + \frac{2c_1 \sigma^2}{\overline{M} - c_2 L_f}\sum_{i=1}^{k} a_i^2 + c_3 \delta_f \sum_{i=1}^{k} A_i\Big)$$

$$= \frac{4}{k(k+1)}\Big(c_4 \overline{M}D^2 + \frac{c_1 k(k+1)(2k+1)\sigma^2}{12(\overline{M} - c_2 L_f)} + \frac{c_3}{12}k(k+1)(k+2)\delta_f\Big)$$

$$= \frac{4c_4 \overline{M}D^2}{k(k+1)} + \frac{c_1(2k+1)\sigma^2}{3(\overline{M} - c_2 L_f)} + \delta_k,$$

where $\delta_k := \frac{c_3}{3}(k+2)\delta_f$. We now choose $\overline{M} > c_2 L_f$ which minimizes the right-hand side. This is $\overline{M} = c_2 L_f + \frac{\sigma}{2D}\sqrt{\frac{c_1}{3c_4}k(k+1)(2k+1)}$, for which we get

$$F_k \leq \frac{4c_4 D^2}{k(k+1)}\Big(c_2 L_f + \frac{\sigma}{2D}\sqrt{\frac{c_1}{3c_4}k(k+1)(2k+1)}\Big) + \frac{c_1(2k+1)\sigma^2}{3\frac{\sigma}{2D}\sqrt{\frac{c_1}{3c_4}k(k+1)(2k+1)}} + \delta_k$$

$$= \frac{4c_2 c_4 L_f D^2}{k(k+1)} + 4\sigma D\sqrt{\frac{c_1 c_4(2k+1)}{3k(k+1)}} + \delta_k \leq \frac{4c_2 c_4 L_f D^2}{k(k+1)} + 4\sigma D\sqrt{\frac{2c_1 c_4}{3k}} + \delta_k. \qquad \square$$

# D Omitted Proofs for Section 5

## D.1 Universal SGD

**Theorem 7.** *Let Algorithm 1 with $M_0 = 0$ be applied to problem* (1) *under Assumptions 1, 2 and 6, and let $\sigma_*^2 := \mathrm{Var}_{\hat{g}}(x^*)$. Then, for the point $\bar{x}_N$ produced by the method, we have*

$$\mathbb{E}[F(\bar{x}_N)] - F^* \leq \frac{c_4(c_2 L_f + 12 c_1 L_{\hat{g}}) D^2}{N} + 2\sigma_* D\sqrt{\frac{6 c_1 c_4}{N}} + c_3 \delta_f + \frac{4}{3}\delta_{\hat{g}}.$$

*Proof.* Let $x_0, \ldots, x_N$ be the points generated inside the method and let $F_N := \mathbb{E}[F(\bar{x}_N)] - F^*$. Using Lemma 19 and Assumption 6, we can estimate, for any $0 \leq k \leq N - 1$,

$$\begin{aligned}
\mathrm{Var}_{\hat{g}}(x_{k+1}) + \mathrm{Var}_{\hat{g}}(x_k) &\leq 3\,\mathrm{Var}_{\hat{g}}(x_k) + 2\,\mathrm{Var}_{\hat{g}}(x_{k+1}, x_k) \\
&\leq 6\sigma_*^2 + 6\,\mathrm{Var}_{\hat{g}}(x_k, x^*) + 2\,\mathrm{Var}_{\hat{g}}(x_{k+1}, x_k) \\
&\leq 6\sigma_*^2 + 12 L_{\hat{g}}[\beta_{f,\bar{f},\bar{g}}(x_k, x^*) + \delta_{\hat{g}}] + 4 L_{\hat{g}}[\beta_{f,\bar{f},\bar{g}}(x_{k+1}, x_k) + \delta_{\hat{g}}] \\
&= 6\sigma_*^2 + 4 L_{\hat{g}}[3\beta_{f,\bar{f},\bar{g}}(x_k, x^*) + \beta_{f,\bar{f},\bar{g}}(x_{k+1}, x_k) + 4\delta_{\hat{g}}].
\end{aligned}$$

Substituting this bound into the general guarantee given by Lemma 24 (and taking into account the fact that $M_0 = 0$), we obtain

$$N F_N + \sum_{k=0}^{N-1} \mathbb{E}[\beta_{f,\bar{f},\bar{g}}(x_{k+1}, x_k) + \beta_{f,\bar{f},\bar{g}}(x_k, x^*)]$$

$$\leq c_4 \overline{M} D^2 + \frac{6 c_1 \sigma_*^2 N}{\overline{M} - c_2 L_f} + \alpha \sum_{k=0}^{N-1} \mathbb{E}[\beta_{f,\bar{f},\bar{g}}(x_{k+1}, x_k) + 3\beta_{f,\bar{f},\bar{g}}(x_k, x^*)] + N(c_3\delta_f + 4\alpha\delta_{\hat{g}}),$$

where $\overline{M} > c_2 L_f$ is an arbitrary constant and $\alpha := \frac{4 c_1 L_{\hat{g}}}{\overline{M} - c_2 L_f}$. Requiring now that $3\alpha \leq 1$ or, equivalently, that $\overline{M} \geq c_2 L_f + 12 c_1 L_{\hat{g}} =: \overline{M}_{\min}$, we can cancel the nonnegative $\beta_{f,\bar{f},\bar{g}}(\cdot, \cdot)$ terms on both sides and obtain

$$F_N \leq \frac{c_4 \overline{M} D^2}{N} + \frac{6 c_1 \sigma_*^2}{\overline{M} - c_2 L_f} + \delta,$$

where $\delta := c_3 \delta_f + \frac{4}{3}\delta_{\hat{g}}$. The optimal coefficient $\overline{M}_*$ minimizing the right-hand side is $\overline{M}_* = c_2 L_f + \frac{\sigma_*}{D}\sqrt{\frac{6 c_1 N}{c_4}}$. However, we still need to respect the constraint $\overline{M} \geq \overline{M}_{\min}$. Choosing $\overline{M} = c_2 L_f + 12 c_1 L_{\hat{g}} + \frac{\sigma_*}{D}\sqrt{\frac{6 c_1 N}{c_4}}$, we conclude that

$$\begin{aligned}
F_N &\leq \frac{c_4 D^2}{N}\left(c_2 L_f + 12 c_1 L_{\hat{g}} + \frac{\sigma_*}{D}\sqrt{\frac{6 c_1 N}{c_4}}\right) + \frac{6 c_1 \sigma_*^2}{\frac{\sigma_*}{D}\sqrt{\frac{6 c_1 N}{c_4}}} + \delta \\
&= \frac{c_4(c_2 L_f + 12 c_1 L_{\hat{g}}) D^2}{N} + 2\sigma_* D\sqrt{\frac{6 c_1 c_4}{N}} + \delta. \qquad \square
\end{aligned}$$

## D.2 Universal Fast SGD

**Theorem 8.** *Let Algorithm 2 be applied to problem* (1) *under Assumptions 1, 2 and 6, and let $\sigma_*^2 := \mathrm{Var}_{\hat{g}}(x^*)$. Then, for any $k \geq 1$, we have*

$$\mathbb{E}[F(x_k)] - F^* \leq \frac{4 c_2 c_4 L_f D^2}{k(k+1)} + \frac{24 c_1 c_4 L_{\hat{g}} D^2}{k+1} + 4\sigma_* D\sqrt{\frac{2 c_1 c_4}{k}} + \frac{c_3}{3}(k+2)\delta_f + \frac{4}{3}\delta_{\hat{g}}.$$

*Proof.* Let $k \geq 1$ be arbitrary and $F_k := \mathbb{E}[F(x_k)] - F^*$. Using Lemma 19 and Assumption 6, we can estimate, for each $i$,

$$\begin{aligned}
\mathrm{Var}_{\hat{g}}(x_{i+1}) + \mathrm{Var}_{\hat{g}}(y_i) &\leq 3\,\mathrm{Var}_{\hat{g}}(y_i) + 2\,\mathrm{Var}_{\hat{g}}(x_{i+1}, y_i) \\
&\leq 6\sigma_*^2 + 6\,\mathrm{Var}_{\hat{g}}(y_i, x^*) + 2\,\mathrm{Var}_{\hat{g}}(x_{i+1}, y_i) \\
&\leq 6\sigma_*^2 + 12 L_{\hat{g}}[\beta_{f,\bar{f},\bar{g}}(y_i, x^*) + \delta_{\hat{g}}] + 4 L_{\hat{g}}[\beta_{f,\bar{f},\bar{g}}(x_{i+1}, y_i) + \delta_{\hat{g}}] \\
&= 6\sigma_*^2 + 4 L_{\hat{g}}[3\beta_{f,\bar{f},\bar{g}}(y_i, x^*) + \beta_{f,\bar{f},\bar{g}}(x_{i+1}, y_i) + 4\delta_{\hat{g}}].
\end{aligned}$$

Substituting this bound into the guarantee given by Lemma 27, we obtain

$$A_k F_k + \sum_{i=0}^{k-1} \mathbb{E}[A_{i+1}\beta_{f,\bar{f},\bar{g}}(x_{i+1}, y_i) + a_{i+1}\beta_{f,\bar{f},\bar{g}}(y_i, x^*)]$$

$$\leq c_4 \overline{M} D^2 + \sum_{i=0}^{k-1} \alpha_{i+1} \mathbb{E}[\beta_{f,\bar{f},\bar{g}}(x_{i+1}, y_i) + 3\beta_{f,\bar{f},\bar{g}}(y_i, x^*) + 4\delta_{\hat{g}}] + \frac{6c_1\sigma_*^2}{\overline{M} - c_2 L_f} \sum_{i=1}^{k} a_i^2 + c_3 \delta_f \sum_{i=1}^{k} A_i,$$

where $\alpha_{i+1} := \frac{4c_1 L_{\hat{g}} a_{i+1}^2}{\overline{M} - c_2 L_f}$, $a_i = \frac{1}{2}i$, $A_k = \frac{1}{4}k(k+1)$, $\sum_{i=1}^{k} a_i^2 = \frac{1}{24}k(k+1)(2k+1)$, $\sum_{i=1}^{k} A_i = \frac{1}{12}k(k+1)(k+2)$. Requiring now that $3\alpha_{i+1} \leq a_{i+1}$ for all $i = 0, \ldots, k-1$ or, equivalently, that $\overline{M} \geq c_2 L_f + 12c_1 L_{\hat{g}} a_k \equiv c_2 L_f + 6c_1 L_{\hat{g}} k$, we can cancel the nonnegative $\beta_{f,\bar{f},\bar{g}}(\cdot, \cdot)$ terms on both sides and obtain

$$F_k \leq \frac{1}{A_k} \left( c_4 \overline{M} D^2 + \frac{6c_1\sigma_*^2}{\overline{M} - c_2 L_f} \sum_{i=1}^{k} a_i^2 + c_3 \delta_f \sum_{i=1}^{k} A_i + \frac{4}{3} A_k \delta_{\hat{g}} \right)$$

$$= \frac{4}{k(k+1)} \left( c_4 \overline{M} D^2 + \frac{c_1\sigma_*^2 k(k+1)(2k+1)}{4(\overline{M} - c_2 L_f)} + \frac{c_3}{12} \delta_f k(k+1)(k+2) \right) + \frac{4}{3} \delta_{\hat{g}}$$

$$= \frac{4c_4 \overline{M} D^2}{k(k+1)} + \frac{c_1\sigma_*^2(2k+1)}{\overline{M} - c_2 L_f} + \delta_k,$$

where $\delta_k := \frac{c_3}{3}(k+2)\delta_f + \frac{4}{3}\delta_{\hat{g}}$.

The minimizer of the right-hand side is $\overline{M}_* = c_2 L_f + \frac{\sigma_*}{2D}\sqrt{\frac{c_1}{c_4}k(k+1)(2k+1)}$. However, recall that we also need to satisfy the constraint $\overline{M} \geq c_2 L_f + 6c_1 L_{\hat{g}} k$. Choosing $\overline{M} = c_2 L_f + 6c_1 L_{\hat{g}} k + \frac{\sigma_*}{2D}\sqrt{\frac{c_1}{c_4}k(k+1)(2k+1)}$, we obtain

$$F_k \leq \frac{4c_4 D^2}{k(k+1)} \left( c_2 L_f + 6c_1 L_{\hat{g}} k + \frac{\sigma_*}{2D}\sqrt{\frac{c_1}{c_4}k(k+1)(2k+1)} \right) + \frac{c_1\sigma_*^2(2k+1)}{\frac{\sigma_*}{2D}\sqrt{\frac{c_1}{c_4}k(k+1)(2k+1)}} + \delta_k$$

$$= \frac{4c_2 c_4 L_f D^2}{k(k+1)} + \frac{24c_1 c_4 L_{\hat{g}} D^2}{k+1} + 4\sigma_* D\sqrt{\frac{c_1 c_4(2k+1)}{k(k+1)}} + \delta_k$$

$$\leq \frac{4c_2 c_4 L_f D^2}{k(k+1)} + \frac{24c_1 c_4 L_{\hat{g}} D^2}{k+1} + 4\sigma_* D\sqrt{\frac{2c_1 c_4}{k}} + \delta_k. \qquad \square$$

# E   Omitted Proofs for Section 6

**Lemma 28** (Basic property of SVRG oracle). *Let $\hat{g}$ be a stochastic oracle in $\mathbb{R}^d$, and let $\hat{G} = \mathrm{SvrgOrac}_{\hat{g}}(\tilde{x})$ for some $\tilde{x} \in \mathbb{R}^d$. Then, for any $x \in \mathbb{R}^d$, the mean value of $\hat{G}$ at $x$ is the same as that of $\hat{g}$ at $x$, while $\mathrm{Var}_{\hat{G}}(x) = \mathrm{Var}_{\hat{g}}(x, \tilde{x})$.*

*Proof.* Let $g$ and $\xi$ be, respectively, the function and the random variable components of $\hat{g}$, and let $g(x) := \mathbb{E}_\xi[g(x, \xi)]$, $g(\tilde{x}) := \mathbb{E}_\xi[g(\tilde{x}, \xi)]$. Then, by definition, $\hat{G}$ is the oracle with the same random variable component $\xi$ and the function component $G$ defined by $G(x, \xi) = g(x, \xi) - g(\tilde{x}, \xi) + g(\tilde{x})$. Consequently, $\mathbb{E}_\xi[G(x, \xi)] = g(x)$, and

$$\mathrm{Var}_{\hat{G}}(x) = \mathbb{E}_\xi\left[\|G(x, \xi) - g(x)\|_*^2\right]$$
$$= \mathbb{E}_\xi\left[\|[g(x, \xi) - g(\tilde{x}, \xi)] - [g(x) - g(\tilde{x})]\|_*^2\right] = \mathrm{Var}_{\hat{g}}(x, \tilde{x}). \qquad \square$$

## E.1   Universal SVRG

**Lemma 29** (Universal SVRG Epoch). *Consider problem* (1) *under Assumptions 1, 2, 6 and 9. Let $x, \tilde{x} \in \mathrm{dom}\,\psi$ be points, $M \geq 0$ be a coefficient, $N \geq 1$ be an integer, $\hat{G} = \mathrm{SvrgOrac}_{\hat{g}}(\tilde{x})$, and let*

$$(\tilde{x}_+, x_+, M_+) \cong \mathrm{UniSgd}_{\hat{G}, \psi}(x, M, N; D),$$

*as defined by Algorithm 1. Then, for any $\overline{M} \geq c_2 L_f + 12 c_1 L_{\widehat{g}}$, $\alpha := \frac{4 c_1 L_{\widehat{g}}}{M - c_2 L_f}$, and any $\nabla f(x^*) \in \partial f(x^*)$, it holds that*

$$
\mathbb{E}\Big[ N[F(\tilde{x}_+) - F^*] + \frac{M_+}{2} \|x_+ - x^*\|^2 \Big]
$$
$$
\leq 6\alpha N \beta_f^{\nabla f(x^*)}(x^*, \tilde{x}) + \frac{M}{2} \|x - x^*\|^2 + N(c_3 \delta_f + 16\alpha \delta_{\widehat{g}}) + c_4 D^2 \, \mathbb{E}\big\{ [\min\{M_+, \overline{M}\} - M]_+ \big\}.
$$

*Proof.* Since $\widehat{g}$ is an unbiased oracle for $\overline{g}$, so is $\widehat{G}$ (Lemma 28). Therefore, we can apply Lemma 24 to get

$$
\mathbb{E}\Big[ N[F(\tilde{x}_+) - F^*] + \frac{M_+}{2}\|x_+ - x^*\|^2 + \sum_{k=0}^{N-1} [\beta_{f,\bar{f},\bar{g}}(x_{k+1}, x_k) + \beta_{f,\bar{f},\bar{g}}(x_k, x^*)] \Big]
$$
$$
\leq \frac{M}{2}\|x - x^*\|^2 + \frac{c_1}{\overline{M} - c_2 L_f} \sum_{k=0}^{N-1} \mathbb{E}[\mathrm{Var}_{\widehat{G}}(x_{k+1}) + \mathrm{Var}_{\widehat{G}}(x_k)] + c_3 N \delta_f
$$
$$
+ c_4 \, \mathbb{E}\big\{ [\min\{M_+, \overline{M}\} - M]_+ D^2 \big\},
$$

where $\overline{M} > c_2 L_f$ is an arbitrary constant and $x_k$ are the points generated inside UniSgd.

Applying now Lemmas 19 and 28 and Assumptions 6 and 9, we can estimate, for each $k$,

$$
\mathrm{Var}_{\widehat{G}}(x_{k+1}) + \mathrm{Var}_{\widehat{G}}(x_k) = \mathrm{Var}_{\widehat{g}}(x_{k+1}, \tilde{x}) + \mathrm{Var}_{\widehat{g}}(x_k, \tilde{x}) \leq 2 \, \mathrm{Var}_{\widehat{g}}(x_{k+1}, x_k) + 3 \, \mathrm{Var}_{\widehat{g}}(x_k, \tilde{x})
$$
$$
\leq 2 \, \mathrm{Var}_{\widehat{g}}(x_{k+1}, x_k) + 6 \, \mathrm{Var}_{\widehat{g}}(x_k, x^*) + 6 \, \mathrm{Var}_{\widehat{g}}(x^*, \tilde{x})
$$
$$
\leq 4 L_{\widehat{g}}[\beta_{f,\bar{f},\bar{g}}(x_{k+1}, x_k) + \delta_{\widehat{g}}] + 12 L_{\widehat{g}}[\beta_{f,\bar{f},\bar{g}}(x_k, x^*) + \delta_{\widehat{g}}] + 24 L_{\widehat{g}}[\beta_f^{\nabla f(x^*)}(x^*, \tilde{x}) + 2\delta_{\widehat{g}}]
$$
$$
= 4 L_{\widehat{g}}[\beta_{f,\bar{f},\bar{g}}(x_{k+1}, x_k) + 3\beta_{f,\bar{f},\bar{g}}(x_k, x^*) + 6\beta_f^{\nabla f(x^*)}(x^*, \tilde{x}) + 16\delta_{\widehat{g}}],
$$

where $\nabla f(x^*) \in \partial f(x^*)$ is arbitrary. Denoting $\alpha := \frac{4 c_1 L_{\widehat{g}}}{M - c_2 L_f}$, we thus obtain

$$
\mathbb{E}\Big[ N[F(\tilde{x}_+) - F^*] + \frac{M_+}{2}\|x_+ - x^*\|^2 + \sum_{k=0}^{N-1} [\beta_{f,\bar{f},\bar{g}}(x_{k+1}, x_k) + \beta_{f,\bar{f},\bar{g}}(x_k, x^*)] \Big]
$$
$$
\leq 6\alpha N \beta_f^{\nabla f(x^*)}(x^*, \tilde{x}) + \frac{M}{2}\|x - x^*\|^2 + N(c_3 \delta_f + 16\alpha \delta_{\widehat{g}}) + c_4 \, \mathbb{E}\big\{ [\min\{M_+, \overline{M}\} - M]_+ D^2 \big\}
$$
$$
+ \alpha \sum_{k=0}^{N-1} \mathbb{E}[\beta_{f,\bar{f},\bar{g}}(x_{k+1}, x_k) + 3\beta_{f,\bar{f},\bar{g}}(x_k, x^*)].
$$

Requiring now $\overline{M} \geq c_2 L_f + 12 c_1 L_{\widehat{g}}$, we get $\alpha \leq \frac{1}{3}$ which allows us to cancel the nonnegative $\beta_{f,\bar{f},\bar{g}}(\cdot, \cdot)$ terms on both sides. The claim now follows. $\square$

**Theorem 10.** *Let* UniSvrg *(as defined by Algorithm 3) be applied to problem* (1) *under Assumptions 1, 2, 6 and 9. Then, for any $t \geq 1$ and $\bar{c}_3 := \max\{c_3, 1\}$, we have*

$$
\mathbb{E}[F(\tilde{x}_t)] - F^* \leq \frac{[(c_2 c_4 + 1) L_f + 48 c_1 c_4 L_{\widehat{g}}] D^2}{2^t} + 2\bar{c}_3 \delta_f + \frac{8}{3} \delta_{\widehat{g}}.
$$

*To construct $\tilde{x}_t$, the algorithm needs to make $O(2^t)$ queries to $\widehat{g}$ and $O(t)$ queries to $\overline{g}$.*

*Proof.* The algorithm iterates $(\tilde{x}_{t+1}, x_{t+1}, M_{t+1}) \cong \mathrm{UniSgd}_{\widehat{G}_t, \psi}(x_t, M_t, 2^{t+1}; D)$ for $t \geq 0$, where $\widehat{G}_t = \mathrm{SvrgOrac}_{\widehat{g}}(\tilde{x}_t)$. Applying Lemma 29 with $\overline{M} := c_2 L_f + 48 c_1 L_{\widehat{g}}$ (for which $\alpha = \frac{1}{12}$ so that $6\alpha 2^{t+1} = 2^t$) and passing to full expectations, we obtain, for any $t \geq 0$,

$$
\mathbb{E}\Big[ 2^{t+1}[F(\tilde{x}_{t+1}) - F^*] + \frac{M_{t+1}}{2}\|x_{t+1} - x^*\|^2 \Big]
$$
$$
\leq \mathbb{E}\Big[ 2^t \beta_t + \frac{M_t}{2}\|x_t - x^*\|^2 + c_4[\min\{M_{t+1}, \overline{M}\} - M_t]_+ D^2 \Big] + 2^{t+1}\Big( c_3 \delta_f + \frac{4}{3}\delta_{\widehat{g}} \Big),
$$

where $\beta_t := \beta_f^{\nabla f(x^*)}(x^*, \tilde{x}_t)$ and $\nabla f(x^*) \in \partial f(x^*)$ can be chosen arbitrarily. Rewriting $F_{t+1} :=$ $F(\tilde{x}_{t+1}) - F^*$ as $F_{t+1} = \beta_{t+1} + (F_{t+1} - \beta_{t+1})$ and telescoping the above inequalities (using, Lemma 18), we get, for any $t \geq 1$,

$$\mathbb{E}\Big[2^t \beta_t + \sum_{i=1}^{t} 2^i (F_i - \beta_i) + \frac{M_t}{2} \|x_t - x^*\|^2\Big]$$

$$\leq \beta_0 + \frac{M_0}{2}\|x_0 - x^*\|^2 + \Big(c_3 \delta_f + \frac{4}{3}\delta_{\hat{g}}\Big)\sum_{i=1}^{t} 2^i + c_4 \mathbb{E}\{[\min\{M_t, \overline{M}\} - M_0]_+ D^2\}$$

$$\leq \beta_0 + 2(2^t - 1)\Big(c_3\delta_f + \frac{4}{3}\delta_{\hat{g}}\Big) + c_4 \overline{M} D^2 =: \Phi_0,$$

where the final inequality is due to the fact that $M_0 = 0$, while $\sum_{i=1}^{t} 2^i = 2(2^t - 1)$. According to Lemma 30, we can choose $\nabla f(x^*) \in \partial f(x^*)$ such that $\beta_i \leq F(\tilde{x}_i) - F^*$ for all $i \geq 0$. Dropping now various nonnegative terms from the left-hand side of the above display, we conclude that

$$2^t \mathbb{E}[F_t] \leq \Phi_0.$$

Let us estimate $\Phi_0$. Using our Assumptions 1 and 2 and Theorem 14 (inequality (14)), we can bound $\beta_0 \leq L_f \|\tilde{x}_0 - x^*\|^2 + 2\delta_f \leq L_f D^2 + 2\delta_f$. Therefore,

$$\Phi_0 \leq L_f D^2 + 2\delta_f + c_4 \overline{M} D^2 + 2(2^t - 1)(c_3 \delta_f + \tfrac{4}{3}\delta_{\hat{g}}) \leq LD^2 + 2(\overline{c}_3 \delta_f + \tfrac{4}{3}\delta_{\hat{g}}) \cdot 2^t$$

where $L := L_f + c_4 \overline{M} \equiv (c_2 c_4 + 1)L_f + 48 c_1 c_4 L_{\hat{g}}$ and $\overline{c}_3 := \max\{c_3, 1\}$. Thus,

$$\mathbb{E}[F_t] \leq \frac{\Phi_0}{2^t} \leq \frac{LD^2}{2^t} + 2\overline{c}_3 \delta_f + \frac{8}{3}\delta_{\hat{g}},$$

which proves the claimed convergence rate.

Let us now estimate the number of oracle queries. At each iteration $t$, the algorithm first queries $\overline{g}$ to construct the SVRG oracle $\widehat{G}_t$ (by precomputing $\overline{g}(\tilde{x}_t)$). All other queries are then done only to $\widehat{G}_t$ or, equivalently, to $\hat{g}$ inside $\text{UniSgd}_{\widehat{G}_t, \psi}$ which is run for $N_{t+1} = 2^{t+1}$ iterations and thus requiring $O(N_{t+1})$ queries to $\hat{g}$. Summing up, after $T$ iterations, we obtain the total number of $\sum_{t=1}^{T} O(N_t) = \sum_{t=1}^{T} O(2^t) = O(2^T)$ queries to $\hat{g}$, and $T$ queries to $\overline{g}$. $\qquad\square$

**Helper Lemmas**

**Lemma 30.** *Let $F: \mathbb{R}^d \to \mathbb{R} \cup \{+\infty\}$ be the function $F(x) := f(x) + \psi(x)$, where $f: \mathbb{R}^d \to \mathbb{R}$ is a convex function, and $\psi: \mathbb{R}^d \to \mathbb{R} \cup \{+\infty\}$ is a proper closed convex function. Let $x^*$ be a minimizer of $F$ and let $F^* := F(x^*)$. Then, there exists $\nabla f(x^*) \in \partial f(x^*)$ such that, for any $x \in \text{dom}\,\psi$,*

$$F(x) - F^* \geq \beta_f^{\nabla f(x^*)}(x^*, x).$$

*Proof.* Since $x^*$ is a minimizer of $F$, we have $0 \in \partial F(x^*) = \partial f(x^*) + \partial \psi(x^*)$. In other words, there exists $\nabla f(x^*) \in \partial f(x^*)$ such that $\nabla \psi(x^*) := -\nabla f(x^*) \in \partial \psi(x^*)$. Consequently, for any $x \in \text{dom}\,\psi$,

$$F(x) - F^* = f(x) - f(x^*) + [\psi(x) - \psi(x^*)] \geq f(x) - f(x^*) + \langle \nabla \psi(x^*), x - x^* \rangle$$

$$= f(x) - f(x^*) - \langle \nabla f(x^*), x - x^* \rangle = \beta_f^{\nabla f(x^*)}(x^*, x). \qquad\square$$

### E.2 Universal Fast SVRG

**Lemma 31** (Universal Triangle SVRG Step). *Consider problem (1) under Assumptions 1, 2 and 6. Let $\tilde{x}, v \in \text{dom}\,\psi$ be points, $M \geq 0$ and $A, a > 0$ be coefficients, $\widehat{G} := \text{SvrgOrac}_{\hat{g}}(\tilde{x})$. Further, let, for $A_+ := A + a$,*

$$x := \frac{A\tilde{x} + av}{A_+}, \quad \widehat{G}_x \cong \widehat{G}(x), \quad \hat{v}_+ = \text{Prox}_\psi(v, \widehat{G}_x, M/a), \quad \hat{x}_+ = \frac{A\tilde{x} + a\hat{v}_+}{A_+},$$

$$\widehat{G}_{x_+} \cong \widehat{G}(\hat{x}_+), \quad \widehat{M}_+ = \frac{a^2}{A_+}M_+\Big(\frac{A_+}{a^2}M, \frac{a^2}{A_+^2}D^2, x, \hat{x}_+, \widehat{G}_x, \widehat{G}_{x_+}\Big).$$

*Then, for* $\overline{M} := c_2 L_f \frac{a^2}{A_+} + 6c_1 L_{\widehat{g}} \frac{a^2}{A}$, *it holds that*

$$\mathbb{E}\Big[A_+[F(\widehat{x}_+) - F^*] + \frac{\widehat{M}_+}{2}\|\widehat{v}_+ - x^*\|^2\Big]$$

$$\leq A[F(\widetilde{x}) - F^*] + \frac{M}{2}\|v - x^*\|^2 + c_4 D^2 \, \mathbb{E}\big\{[\min\{\widehat{M}_+, \overline{M}\} - M]_+\big\} + c_3 A_+ \delta_f + \frac{5}{3} A \delta_{\widehat{g}}.$$

*Proof.* Since $\widehat{g}$ is an unbiased oracle for $\overline{g}$, so is $\widehat{G}$ (Lemma 28). Therefore, we can apply Lemma 26 to obtain

$$\mathbb{E}\Big[A_+[F(\widehat{x}_+) - F^*] + \frac{\widehat{M}_+}{2}\|\widehat{v}_+ - x^*\|^2 + A_+ \beta_{f,\bar{f},\bar{g}}(\widehat{x}_+, x)\Big] + A\beta_{f,\bar{f},\bar{g}}(x, \widetilde{x})$$

$$\leq A[F(\widetilde{x}) - F^*] + \frac{M}{2}\|v - x^*\|^2 + \frac{c_1 a^2}{\overline{M} - c_2 L_f \frac{a^2}{A_+}} \, \mathbb{E}[\text{Var}_{\widehat{G}}(\widehat{x}_+) + \text{Var}_{\widehat{G}}(x)]$$

$$+ c_3 A_+ \delta_f + c_4 \, \mathbb{E}\big\{[\min\{\widehat{M}_+, \overline{M}\} - M]_+ D^2\big\},$$

*where* $\overline{M} > c_2 L_f \frac{a^2}{A_+}$ *is an arbitrary coefficient. Using Lemmas 19 and 28 and Assumption 6, we can further bound*

$$\text{Var}_{\widehat{G}}(\widehat{x}_+) + \text{Var}_{\widehat{G}}(x) = \text{Var}_{\widehat{g}}(\widehat{x}_+, \widetilde{x}) + \text{Var}_{\widehat{g}}(x, \widetilde{x}) \leq 2\,\text{Var}_{\widehat{g}}(\widehat{x}_+, x) + 3\,\text{Var}_{\widehat{g}}(x, \widetilde{x})$$

$$\leq 2L_{\widehat{g}}[2\beta_{f,\bar{f},\bar{g}}(\widehat{x}_+, x) + 3\beta_{f,\bar{f},\bar{g}}(x, \widetilde{x}) + 5\delta_{\widehat{g}}].$$

Denoting $\alpha := \frac{2c_1 L_{\widehat{g}} a^2}{\overline{M} - c_2 L_f \frac{a^2}{A_+}}$, we thus obtain

$$\mathbb{E}\Big[A_+[F(\widehat{x}_+) - F^*] + \frac{\widehat{M}_+}{2}\|\widehat{v}_+ - x^*\|^2 + A_+ \beta_{f,\bar{f},\bar{g}}(\widehat{x}_+, x)\Big] + A\beta_{f,\bar{f},\bar{g}}(x, \widetilde{x})$$

$$\leq A[F(\widetilde{x}) - F^*] + \frac{M}{2}\|v - x^*\|^2 + c_3 A_+ \delta_f + 5\alpha\delta_{\widehat{g}} + c_4 \, \mathbb{E}\big\{[\min\{\widehat{M}_+, \overline{M}\} - M]_+ D^2\big\}$$

$$+ 2\alpha \, \mathbb{E}[\beta_{f,\bar{f},\bar{g}}(\widehat{x}_+, x)] + 3\alpha\beta_{f,\bar{f},\bar{g}}(x, \widetilde{x}).$$

Choosing now $\overline{M} = c_2 L_f \frac{a^2}{A_+} + 6c_1 L_{\widehat{g}} \frac{a^2}{A}$, we get $\alpha = \frac{1}{3}A \ (\leq \frac{1}{3}A_+)$, which allows us to drop the nonnegative $\beta_{f,\bar{f},\bar{g}}(\cdot, \cdot)$ terms from both sides. The claim now follows. $\square$

**Lemma 32** (Universal Triangle SVRG Epoch)**.** *Consider problem* (1) *under Assumptions 1, 2 and 6. Let* $\widetilde{x}, v \in \text{dom}\,\psi$ *be points,* $M \geq 0$ *and* $A, a > 0$ *be coefficients,* $N \geq 1$ *be an integer, and let*

$$(\widetilde{x}_+, v_+, M_+) \cong \text{UniTriSvrgEpoch}_{\widehat{g}, \psi}(\widetilde{x}, v, M, A, a, N; D),$$

*as defined by Algorithm 5. Then, for* $A_+ := A + a$ *and* $\overline{M} := c_2 L_f \frac{a^2}{A_+} + 6c_1 L_{\widehat{g}} \frac{a^2}{A}$, *it holds that*

$$\mathbb{E}\Big[A_+ N[F(\widetilde{x}_+) - F^*] + \frac{M_+}{2}\|v_+ - x^*\|^2\Big]$$

$$\leq AN[F(\widetilde{x}) - F^*] + \frac{M}{2}\|v - x^*\|^2 + c_4 D^2 \, \mathbb{E}\big\{[\min\{M_+, \overline{M}\} - M]_+\big\} + N\Big(c_3 A_+ \delta_f + \frac{5}{3} A \delta_{\widehat{g}}\Big).$$

*Proof.* Each iteration $k$ of the algorithm, when conditioned on $v_k$, follows the construction from Lemma 31 (with $v = v_k$, $M = M_k$, $A = A_k$, $a = a_{k+1}$, $A_+ = A_{k+1}$, $x = x_k$, $\widehat{G}_x = G_{x_k}$, $\widehat{v}_+ = v_{k+1}$, $\widehat{x}_+ = x_{k+1}$, $\widehat{G}_{x_+} = G_{x_{k+1}}$, $\widehat{M}_+ = M_{k+1}$). Hence, we can write, after passing to full expectations, for each $k = 0, \ldots, N - 1$,

$$\mathbb{E}\Big[A_+[F(x_{k+1}) - F^*] + \frac{M_{k+1}}{2}\|v_{k+1} - x^*\|^2\Big]$$

$$\leq A[F(\widetilde{x}) - F^*] + \mathbb{E}\Big[\frac{M_k}{2}\|v_k - x^*\|^2 + c_4[\min\{M_{k+1}, \overline{M}\} - M_k]_+ D^2\Big] + \delta,$$

where $\delta := c_3 A_+ \delta_f + \frac{5}{3} A \delta_{\widehat{g}}$. Telescoping the above inequalities (using Lemma 18), we get

$$\mathbb{E}\Big[A_+ \sum_{k=1}^{N}[F(x_k) - F^*] + \frac{M_N}{2}\|v_N - x^*\|^2\Big]$$

$$\leq AN[F(\tilde{x}) - F^*] + \frac{M_0}{2}\|v_0 - x^*\|^2 + c_4 D^2 \, \mathbb{E}\{[\min\{M_N, \overline{M}\} - M_0]_+\} + N\delta.$$

The claim now follows from the convexity of $F$ and our definitions $\tilde{x}_+ = \bar{x}_N = \frac{1}{N}\sum_{k=1}^{N} x_k$, $v_+ = v_N$, $M_+ = M_N$, $M_0 = M$, $v_0 = v$. $\qquad\square$

**Theorem 11.** *Let* UniFastSvrg *(Algorithm 4) be applied to problem* (1) *under Assumptions 1, 2 and 6, and let $N \geq 9$. Then, for any $t \geq t_0 := \lceil \log_2 \log_3 N \rceil - 1 \ (\geq 0)$, it holds that*

$$\mathbb{E}[F(\tilde{x}_t)] - F^* \leq \frac{9[(c_2 c_4 + \frac{1}{2})L_f + 6c_1 c_4 L_{\widehat{g}}]D^2}{N(t - t_0 + 1)^2} + (c_3 t + 1)\delta_f + \frac{5}{3}t\delta_{\widehat{g}}.$$

*To construct $\tilde{x}_t$, the algorithm needs to make $O(Nt)$ queries to $\widehat{g}$ and $O(t)$ queries to $\bar{g}$. Assuming that the complexity of querying $\bar{g}$ is $n$ times bigger than that of querying $\widehat{g}$ and choosing $N = \Theta(n)$, we get the total stochastic-oracle complexity of $O(nt)$.*

*Proof.* By our definition, the algorithm iterates for $t \geq 0$:

$$(\tilde{x}_{t+1}, v_{t+1}, M_{t+1}) \cong \text{UniTriSvrgEpoch}_{\widehat{g}, \bar{g}, \psi}(\tilde{x}_t, v_t, M_t, A_t, a_{t+1}, N; D),$$

where $A_t$ and $a_{t+1}$ are deterministic coefficients satisfying the following equations:

$$A_{t+1} = A_t + a_{t+1}, \qquad a_{t+1} = \sqrt{A_t}. \tag{17}$$

In particular, for any $t \geq 0$, we have $\overline{M}_t' := c_2 L_f \frac{a_{t+1}^2}{A_{t+1}} + 6c_1 L_{\widehat{g}} \frac{a_{t+1}^2}{A_t} \leq c_2 L_f + 6c_1 L_{\widehat{g}} =: \overline{M}$, and hence $[\min\{M_{t+1}, \overline{M}_t'\} - M_t]_+ \leq [\min\{M_{t+1}, \overline{M}\} - M_t]_+$ (because, for any fixed $a$ and $b$, the function $[\min\{a, \cdot\} - b]_+$ is nondecreasing as the composition of two nondecreasing functions). Applying now Lemma 32 and passing to full expectations, we therefore obtain, for any $t \geq 0$,

$$\mathbb{E}\Big[A_{t+1}N[F(\tilde{x}_{t+1}) - F^*] + \frac{M_{t+1}}{2}\|v_{t+1} - x^*\|^2\Big]$$
$$\leq \mathbb{E}\Big[A_t N[F(\tilde{x}_t) - F^*] + \frac{M_t}{2}\|v_t - x^*\|^2 + c_4[\min\{M_{t+1}, \overline{M}\} - M_t]_+ D^2\Big] + N\Big(c_3 A_{t+1}\delta_f + \frac{5}{3}A_t \delta_{\widehat{g}}\Big).$$

Telescoping the above inequalities (using, in particular, Lemma 18), we obtain, for any $t \geq 1$,

$$A_t N \, \mathbb{E}[F(\tilde{x}_t) - F^*] \leq A_0 N[F(\tilde{x}_0) - F^*] + \frac{M_0}{2}\|v_0 - x^*\|^2$$

$$+ c_4 \, \mathbb{E}\{[\min\{M_t, \overline{M}\} - M_0]_+ D^2\} + N\Big(c_3 \delta_f \sum_{i=1}^{t} A_i + \frac{5}{3}\delta_{\widehat{g}} \sum_{i=0}^{t-1} A_i\Big)$$

$$\leq A_0 N[F(\tilde{x}_0) - F^*] + c_4 \overline{M} D^2 + N S_t(c_3 \delta_f + \tfrac{5}{3}\delta_{\widehat{g}}),$$

where, for the last inequality, we have used the fact that $M_0 = 0$ and denoted $S_t := \sum_{i=1}^{t} A_i$. Thus, for any $t \geq 1$,

$$\mathbb{E}[F(\tilde{x}_t)] - F^* \leq \frac{1}{A_t}\Big(A_0[F(\tilde{x}_0) - F^*] + \frac{c_4 \overline{M} D^2}{N}\Big) + \frac{S_t}{A_t}\Big(c_3 \delta_f + \frac{5}{3}\delta_{\widehat{g}}\Big).$$

At the same time, according to (17), $A_{t+1} - A_t = \sqrt{A_t}$ for any $t \geq 0$. Hence, by Lemma 33 (and our assumption on $A_0$), we can estimate $A_t \geq \frac{1}{9}(t - t_0 + 1)^2$ for any $t \geq t_0 := \lceil \log_2 \log_3 \frac{1}{A_0} \rceil - 1 \ (\geq 0)$. Further, since the sequence $A_t$ is increasing, we can estimate $S_t \equiv \sum_{i=1}^{t} A_i \leq t A_t$, so that $\frac{S_t}{A_t} \leq t$.

Substituting these bounds into the above display and using our formula for $A_0$, we obtain, for any $t \geq t_0$,

$$\mathbb{E}[F(\tilde{x}_t)] - F^* \leq \rho_t[F(\tilde{x}_0) - F^* + c_4 \overline{M} D^2] + t(c_3 \delta_f + \tfrac{5}{3}\delta_{\widehat{g}}),$$

where $\rho_t := \frac{9}{N(t-t_0+1)^2} \leq 1$. By our choice of $\tilde{x}_0$, it holds that $F(\tilde{x}_0) - F^* \leq \frac{1}{2}L_f D^2 + \delta_f$ (see Lemma 34). Denoting $L := \frac{1}{2}L_f + c_4\overline{M} \equiv (c_2 c_4 + \frac{1}{2})L_f + 6c_1 c_4 L_{\hat{g}}$, we get

$$\mathbb{E}[F(\tilde{x}_t)] - F^* \leq \rho_t(LD^2 + \delta_f) + t(c_3 \delta_f + \tfrac{5}{3}\delta_{\hat{g}}) \leq \rho_t LD^2 + (c_3 t + 1)\delta_f + \tfrac{5}{3}t\delta_{\hat{g}},$$

which is exactly the claimed convergence rate bound.

Let us now estimate the number of oracle queries. At the beginning, the algorithm makes 1 query to $\bar{g}$ to compute $\tilde{x}_0$. All other queries to the oracles are then done, at each iteration $t$, only inside the call to UniTriSvrgEpoch (Algorithm 5). Each such a call needs only one query to $\bar{g}$ to construct the SVRG oracle $\widehat{G}$ (by precomputing $\bar{g}(\tilde{x})$), and $O(N)$ queries to $\hat{g}$ (which implements each query to $\widehat{G}$). Summing up, we get, after $t$ iterations, the total number of $O(Nt)$ queries to $\hat{g}$ and $O(t)$ queries to $\bar{g}$. $\qquad\square$

**Helper Lemmas**

**Lemma 33** (c.f. Lemma 1.1 in [23]). *Let $A_t$ be a positive sequence such that*

$$A_{t+1} - A_t \geq \sqrt{\gamma A_t}$$

*for all $t \geq 0$, where $\gamma > 0$, and let $A_0 \leq \frac{1}{9}\gamma$. Then, for any $t \geq 0$, we have*

$$A_t \geq \begin{cases} \gamma(\frac{A_0}{\gamma})^{1/2^t}, & \text{if } t < t_0, \\ \frac{\gamma}{9}(t - t_0 + 1)^2, & \text{if } t \geq t_0, \end{cases}$$

*where $t_0 := \lceil \log_2 \log_3 \frac{\gamma}{A_0} \rceil - 1 \ (\geq 0)$.*

*Proof.* By replacing $A_t$ with $A'_t = A_t/\gamma$, we can assume w.l.o.g. that $\gamma = 1$.

For any $t \geq 0$, we have $A_{t+1} \geq \sqrt{A_t}$, and hence

$$A_t \geq A_0^{1/2^t}.$$

In particular, for $t_0$ (as defined in the statement), we get $t_0 \geq \log_2 \log_3 \frac{1}{A_0} - 1$, so $2^{t_0} \geq \frac{1}{2}\log_3 \frac{1}{A_0}$, and hence

$$A_{t_0} \geq A_0^{2/\log_3(1/A_0)} = \left(3^{-\log_3(1/A_0)}\right)^{2/\log_3(1/A_0)} = 3^{-2} = \frac{1}{9}$$

(recall that $A_0 \leq \frac{1}{9} \leq 1$).

On the other hand, for any $t \geq t_0$, we have

$$\sqrt{A_{t+1}} - \sqrt{A_t} \geq \sqrt{A_t + \sqrt{A_t}} - \sqrt{A_t} = \frac{\sqrt{A_t}}{\sqrt{A_t + \sqrt{A_t}} + \sqrt{A_t}}$$

$$= \frac{1}{\sqrt{1 + \frac{1}{\sqrt{A_t}}} + 1} \geq \frac{1}{\sqrt{1+3} + 1} = \frac{1}{3},$$

where we have used the fact that $A_t \geq A_{t_0} \geq \frac{1}{9}$ since $A_t$ is monotonically increasing. Telescoping these inequalities and rearranging, we get, for any $t \geq t_0$,

$$A_t \geq \left(\frac{1}{3}(t - t_0) + \sqrt{A_{t_0}}\right)^2 \geq \left(\frac{1}{3}(t - t_0) + \frac{1}{3}\right)^2 = \frac{1}{9}(t - t_0 + 1)^2. \qquad\square$$

**Lemma 34.** *Consider problem* (1) *under Assumptions 1 and 2. Let $x \in \operatorname{dom}\psi$, and let $x_+ := \operatorname{Prox}_\psi(x, \bar{g}(x), 0)$. Then, $F(x_+) - F^* \leq \frac{1}{2}L_f D^2 + \delta_f$.*

*Proof.* From the first-order optimality condition for the point $x_+$ (see Lemma 17), it follows that

$$\langle \bar{g}(x), x^* - x_+ \rangle + \psi(x^*) \geq \psi(x_+).$$

Combining the above inequality first with $f(x_+) \le \bar{f}(x) + \langle \bar{g}(x), x_+ - x \rangle + \frac{L_f}{2}\|x_+ - x\|^2 + \delta_f$ and then with $\bar{f}(x) + \langle \bar{g}(x), x^* - x \rangle \le f(x^*)$ (which are both due to our Assumption 1), we obtain

$$F(x_+) = f(x_+) + \psi(x_+) \le f(x_+) + \langle \bar{g}(x), x^* - x_+ \rangle + \psi(x^*)$$

$$\le \bar{f}(x) + \langle \bar{g}(x), x^* - x \rangle + \psi(x^*) + \frac{L_f}{2}\|x_+ - x\|^2 + \delta_f$$

$$\le F^* + \frac{L_f}{2}\|x_+ - x\|^2 + \delta_f.$$

It remains to bound $\|x_+ - x\| \le D$. $\qquad\square$

## F    Omitted Proofs for Section 7

We start with the observation that for our specific example all our main assumptions are satisfied.

*Remark* 35. Under the setting from Example 12, Assumptions 1, 6 and 9 are satisfied with $\bar{f} = f$, $\bar{g}(x) = \nabla f(x) := \mathbb{E}_\xi[\nabla f_\xi(x)]$, any $\delta_f, \delta_{\hat{g}} > 0$ and

$$L_f = \left[\frac{1-\nu}{2(1+\nu)\delta_f}\right]^{\frac{1-\nu}{1+\nu}}[H_f(\nu)]^{\frac{2}{1+\nu}}, \qquad L_{\hat{g}} = \frac{1}{b}\left[\frac{1-\nu}{2(1+\nu)\delta_{\hat{g}}}\right]^{\frac{1-\nu}{1+\nu}}[H_{\max}(\nu)]^{\frac{2}{1+\nu}}.$$

Further, the oracle $\hat{g}_b$ satisfies Assumption 3 with $\sigma_b^2 := \sup_{x \in \mathrm{dom}\,\psi} \mathrm{Var}_{\hat{g}_b}(x) = \frac{1}{b}\sigma^2$, and $\sigma_{*,b}^2 := \mathrm{Var}_{\hat{g}_b}(x^*) = \frac{1}{b}\sigma_*^2$.

*Proof.* For $b = 1$, this follows from Theorem 13 and Lemma 16 and our definitions of $\sigma^2$ and $\sigma_*$. The general case $b \ge 1$ follows from the fact that the standard mini-batching of size $b$ reduces each of the variances $\mathrm{Var}_{\hat{g}_1}(\cdot)$ and $\mathrm{Var}_{\hat{g}_1}(\cdot, \cdot)$ in $b$ times. $\qquad\square$

The following auxiliary result will be useful throughout this section:

**Lemma 36.** *Let $a, b, p > 0$ be real. Then,*

$$\min_{t>0}\left\{\frac{a}{t^p} + bt\right\} = (p+1)a^{\frac{1}{p+1}}\left(\frac{b}{p}\right)^{\frac{p}{p+1}}.$$

*Proof.* The expression inside the $\min$ is a convex function in $t > 0$. Differentiating and setting its derivative to zero, we see that the minimum is attained at $t_* = \left(\frac{ap}{b}\right)^{\frac{1}{p+1}}$. Hence,

$$\min_{t>0}\left\{\frac{a}{t^p} + bt\right\} = a\left(\frac{b}{ap}\right)^{\frac{p}{p+1}} + b\left(\frac{ap}{b}\right)^{\frac{1}{p+1}} = (p+1)a^{\frac{1}{p+1}}\left(\frac{b}{p}\right)^{\frac{p}{p+1}}. \qquad\square$$

### F.1    Uniformly Bounded Variance

**Corollary 37.** *Consider problem* (1) *under the setting from Example 12 and also under Assumption 2. Let Algorithm 1 be applied to this problem with the oracle $\hat{g} = \hat{g}_b$ and initial coefficient $M_0 = 0$. Then, for the point $\bar{x}_N$ generated by the algorithm, we have*

$$\mathbb{E}[F(\bar{x}_N)] - F^* \le \frac{(2c_2c_4)^{\frac{1+\nu}{2}}c_3^{\frac{1-\nu}{2}}}{1+\nu}\frac{H_f(\nu)D^{1+\nu}}{N^{\frac{1+\nu}{2}}} + 2\sigma D\sqrt{\frac{2c_1c_4}{bN}}.$$

*To reach $\mathbb{E}[F(\bar{x}_N)] - F^* \le \epsilon$ for any $\epsilon > 0$, it suffices to make $O\left([\frac{H_f(\nu)}{\epsilon}]^{\frac{2}{1+\nu}}D^2 + \frac{\sigma^2 D^2}{b\epsilon^2}\right)$ queries to $\hat{g}_b$.*

*Proof.* Denote for brevity $H_f := H_f(\nu)$. Taking into account Remark 35 and applying Theorem 4, we get, for any $\delta_f > 0$,

$$F_N := \mathbb{E}[F(\bar{x}_N)] - F^* \le \frac{c_2c_4H_f^{\frac{2}{1+\nu}}D^2}{N}\left[\frac{1-\nu}{2(1+\nu)\delta_f}\right]^{\frac{1-\nu}{1+\nu}} + c_3\delta_f + \sigma_N,$$

where $\sigma_N := 2\sigma_b D\sqrt{\frac{2c_1 c_4}{N}} = 2\sigma D\sqrt{\frac{2c_1 c_4}{bN}}$. Minimizing the right-hand side in $\delta_f$ (using Lemma 36 with $p = \frac{1-\nu}{1+\nu}$ for which $p + 1 = \frac{2}{1+\nu}$), we obtain

$$F_N \leq \frac{2}{1+\nu}\left(\frac{c_2 c_4 H_f^{\frac{2}{1+\nu}} D^2}{N}\left[\frac{1-\nu}{2(1+\nu)}\right]^{\frac{1-\nu}{1+\nu}}\right)^{\frac{1+\nu}{2}}\left(\frac{1+\nu}{1-\nu}c_3\right)^{\frac{1-\nu}{2}} + \sigma_N$$

$$= \frac{(2c_2 c_4)^{\frac{1+\nu}{2}} c_3^{\frac{1-\nu}{2}}}{1+\nu}\frac{H_f D^{1+\nu}}{N^{\frac{1+\nu}{2}}} + \sigma_N.$$

This proves the claimed convergence rate, and the oracle complexity bound easily follows since each iteration of the algorithm requires only 1 query to $\widehat{g}_b$. $\qquad\square$

**Corollary 38.** *Consider problem* (1) *under the setting from Example 12 and also under Assumption 2. Let Algorithm 2 be applied to this problem with the oracle $\widehat{g} = \widehat{g}_b$. Then, for any $k \geq 1$, we have*

$$\mathbb{E}[F(x_k)] - F^* \leq \frac{2^{2+\nu}(c_2 c_4)^{\frac{1+\nu}{2}}(\frac{c_3}{3})^{\frac{1-\nu}{2}}}{1+\nu}\frac{H_f(\nu)D^{1+\nu}}{k^{\frac{1+3\nu}{2}}} + 4\sigma D\sqrt{\frac{2c_1 c_4}{3bk}}.$$

*To reach $\mathbb{E}[F(x_k)] - F^* \leq \epsilon$ for any $\epsilon > 0$, it suffices to make $O\left(\left[\frac{H_f(\nu)D^{1+\nu}}{\epsilon}\right]^{\frac{2}{1+3\nu}} + \frac{\sigma^2 D^2}{b\epsilon^2}\right)$ queries to $\widehat{g}_b$.*

*Proof.* Let $k \geq 1$ be arbitrary and denote for brevity $H_f := H_f(\nu)$. Taking into account Remark 35 and applying Theorem 5, we get, for any $\delta_f > 0$,

$$F_k := \mathbb{E}[F(x_k)] - F^* \leq \frac{4c_2 c_4 H_f^{\frac{2}{1+\nu}} D^2}{k(k+1)}\left[\frac{1-\nu}{2(1+\nu)\delta_f}\right]^{\frac{1-\nu}{1+\nu}} + \frac{c_3}{3}(k+2)\delta_f + \sigma_k,$$

where $\sigma_k := 4\sigma_b D\sqrt{\frac{2c_1 c_4}{3k}} = 4\sigma D\sqrt{\frac{2c_1 c_4}{3bk}}$. Minimizing the right-hand side in $\delta_f$ (using Lemma 36) and estimating $k + 2 \leq 2(k+1)$, we obtain

$$F_k \leq \frac{2}{1+\nu}\left(\frac{4c_2 c_4 H_f^{\frac{2}{1+\nu}} D^2}{k(k+1)}\left[\frac{1-\nu}{2(1+\nu)}\right]^{\frac{1-\nu}{1+\nu}}\right)^{\frac{1+\nu}{2}}\left(\frac{1+\nu}{1-\nu}\frac{2c_3(k+1)}{3}\right)^{\frac{1-\nu}{2}} + \sigma_k$$

$$= \frac{2(4c_2 c_4)^{\frac{1+\nu}{2}}(\frac{c_3}{3})^{\frac{1-\nu}{2}}}{1+\nu}\frac{H_f D^{1+\nu}}{k^{\frac{1+\nu}{2}}(k+1)^\nu} + \sigma_k.$$

This proves the claimed convergence rate, and the oracle complexity bound easily follows since each iteration of the algorithm requires only $O(1)$ queries to $\widehat{g}_b$. $\qquad\square$

*Remark* 39. The efficiency guarantees given by Corollaries 37 and 38 are exactly the same as those from [49], up to absolute constants.

### F.2 Implicit Variance Reduction

**Corollary 40.** *Consider problem* (1) *under the setting from Example 12 and also under Assumption 2. Let Algorithm 1 be applied to this problem with the oracle $\widehat{g} = \widehat{g}_b$ and initial coefficient $M_0 = 0$. Then, for the point $\bar{x}_N$ generated by the algorithm, we have*

$$\mathbb{E}[F(\bar{x}_N)] - F^* \leq \frac{c_f(\nu)H_f(\nu)D^{1+\nu}}{N^{\frac{1+\nu}{2}}} + \frac{c_{\widehat{g}}(\nu)H_{\max}(\nu)D^{1+\nu}}{(bN)^{\frac{1+\nu}{2}}} + 2\sigma_* D\sqrt{\frac{6c_1 c_4}{bN}},$$

*where $c_f(\nu) := \frac{(2c_2 c_4)^{\frac{1+\nu}{2}} c_3^{\frac{1-\nu}{2}}}{1+\nu} = O(1)$ and $c_{\widehat{g}}(\nu) := \frac{(24c_4)^{\frac{1+\nu}{2}}(\frac{4}{3})^{\frac{1-\nu}{2}}}{1+\nu} = O(1)$. To reach $\mathbb{E}[F(\bar{x}_N)] - F^* \leq \epsilon$ for any $\epsilon > 0$, it suffices to make $O\left(\left[\frac{H_f(\nu)}{\epsilon}\right]^{\frac{2}{1+\nu}} D^2 + \frac{1}{b}\left[\frac{H_{\widehat{g}}(\nu)}{\epsilon}\right]^{\frac{2}{1+\nu}} D^2 + \frac{\sigma_*^2 D^2}{b\epsilon^2}\right)$ queries to $\widehat{g}_b$.*

*Proof.* Denote for brevity $F_N := \mathbb{E}[F(\bar{x}_N)] - F^*$, $H_f := H_f(\nu)$ and $H_{\max} := H_{\max}(\nu)$. Taking into account Remark 35 and applying Theorem 7, we get, for any $\delta_f, \delta_{\hat{g}} > 0$,

$$F_N \le \frac{c_2 c_4 H_f^{\frac{2}{1+\nu}} D^2}{N} \left[\frac{1-\nu}{2(1+\nu)\delta_f}\right]^{\frac{1-\nu}{1+\nu}} + \frac{12 c_4 H_{\max}^{\frac{2}{1+\nu}} D^2}{bN} \left[\frac{1-\nu}{2(1+\nu)\delta_{\hat{g}}}\right]^{\frac{1-\nu}{1+\nu}} + c_3 \delta_f + \frac{4}{3}\delta_{\hat{g}} + \sigma_N,$$

where $\sigma_N := 2\sigma_{*,b} D \sqrt{\frac{6c_1 c_4}{N}} = 2\sigma_* D \sqrt{\frac{6c_1 c_4}{bN}}$. Minimizing the right-hand side in $\delta_f, \delta_{\hat{g}}$ (using Lemma 36 twice), we get

$$F_N \le \frac{2}{1+\nu} \left(\frac{c_2 c_4 H_f^{\frac{2}{1+\nu}} D^2}{N} \left[\frac{1-\nu}{2(1+\nu)}\right]^{\frac{1-\nu}{1+\nu}}\right)^{\frac{1+\nu}{2}} \left(\frac{(1+\nu)c_3}{1-\nu}\right)^{\frac{1-\nu}{2}}$$

$$+ \frac{2}{1+\nu} \left(\frac{12 c_4 H_{\max}^{\frac{2}{1+\nu}} D^2}{bN} \left[\frac{1-\nu}{2(1+\nu)}\right]^{\frac{1-\nu}{1+\nu}}\right)^{\frac{1+\nu}{2}} \left(\frac{(1+\nu)\frac{4}{3}}{1-\nu}\right)^{\frac{1-\nu}{2}} + \sigma_N$$

$$= \frac{(2c_2 c_4)^{\frac{1+\nu}{2}} c_3^{\frac{1-\nu}{2}}}{1+\nu} \frac{H_f D^{1+\nu}}{N^{\frac{1+\nu}{2}}} + \frac{(24 c_4)^{\frac{1+\nu}{2}} (\frac{4}{3})^{\frac{1-\nu}{2}}}{1+\nu} \frac{H_{\max} D^{1+\nu}}{(bN)^{\frac{1+\nu}{2}}} + \sigma_N.$$

This proves the claimed convergence rate, and the oracle complexity bound easily follows since each iteration of the algorithm requires only 1 query to $\hat{g}_b$. $\qquad \square$

**Corollary 41.** *Consider problem* (1) *under the setting from Example 12 and also under Assumption 2. Let Algorithm 2 be applied to this problem with the oracle $\hat{g} = \hat{g}_b$. Then, for any $k \ge 1$, we have*

$$\mathbb{E}[F(x_k)] - F^* \le \frac{c_f(\nu) H_f(\nu) D^{1+\nu}}{k^{\frac{1+3\nu}{2}}} + \frac{c_{\hat{g}}(\nu) H_{\max}(\nu) D^{1+\nu}}{(bk)^{\frac{1+\nu}{2}}} + 4\sigma_* D \sqrt{\frac{2c_1 c_4}{bk}},$$

*where $c_f(\nu) := \frac{(8c_2 c_4)^{\frac{1+\nu}{2}} (\frac{2}{3}c_3)^{\frac{1-\nu}{2}}}{1+\nu} = O(1)$ and $c_{\hat{g}}(\nu) := \frac{(48 c_1 c_4)^{\frac{1+\nu}{2}} (\frac{4}{3})^{\frac{1-\nu}{2}}}{1+\nu} = O(1)$. To reach $\mathbb{E}[F(x_k)] - F^* \le \epsilon$ for any $\epsilon > 0$, it suffices to make $O\big(\big[\frac{H_f(\nu) D^{1+\nu}}{\epsilon}\big]^{\frac{2}{1+3\nu}} + \frac{1}{b}\big[\frac{H_{\max}(\nu)}{\epsilon}\big]^{\frac{2}{1+\nu}} D^2 + \frac{\sigma_*^2 D^2}{b\epsilon^2}\big)$ queries to $\hat{g}_b$.*

*Proof.* Let $k \ge 1$ be arbitrary and denote for brevity $F_k := \mathbb{E}[F(x_k)] - F^*$, $H_f := H_f(\nu)$ and $H_{\max} := H_{\max}(\nu)$. Taking into account Remark 35 and applying Theorem 8, we get, for any $\delta_f, \delta_{\hat{g}} > 0$,

$$F_k \le \frac{4c_2 c_4 H_f^{\frac{2}{1+\nu}} D^2}{k(k+1)} \left[\frac{1-\nu}{2(1+\nu)\delta_f}\right]^{\frac{1-\nu}{1+\nu}} + \frac{24 c_1 c_4 H_{\max}^{\frac{2}{1+\nu}} D^2}{bk} \left[\frac{1-\nu}{2(1+\nu)\delta_{\hat{g}}}\right]^{\frac{1-\nu}{1+\nu}}$$

$$+ \frac{c_3}{3}(k+2)\delta_f + \frac{4}{3}\delta_{\hat{g}} + \sigma_k,$$

where $\sigma_k := 4\sigma_{*,b} D \sqrt{\frac{2c_1 c_4}{k}} = 4\sigma_* D \sqrt{\frac{2c_1 c_4}{bk}}$. Minimizing the right-hand side in $\delta_f$ and $\delta_{\hat{g}}$ (using Lemma 36 twice) and estimating $\frac{1}{3}(k+2) \le \frac{2}{3}(k+1)$, we obtain

$$F_k \le \frac{2}{1+\nu} \left(\frac{4c_2 c_4 H_f^{\frac{2}{1+\nu}} D^2}{k(k+1)} \left[\frac{1-\nu}{2(1+\nu)}\right]^{\frac{1-\nu}{1+\nu}}\right)^{\frac{1+\nu}{2}} \left(\frac{(1+\nu)\frac{2c_3}{3}(k+1)}{1-\nu}\right)^{\frac{1-\nu}{2}}$$

$$+ \frac{2}{1+\nu} \left(\frac{24 c_1 c_4 H_{\max}^{\frac{2}{1+\nu}} D^2}{bk} \left[\frac{1-\nu}{2(1+\nu)}\right]^{\frac{1-\nu}{1+\nu}}\right)^{\frac{1+\nu}{2}} \left(\frac{(1+\nu)\frac{4}{3}}{1-\nu}\right)^{\frac{1-\nu}{2}} + \sigma_k$$

$$= \frac{(8c_2 c_4)^{\frac{1+\nu}{2}} (\frac{2}{3}c_3)^{\frac{1-\nu}{2}}}{1+\nu} \frac{H_f D^{1+\nu}}{k^{\frac{1+\nu}{2}}(k+1)^{\nu}} + \frac{(48 c_1 c_4)^{\frac{1+\nu}{2}} (\frac{4}{3})^{\frac{1-\nu}{2}}}{1+\nu} \frac{H_{\max} D^{1+\nu}}{(bk)^{\frac{1+\nu}{2}}} + \sigma_k.$$

This proves the claimed convergence rate, and the oracle complexity bound easily follows since each iteration of the algorithm requires only $O(1)$ queries to $\hat{g}_b$. $\qquad \square$

*Remark* 42. In the proof of Corollary 41, it was important that $\delta_f$ and $\delta_{\hat{g}}$ were allowed to be two separate constants. If we were not paying attention to such a separation and simply used the same $\delta$ everywhere, we would end up with the much weaker rate of $O(\frac{H_{\max}(\nu) D^{1+\nu}}{b^{\frac{1+\nu}{2}} k^{\nu}})$ for the second term.

### F.3 Explicit Variance Reduction with SVRG

**Corollary 43.** *Consider problem* (1) *under the setting from Example 12 and also under Assumption 2. Let* UniSvrg *(as defined by Algorithm 3) be applied to this problem with the stochastic oracle $\widehat{g} = \widehat{g}_b$ and the full-gradient oracle $\bar{g} = \nabla f$. Then, for any $t \geq 1$,*

$$\mathbb{E}[F(\tilde{x}_t)] - F^* \leq \frac{c_f(\nu)H_f(\nu)D^{1+\nu}}{(2^t)^{\frac{1+\nu}{2}}} + \frac{c_{\widehat{g}}(\nu)H_{\max}(\nu)D^{1+\nu}}{(b2^t)^{\frac{1+\nu}{2}}},$$

*where $c_f(\nu) := \frac{[2(c_2c_4+1)]^{\frac{1+\nu}{2}}(2\bar{c}_3)^{\frac{1-\nu}{2}}}{1+\nu} = O(1)$, $c_{\widehat{g}}(\nu) := \frac{(96c_1c_4)^{\frac{1+\nu}{2}}(\frac{8}{3})^{\frac{1-\nu}{2}}}{1+\nu} = O(1)$, $\bar{c}_3 := \max\{c_3, 1\}$. To get $\mathbb{E}[F(\tilde{x}_t)] - F^* \leq \epsilon$, it suffices to make $O(N_\nu(\epsilon))$ queries to $\widehat{g}_b$ and $O(\log_+ N_\nu(\epsilon))$ queries to $\nabla f$, where $N_\nu(\epsilon) := [\frac{H_f(\nu)}{\epsilon}]^{\frac{2}{1+\nu}}D^2 + \frac{1}{b}[\frac{H_{\max}(\nu)}{\epsilon}]^{\frac{2}{1+\nu}}D^2$. Assuming that the complexity of querying $\bar{g}_b$ is $n_b$ times bigger than that of querying $\nabla f$, we get the total stochastic-oracle complexity of $O(N_\nu(\epsilon) + n_b \log_+ N_\nu(\epsilon))$.*

*Proof.* Let $t \geq 1$ be arbitrary and denote for brevity $F_t := \mathbb{E}[F(\tilde{x}_t)] - F^*$, $H_f := H_f(\nu)$ and $H_{\max} := H_{\max}(\nu)$. Taking into account Remark 35 and applying Theorem 10, we get, for any $\delta_f, \delta_{\widehat{g}} > 0$,

$$F_t \leq \frac{(c_2c_4+1)H_f^{\frac{2}{1+\nu}}D^2}{2^t}\left[\frac{1-\nu}{2(1+\nu)\delta_f}\right]^{\frac{1-\nu}{1+\nu}} + \frac{48c_1c_4 H_{\max}^{\frac{2}{1+\nu}}D^2}{b2^t}\left[\frac{1-\nu}{2(1+\nu)\delta_{\widehat{g}}}\right]^{\frac{1-\nu}{1+\nu}} + 2\bar{c}_3\delta_f + \frac{8}{3}\delta_{\widehat{g}}.$$

Minimizing the right-hand side in $\delta_f, \delta_{\widehat{g}}$ (using Lemma 36 twice), we obtain

$$F_t \leq \frac{2}{1+\nu}\left(\frac{(c_2c_4+1)H_f^{\frac{2}{1+\nu}}D^2}{2^t}\left[\frac{1-\nu}{2(1+\nu)}\right]^{\frac{1-\nu}{1+\nu}}\right)^{\frac{1+\nu}{2}}\left(\frac{(1+\nu)2\bar{c}_3}{1-\nu}\right)^{\frac{1-\nu}{2}}$$
$$+ \frac{2}{1+\nu}\left(\frac{48c_1c_4 H_{\max}^{\frac{2}{1+\nu}}D^2}{b2^t}\left[\frac{1-\nu}{2(1+\nu)}\right]^{\frac{1-\nu}{1+\nu}}\right)^{\frac{1+\nu}{2}}\left(\frac{(1+\nu)\frac{8}{3}}{1-\nu}\right)^{\frac{1-\nu}{2}}$$
$$= \frac{c_f H_f D^{1+\nu}}{(2^t)^{\frac{1+\nu}{2}}} + \frac{c_{\widehat{g}}H_{\max}D^{1+\nu}}{(b2^t)^{\frac{1+\nu}{2}}},$$

where $c_f := \frac{[2(c_2c_4+1)]^{\frac{1+\nu}{2}}(2\bar{c}_3)^{\frac{1-\nu}{2}}}{1+\nu}$ and $c_{\widehat{g}} := \frac{(96c_1c_4)^{\frac{1+\nu}{2}}(\frac{8}{3})^{\frac{1-\nu}{2}}}{1+\nu}$. This proves the claimed convergence rate.

Let us now estimate the oracle complexity. From the already proved convergence rate bound, we see that $F_t \leq \epsilon$ once $2^t \geq O(1)N(\epsilon)$, where $N(\epsilon) := [\frac{H_f}{\epsilon}]^{\frac{2}{1+\nu}}D^2 + \frac{1}{b}[\frac{H_{\max}}{\epsilon}]^{\frac{2}{1+\nu}}D^2$. At the same time, according to Theorem 10, to generate the corresponding $\tilde{x}_t$, the algorithm needs to make $O(2^t)$ queries to $\widehat{g}_b$ and $O(t)$ queries to $\nabla f$. Combining these two facts together, we get the claimed $O(N(\epsilon))$ queries to $\widehat{g}_b$ and $O(\log_2 N(\epsilon) + 1) = O(\log_+ N(\epsilon))$ queries to $\nabla f$. $\square$

**Corollary 44.** *Consider problem* (1) *under the setting from Example 12 and also under Assumption 2. Let* UniFastSvrg *(Algorithm 4) be applied to this problem with the stochastic oracle $\widehat{g} = \widehat{g}_b$, the full-gradient oracle $\bar{g} = \nabla f$, and the epoch length $N \geq 9$. Then, for any $t \geq 2t_0$, where $t_0 := \lceil \log_2 \log_3 N \rceil - 1 \ (\geq 0)$, it holds that*

$$\mathbb{E}[F(\tilde{x}_t)] - F^* \leq \frac{c_f(\nu)H_f(\nu)D^{1+\nu}}{N^{\frac{1+\nu}{2}}(t+1)^{\frac{1+3\nu}{2}}} + \frac{c_{\widehat{g}}(\nu)H_{\max}(\nu)D^{1+\nu}}{(bN)^{\frac{1+\nu}{2}}(t+1)^{\frac{1+3\nu}{2}}},$$

*where $c_f(\nu) := \frac{[72(c_2c_4+\frac{1}{2})]^{\frac{1+\nu}{2}}\bar{c}_3^{\frac{1-\nu}{2}}}{1+\nu} = O(1)$, $c_{\widehat{g}}(\nu) := \frac{(432c_1c_4)^{\frac{1+\nu}{2}}(\frac{5}{3})^{\frac{1-\nu}{2}}}{1+\nu} = O(1)$, $\bar{c}_3 := \max\{c_3, 1\}$. To get $\mathbb{E}[F(\tilde{x}_t)] - F^* \leq \epsilon$, it suffices to make $O(NT_\nu(\epsilon))$ queries to $\widehat{g}_b$ and $O(T_\nu(\epsilon))$ queries to $\nabla f$, where $T_\nu(\epsilon) := [\frac{H_f(\nu)D^{1+\nu}}{N^{\frac{1+\nu}{2}}\epsilon}]^{\frac{2}{1+3\nu}} + [\frac{H_{\max}(\nu)D^{1+\nu}}{(bN)^{\frac{1+\nu}{2}}\epsilon}]^{\frac{2}{1+3\nu}} + \log_2\log_3 N$. Assuming that the complexity of querying $\bar{g}_b$ is $n_b$ times bigger than that of querying $\nabla f$ and choosing $N = \Theta(n_b)$, we get the total stochastic-oracle complexity of $O\left([\frac{n_b^\nu H_f(\nu)D^{1+\nu}}{\epsilon}]^{\frac{2}{1+3\nu}} + [\frac{n_b^\nu H_{\max}(\nu)D^{1+\nu}}{b^{(1+\nu)/2}\epsilon}]^{\frac{2}{1+3\nu}} + n_b\log\log n_b\right)$*

*Proof.* Let $t \geq 2t_0$ be arbitrary, $F_t := \mathbb{E}[F(\tilde{x}_t)] - F^*$, $H_f := H_f(\nu)$, $H_{\max} := H_{\max}(\nu)$. Taking into account Remark 35 and applying Theorem 11, we get, for any $\delta_f, \delta_{\hat{g}} > 0$,

$$F_t \leq \frac{9(c_2 c_4 + \frac{1}{2})H_f^{\frac{2}{1+\nu}}D^2}{N(t-t_0+1)^2}\left[\frac{1-\nu}{2(1+\nu)\delta_f}\right]^{\frac{1-\nu}{1+\nu}} + \frac{54 c_1 c_4 H_{\max}^{\frac{2}{1+\nu}}D^2}{bN(t-t_0+1)^2}\left[\frac{1-\nu}{2(1+\nu)\delta_{\hat{g}}}\right]^{\frac{1-\nu}{1+\nu}}$$
$$+ (c_3 t + 1)\delta_f + \frac{5}{3}t\delta_{\hat{g}}.$$

Since $t \geq 2t_0$, we can estimate $t - t_0 + 1 = \frac{1}{2}t + \frac{1}{2}t - t_0 + 1 \geq \frac{1}{2}(t+1)$, which gives us

$$F_t \leq \frac{36(c_2 c_4 + \frac{1}{2})H_f^{\frac{2}{1+\nu}}D^2}{N(t+1)^2}\left[\frac{1-\nu}{2(1+\nu)\delta_f}\right]^{\frac{1-\nu}{1+\nu}} + \frac{216 c_1 c_4 H_{\max}^{\frac{2}{1+\nu}}D^2}{bN(t+1)^2}\left[\frac{1-\nu}{2(1+\nu)\delta_{\hat{g}}}\right]^{\frac{1-\nu}{1+\nu}}$$
$$+ \bar{c}_3(t+1)\delta_f + \frac{5}{3}(t+1)\delta_{\hat{g}}.$$

Minimizing the right-hand side in $\delta_f, \delta_{\hat{g}}$ (using Lemma 36 twice), we obtain

$$F_t \leq \frac{2}{1+\nu}\left(\frac{36(c_2 c_4 + \frac{1}{2})H_f^{\frac{2}{1+\nu}}D^2}{N(t+1)^2}\left[\frac{1-\nu}{2(1+\nu)}\right]^{\frac{1-\nu}{1+\nu}}\right)^{\frac{1+\nu}{2}}\left(\frac{(1+\nu)\bar{c}_3(t+1)}{1-\nu}\right)^{\frac{1-\nu}{2}}$$
$$+ \frac{2}{1+\nu}\left(\frac{216 c_1 c_4 H_{\max}^{\frac{2}{1+\nu}}D^2}{bN(t+1)^2}\left[\frac{1-\nu}{2(1+\nu)}\right]^{\frac{1-\nu}{1+\nu}}\right)^{\frac{1+\nu}{2}}\left(\frac{(1+\nu)\frac{5}{3}(t+1)}{1-\nu}\right)^{\frac{1-\nu}{2}}$$
$$= \frac{c_f H_f D^{1+\nu}}{N^{\frac{1+\nu}{2}}(t+1)^{\frac{1+3\nu}{2}}} + \frac{c_{\hat{g}}H_{\max}D^{1+\nu}}{(bN)^{\frac{1+\nu}{2}}(t+1)^{\frac{1+3\nu}{2}}},$$

where $c_f := \frac{[72(c_2 c_4 + \frac{1}{2})]^{\frac{1+\nu}{2}}\bar{c}_3^{\frac{1-\nu}{2}}}{1+\nu}$ and $c_{\hat{g}} := \frac{(432 c_1 c_4)^{\frac{1+\nu}{2}}(\frac{5}{3})^{\frac{1-\nu}{2}}}{1+\nu}$. This proves the claimed convergence rate.

Let us now estimate the number of oracle queries. In view of the above convergence rate bound, we have $F_t \leq \epsilon$ once $t \geq T(\epsilon) := T_1(\epsilon) + 2t_0 = O\big(T_1(\epsilon) + \log\log N\big)$, where $T_1(\epsilon) := [\frac{H_f D^{1+\nu}}{N^{(1+\nu)/2}\epsilon}]^{\frac{2}{1+3\nu}} + [\frac{H_{\max}D^{1+\nu}}{(bN)^{(1+\nu)/2}\epsilon}]^{\frac{2}{1+3\nu}} = \frac{T_2(\epsilon)}{N^{\frac{1+\nu}{1+3\nu}}}$, where $T_2(\epsilon) := [\frac{H_f D^{1+\nu}}{\epsilon}]^{\frac{2}{1+3\nu}} + [\frac{H_{\max}D^{1+\nu}}{b^{(1+\nu)/2}\epsilon}]^{\frac{2}{1+3\nu}}$ does not depend on $N$. Combining this with Theorem 11 saying that, to generate the corresponding $\tilde{x}_t$, the algorithm needs to make $O(Nt)$ queries to $\hat{g}_b$ and $O(t)$ queries to $\nabla f$, we get the claimed $O(NT(\epsilon))$ queries to $\hat{g}_b$ and $O(T(\epsilon))$ queries to $\nabla f$.

Assuming now that the complexity of querying $\nabla f$ is $n_b$ times bigger than that of querying $\hat{g}_b$, we get the total stochastic-oracle complexity of $O\big((N + n_b)T(\epsilon)\big) = O\big((N + n_b)\big[\frac{T_2(\epsilon)}{N^{(1+\nu)/(1+3\nu)}} + \log\log N\big]\big)$. Ignoring the doubly-logarithmic term, we get the expression of the form $(N + n_b)\frac{1}{N^q} = N^{1-q} + \frac{n_b}{N^q}$ with $q := \frac{1+\nu}{1+3\nu} \in [0, 1]$, whose minimal value is achieved at $N = \Theta(n_b)$. Substituting this value into our complexity bound, we get the stochastic-oracle complexity of $O\big(n_b(\frac{T_2(\epsilon)}{n_b^{(1+\nu)/(1+3\nu)}} + \log\log n_b)\big) = O\big(n_b^{\frac{2\nu}{1+3\nu}}T_2(\epsilon) + n_b \log\log n_b\big)$. $\quad\square$

## G   Additional Discussion of Related Work

**Inexact Oracle and Approximate Smoothness.**   Devolder, Glineur, and Nesterov [15] introduced the notion of the inexact first-order oracle and analyzed the behaviour of several first-order methods for smooth convex optimization using such an oracle. Although their work was motivated by the desire to present the general definition of an inexact oracle covering many different applications, it was also observed that this oracle model is suitable for studying weakly smooth problems. This insight was later used in [45] to develop universal gradient methods for Hölder smooth problems. First stochastic gradient methods for approximately smooth functions with inexact oracle were proposed in [13]. These algorithms however are not adaptive and require the knowledge of problem-dependent constants. For more details on the subject, see [14].

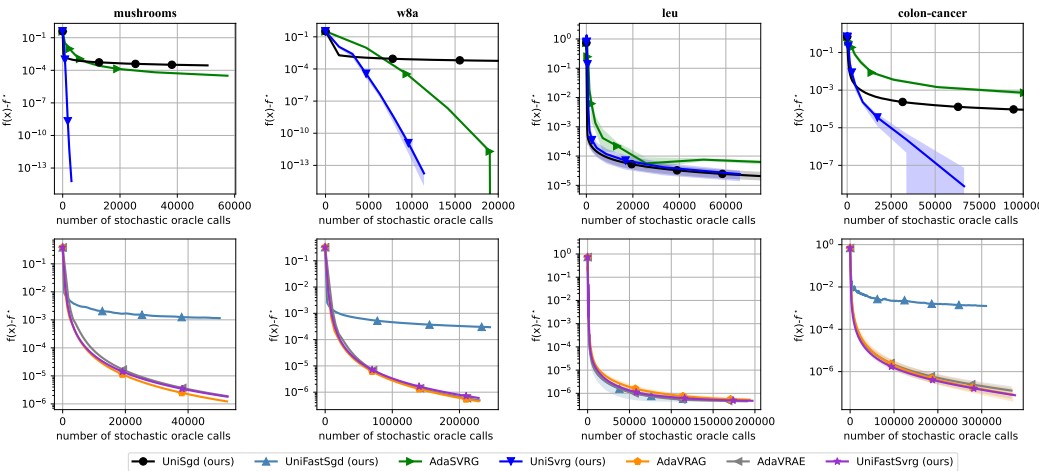

Figure 3: Comparison of various methods on the logistic regression problem with real-world data.

**Parameter-Free Methods.** Parameter-free algorithms originating from the literature on online learning [9, 10, 26, 40, 47, 55] is another popular type of adaptive methods. They are usually endowed with mechanisms helping achieving efficiency bounds that are almost insensitive (typically, with logarithmic dependency) to the error of estimating certain problem parameters, such as the diameter of the feasible set [6, 11, 25, 31, 41].

**Variance Reduction.** Variance reduction techniques encompass a set of strategies that enhance the convergence speed of SGD when multiple passes are possible over the training dataset. Various researchers simultaneously introduced methods to reduce variance around the same period [27, 38, 50, 52, 58, 60]. The consideration of mini-batching in the context of these methods is documented in [3], while, in [20], it is shown that the convergence rate is influenced by both the average and the maximum smoothness of individual components. For further details, see [21] and the references therein.

Sometimes, it is even not necessary to use an explicit variance reduction mechanism. SGD may converge fast in the so-called over-parameterized regime, or when the stochastic noise is small at the optimal solution [8, 35, 37, 43, 44, 51]. In this work, we call this effect implicit variance reduction. Such a situation is also considered in [19, 54] and, more recently, Woodworth and Srebro [59] proposed an accelerated SGD algorithm for this setting, under the assumption that the smoothness and noise constants are known.

## H    Additional Experiments

### H.1    Logistic Regression with Real-World Data

In this section, we present experiments on the *logistic regression problem*:

$$f^* = \min_{\|x\| \leq R} \left\{ f(x) \coloneqq \frac{1}{n} \sum_{i=1}^{n} \log(1 + e^{-b_i \langle a_i, x \rangle}) \right\},$$

where $a_i \in \mathbb{R}^d$ and $b_i \in \{-1, 1\}$ are features and labels taken from diverse real-world datasets from LIBSVM [7]: mushrooms ($d \ll n$), w8a ($d \ll n$), leu ($d \gg n$) and colon-cancer ($d \gg n$). The dataset leu is quite special because it satisfies the so-called interpolation condition, meaning that the variance at the optimum is zero. We fix $R = 1$ and use the mini-batch size of $b = 32$ for the first two datasets and $b = 1$ for the last two.

Figure 3 shows the results of our experiments. The solid lines and the shaded area for each method represent, respectively, the mean and the region between the minimum and the maximum values after three independent runs of the algorithm. We see that, on the leu dataset, UniSgd and UniFastSgd converge as fast as the best non-accelerated and accelerated SVRG methods, respectively, which

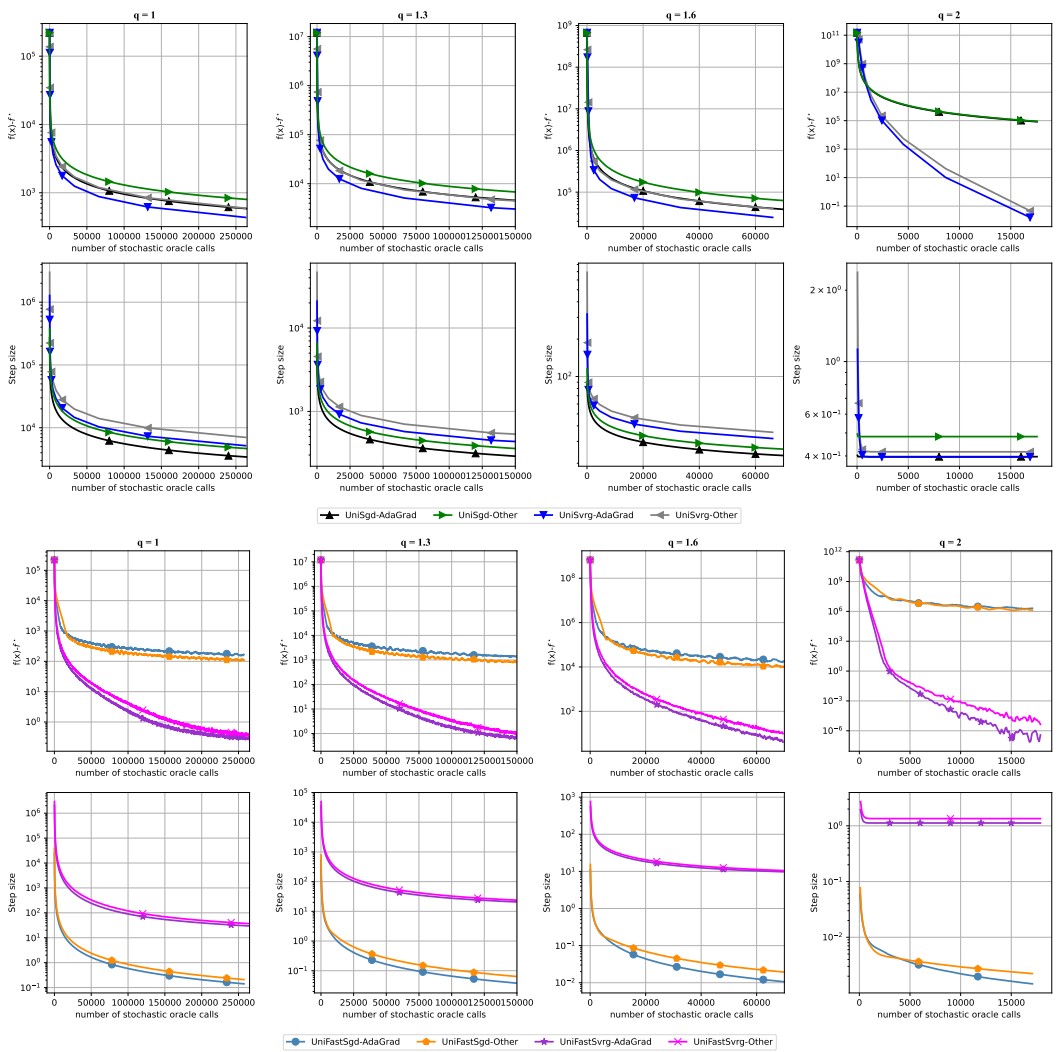

Figure 4: Comparison of our methods for different stepsize update rules on the polyhedron feasibility problem.

confirms our theory on implicit variance reduction. Otherwise, these two SGD methods are typically much slower than the SVRG algorithms. Our UniSvrg method performs consistently better than AdaSVRG across all the datasets. Overall, all adaptive accelerated SVRG methods demonstrate comparable performance for solving these smooth problems.

## H.2 Comparison between Stepsize Update Rules

In this section, we compare the AdaGrad stepsize rule (3) with the other rule (4) for UniSgd (Algorithm 1), UniFastSgd (Algorithm 2), UniSvrg (Algorithm 3), and UniFastSvrg (Algorithm 4). We consider the polyhedron feasibility and logistic regression problems under the same setups as in Section 8 and Appendix H.1.

The results are shown in Figs. 4 and 5, where we plot the function residual and the stepsize (inverse of $M$) against stochastic oracle calls. We see that the two stepsize rules work very similarly across all test cases, which was not evident from the theory alone.

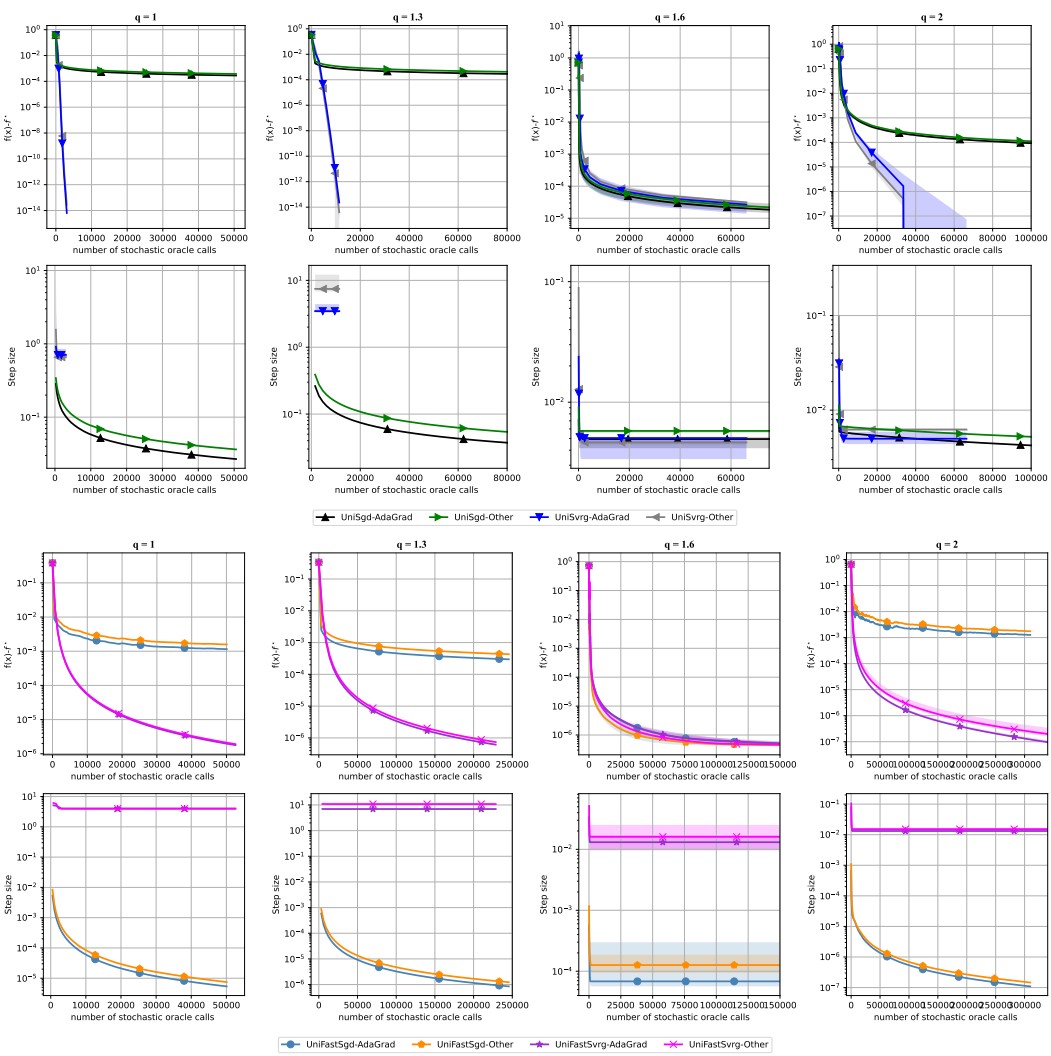

Figure 5: Comparison of our methods for different stepsize update rules on the logistic regression problem with real-world data.

