# OpenReview forum: "Universality of AdaGrad Stepsizes for Stochastic Optimization: Inexact Oracle, Acceleration and Variance Reduction"
_NeurIPS.cc/2024/Conference — NeurIPS 2024 poster_

### Official Review · Reviewer_AhYx · 2024-06-14

**Soundness:** 3
**Presentation:** 2
**Contribution:** 2
**Rating:** 5
**Confidence:** 4

**Summary:**

The paper explores the adaptive gradient methods for solving convex composite optimization problems. The authors propose two main algorithms, UniSgd and UniFastSgd, equipped with AdaGrad stepsizes. They establish efficiency guarantees under various conditions, demonstrating implicit and explicit variance reduction properties. The results are also extended to incorporate SVRG-type variance reduction, leading to faster convergence rates.

**Strengths:**

1. This paper extends existing results by showing the universality of AdaGrad stepsizes in a more general setting, providing efficiency guarantees and state-of-the-art complexity bounds.
2. The proposed methods are adaptive, not requiring problem-specific constants, making them practical for real-world applications.
3. Numerical experiments validate the theoretical findings, showing the practical effectiveness of the proposed methods.

**Weaknesses:**

1. Although the authors call their algorithm adaptive methods, the proposed algorithm still requires knowing the domain diameter, which is an obvious drawback. Also, the reliance on the boundedness of the domain might also limit the application of the methods in more general or practical scenarios.
2. The main body of this paper is difficult to understand. It seems that the algorithms are all rather complicated and the authors do not explain the methods clearly. Also, I think the authors should further emphasize the novelty and the challenge of this paper.
3. More extensive empirical evaluations across a wider range of practical problems on real-world datasets would strengthen the experiments part of this paper.
4. The authors claim in the abstract that “the main part can be accessed only via a (potentially **biased**) stochastic gradient oracle”, but it seems that the paper assumes an unbiased stochastic oracle. Maybe I missed something, but I do not find where we use the biased oracle. Can you explain this more clearly?

**Questions:**

See the weaknesses part.

---

> ### Author Rebuttal · Authors · 2024-08-07
>
> Thanks for your time and efforts spent on reviewing this manuscript. We
> appreciate the feedback and would like to make some comments on the points you
> raised in your review.
>
> - [W1]:
>   We agree that this is a drawback and it was explicitly written in the paper.
>   Note, however, that $D$ is indeed the **only** parameter that the methods need
>   to know. This is the price one pays for getting completely universal methods,
>   automatically adapting to a large variety of other parameters (degree of
>   smoothness $\nu$ and the corresponding smoothness constant $H_{\nu}$, oracle's
>   variance $\sigma$, variance at the minimizer $\sigma_*$, oracle's inexactness
>   $\delta$, etc.), while enjoying the optimal worst-case complexity estimates,
>   typical for non-adaptive methods. We are not aware of any other algorithms
>   with similar properties, especially, working in such a large number of
>   situations and with such a unified treatment as in our work (basic methods and
>   accelerated ones, with variance reduction and without it, etc.).
>
>   While we do agree that it would be very nice to have completely parameter-free
>   algorithms, it seems that this goal may not be achievable for stochastic
>   optimization problems, at least not without additional, somewhat restrictive,
>   assumptions. Please refer to our detailed reply to Reviewer PtE8.
>
> - [W2]:
>   Thank you for the comment. We agree that it would be better to provide more
>   explanations and intuitions. However, we are rather limited by the page limit
>   and therefore must think twice if we want to add anything extra. Could you
>   please clarify where exactly you would like us to elaborate, except better
>   emphasizing the novelty and the challenge?
>
> - [W3]:
>   Note that our paper is mostly theoretical, and we do not attempt to be
>   exhaustive with experiments. But we will take your point into account and add
>   more experiments in the revised version of the paper.
>
> - [W4]:
>   Yes, of course. Our oracle $\hat{g}$ is assumed to be an unbiased estimate of
>   $\bar{g}$, which is itself only an **approximation** to the true gradient
>   $\nabla f$. Thus, $\hat{g}$ is, in general, a biased estimate of $\nabla f$:
>   $\mathbb{E}\_{\xi}[\hat{g}(x, \xi)] = \bar{g}(x) \neq \nabla f(x)$. Some
>   particular examples where we have biased gradients are mentioned in lines
>   114-122 and also in lines 186-194. For instance, if we work with finite-sum
>   problems, $f(x) = \frac{1}{n} \sum_{i = 1}^n f_i(x)$, where each function
>   $f_i$ represents another nontrivial optimization problem with a strongly
>   convex objective, $f_i(x) = \max_{u_i} \Psi_i(x, u_i)$, whose solution
>   $\bar{u}\_i(x)$ can be found with accuracy $\delta$, then the natural
>   stochastic gradient $\hat{g}(x, \xi) = \nabla_x \Psi_{\xi}(x,  \bar{u}_{\xi}(x))$
>   (with uniformly chosen index $\xi$) is a biased estimate of
>   $\nabla f(x)$, and $\delta_f = \delta$.
>
> We hope we were able to address all your concerns and kindly ask you to increase
> your score.

---

> > ### Comment · Reviewer_AhYx · 2024-08-12
> >
> > Thanks for your rebuttal. It addresses most of my concerns and I am happy to increase my score from 4 to 5.

---

> > > ### Author Response · Authors · 2024-08-13
> > >
> > > Thank you. Let us know please if you have any other questions that we can answer
> > > to resolve your remaining concerns.

---

### Official Review · Reviewer_aWSb · 2024-07-08

**Soundness:** 4
**Presentation:** 3
**Contribution:** 3
**Rating:** 6
**Confidence:** 4

**Summary:**

This paper proves new convergence rates for stochastic gradient methods with
AdaGrad step-sizes. The authors' build on the fact that AdaGrad step-sizes
adapt to both the smooth and non-smooth settings and extend these results to
show convergence with biased stochastic gradient oracles on functions which are
only approximately smooth. Because Hölder-smooth functions satisfy this class,
this paper shows that AdaGrad step-sizes lead to universal methods which only
require knowledge of the diameter, $D$. As part of their analysis, the authors
also show how to relax the bounded variance assumption and combine AdaGrad
step-sizes with acceleration and variance reduction.

**Strengths:**

- The theoretical setting is extremely general, covering composite optimization
    with Hölder smooth functions with biased stochastic gradient oracles
    satisfying a relaxed variance bound.

- The analysis covers SGD as well as accelerated SGD and (accelerated) SGD
    with variance reduction.

- The synthetic experiments show that the proposed methods are faster than
    existing methods and adapt to different degrees of smoothness in the
    objective function.

**Weaknesses:**

- The contributions are fairly niche. Universality of (accelerated) SGD with
    modified AdaGrad step-sizes is known [14], Adagrad is known to converge
    without bounded variance [2], and AdaGrad step-sizes have been used with
    (accelerated) SVRG previously [14, 34]. Thus, this paper fits into a small
    gap by proving universality with a relaxed variance condition and covering
    both acceleration and variance reduction.

- ~The actually degree to which bounded variance is relaxed is not clear since
    the connections between Assumption 3 and Assumption 6 are not developed.~

- ~Variance reduce SGD with AdaGrad step-sizes (UniSvrg) only converges to a
    neighbourhood of the optimal solution even when using an unbiased stochastic
    gradient oracle.~

- Experiments are only for synthetic problems and do not show the actual relationship
    between oracle calls and optimization error.

### Detailed Comments

**Note**: The authors have clarified the connection between Assumptions 3 and 6 as well the role of $\delta_g$ in Theorem 10.

**Assumption 3 vs Assumption 6/9**:

An important and unaddressed issue is the comparison between between
Assumption 3 and Assumption 6/9. Clearly Assumption 3 implies Assumption 6/9
up to constant factors. But, in what circumstances do Assumptions 6/9 hold
but the variance is not bounded, i.e. Assumption 3 fails? In comparison to this
work, Nguyen et al. [2] give results for (strongly-convex) SGD that depend
only on the stochastic gradient variance at the minimizer and without
any variance assumption. Indeed, the typical assumption in this setting is only
that $\mathbb{E}[\|g(x^*, \xi)\|_2^2]$ is finite.


**Theorem 10**

I'm somewhat disturbed that SVRG with Adagrad step-sizes doesn't converge
exactly to a minimizer $x^*$ under your assumptions. For example, suppose
$\nabla f$ is $L$-Lipschitz and $\bar g = \nabla f$ so that $\delta_f$ is zero.
In this setting, I expect variance-reduced methods like SVRG to converge
exactly to a minimizer, but Theorem 10 shows UniSVRG converges to a
neighbourhood of size $O(\delta_{\hat g})$. In comparison, standard SVRG, which
does not require any variance bound (Allen-Zhu and Yuan, [1]), converges
exactly. Can you please give sufficient conditions for $\delta_{\hat g}$ to be
zero and explain how they compare to standard SVRG assumptions?

Also, why are both Assumptions 6 and 9 required simultaneously for this
theorem? They provide similar bounds on the variance of the gradient
estimators, so I would have thought that Assumption 9 supplants Assumption 6.

**Experiments**:

I would have liked to see more realistic experiments. I understand that finding
real-world problems for which the diameter $D$ is bounded can be difficult, but
one synthetic regression problem is not sufficient to judge the performance of
the methods analyzed in this paper. In comparison, Dubois-Taine et al. [14]
include a large number of experiments on real-world data.

In addition, I strongly dislike the choice to treat one mini-batch gradient
computation as a single oracle call instead of $b$ calls.  While the choice has
no effects in Figure 1, it is actually quite misleading in Figure 2, where it
makes the total SFO complexity of all the algorithms appear to decrease as
batch-size increases. In reality, the SFO complexity should decrease until a
threshold and then $b$ and then begin to increase again. That is, there is some
optimal batch-size (usually less than $n$) which minimizes total complexity


### Minor Comments

- "Gradient methods are the most popular and efficient optimization algorithms
    for solving machine learning problems." --- They are not the most efficient
    methods depending on the problem class and desired sub-optimality, so I
    wouldn't say this.

- "However, the line-search approach is unsuitable for problems of stochastic
    optimization, where gradients are observed with random noise." --- you may
    want to mention that line-search works with stochastic gradients when
    interpolation is satisfied [1].

- So the difference between the results in Section 4 and [47] is that [47]
    considers a modification of the Adagrad step-size? You might want to
    state this modification so the difference is clear.

- Contributions, Bullet 3: I don't like calling this variance reduction. It is
    well-known that SGD with a constant step-size depends only on the variance
    of the stochastic gradients at the minimizer [2], but no one would call SGD
    variance reduced.

- Table 1:
    - Can you make the font-size larger? It's a little hard to read this
    table.
    - If you are reporting Big-O complexities, then can you please put them
        in Big-O notation?
    - You should explain in the caption why UniSgd and UniFastSgd have two
        convergence rates each (e.g. variance assumptions).

- Line 117: If it's satisfied for any $\delta_f > 0$ then taking limits as
    $\delta_f \rightarrow 0$ would imply that it's also satisfies for
    $\delta_f = 0$.

- Line 136: I would say the classical method is due to Nesterov [3], while you
    are referring to more recent acceleration methods for composite
    optimization.

- Eq. (4): How you solve this implicit equation, given that $\hat x_+$ depends
    on $\hat M_+$ through the stochastic gradient update?

- Assumption 6: In line 126, you say that the oracle outputs are assumed to be
    independent. However, from the notation in Assumption 6, it seems that
    in this case the oracle is evaluated with the same randomness $\xi$.
    Indeed, this must be the case else Assumption 6 reduces to Assumption 3.
    Maybe you can add a remark clarifying this in the text?

- Theorems 4,7: You should comment somewhere in the text that these rates
    (according to Algorithm 1) are not anytime, but require knowledge of $N$.

- Line 230: This sentence isn't finished.

- Page 7: The spacing between the algorithm blocks is quite weird. You
    should probably fix this.

- Theorem 11: I would prefer if you stated this without assuming
    $N \in \Theta(n)$.

- Table 2, row 3: I think there's a typographic issue here since you appear to
    be adding the definition of $N_\nu(\epsilon)$ with the log.

- Figures 1, 2:
    - The font sizes are much too small to read. As a rule of thumb,
        the size of text in a figure should be at least as large as the size of
        text in paragraph mode.
    - Figure 1: It would be nice if you reminded the reader that $q$ is the
        power in the test problem in Eq. (5).

- Line 916: You should also reference SDCA [4] and SAG [5], which were very
    early (if not the earliest) variance reduction methods.

### References


[1] Vaswani, Sharan, et al. "Painless stochastic gradient: Interpolation, line-search, and convergence rates." Advances in neural information processing systems 32 (2019).

[2] Nguyen, Lam, et al. "SGD and Hogwild! convergence without the bounded gradients assumption." International Conference on Machine Learning. PMLR, 2018.

[3] Nesterov, Yurii. "A method for solving the convex programming problem with convergence rate O (1/k2)." Dokl akad nauk Sssr. Vol. 269. 1983.

[4] Shalev-Shwartz, Shai, and Tong Zhang. "Stochastic dual coordinate ascent methods for regularized loss minimization." Journal of Machine Learning Research 14.1 (2013).

[5] Schmidt, Mark, Nicolas Le Roux, and Francis Bach. "Minimizing finite sums with the stochastic average gradient." Mathematical Programming 162 (2017): 83-112.

**Questions:**

To reiterate, my questions are:

- Why doesn't SVRG with AdaGrad step-sizes converge exactly and under what
    conditions is $\delta_{\hat g} = 0$ satisfied?

- Why are both Assumptions 6 and 9 required to show convergence of UniSVRG?

- Under what conditions are Assumptions 6/9 weaker than Assumption 3?

**Limitations:**

The contributions fit into a fairly small gap given previous work and experiments are not actually sufficient to justify the methods. See main review for details.

---

> ### Author Rebuttal · Authors · 2024-08-07
>
> Thanks for the positive evaluation!
>
> ## Major remarks
>
> 1. It seems there is a certain misunderstanding of Ass. 6, which we hope to clarify.
>
>    Basically, Ass. 6 is satisfied for finite-sum problems with (Hölder) smooth components (see also lines 186-194 in the paper for a more general example).
>
>     The motivation is as follows. Ass. 6 is the generalization of $\mathbb{E}\_{\xi}[\\| g(x, \xi) - g(y, \xi) \\|^2] \leq 2 L \beta(x, y)$, where $\beta(x, y) = f(y) - f(x) - \langle \nabla f(x), y - x \rangle$, for which the main example is the finite-sum objective $f(x) = \frac{1}{n} \sum_{i = 1}^n f_i(x)$ with the usual stochastic oracle $g(x, \xi) = \nabla f_{\xi}(x)$ (for a uniformly chosen index $\xi$) and $L$-smooth components $f_i$. To generalize it to the case when $f_i$ are $(\nu, H)$-Hölder smooth, we need to add $\delta$: $\mathbb{E}\_{\xi}[\\| g(x, \xi) - g(y, \xi) \\|^2] \leq 2 L [\beta(x, y) + \delta]$; then, we can take an arbitrary $\delta > 0$ and let $L = [\frac{1 - \nu}{2 (1 + \nu) \delta}]^{\frac{1 - \nu}{1 + \nu}} H^{\frac{2}{1 + \nu}}$. This is almost our Ass. 6 except that we have the second moment instead of the variance $\mathbb{E}\_{\xi}[\\| [g(x, \xi) - g(y, \xi)] - [\nabla f(x) - \nabla f(y)] \\|^2]$ in the left-hand side (for simplicity, we ignore the possibility of using inexact gradients $\bar{g}$ and function values $\bar{f}$). Using the variance is better because it is always smaller than the second moment and can easily be reduced with mini-batching.
>
> 1. There is no problem with $\delta_g$ in Thm. 10. First, let us stress that Thm. 10 covers the standard setting of finite-sum problems with $L$-smooth components. In this case, $L_f = L_g = L$, $\delta_f = \delta_g = 0$, and we recover the classical result from (Allen-Zhu and Yuan, [1]), without any extra assumptions.
>
>     However, it is important that our method is actually more powerful and converges in other regimes as well, e.g., for finite-sum problems with Hölder-smooth components. Indeed, as discussed before, for such problems, **$\delta_g$ can be an arbitrary (potentially very small) positive number** (and so can be $\delta_f$, see lines 116-117 in the paper). Choosing $\delta_f$ and $\delta_g$ carefully (by optimizing the rate from Thm. 10 while taking into account the fact that $L_f$ and $L_g$ also depend on these constants), we get the following convergence rate: $F_t \lesssim \frac{H D^{1 + \nu}}{(2^t)^{\frac{1 + \nu}{2}}}$ which goes to zero as $t \to \infty$. For more details, see Cor. 41.
>
> 1. The previous example shows that $\delta_f$ does not only measure the error
> between $\bar{g}$ and $\nabla f$ but is also related to the smoothness
> properties of the objective: even when $\bar{g} = \nabla f$, it is still
> meaningful to allow for $\delta_f > 0$ (e.g., to handle Hölder-smooth problems).
>
> 1. [Ass. 3 vs 6] Indeed, Ass. 6 is weaker than 3 because we can choose any
> $\delta_g > 0$ and let $L_g \sim \frac{\sigma^2}{\delta_g}$. Then, Thms. 4 and 5
> become simple corollaries of Thms. 7 and 8. On the other hand, as already
> discussed above, Asm. 6 is satisfied for finite-sum problems with smooth
> components and the standard oracle, while Asm. 3 may be violated for this case
> (consider the quadratic function on the entire space); even if one defines
> $\sigma$ in Asm. 3 by looking only at $x$ from the feasible set, $\sigma$
> will depend on the diameter of this set and may be potentially very big.
> We will add the corresponding comments.
>
> 1. [Asm. 6 vs 9] When we use exact gradients, $\bar{g} = \nabla f$, both are
> equivalent; otherwise, one does not seem to imply the other. For some reason,
> the current proof of Thm. 10 needs Asm. 9 at one place, namely, in lines
> 693-694, to get the Bregman distance term in Lem. 28, which can then be bounded
> via the function residual using Lem. 29. To remove Asm. 9, we would need to
> obtain a counterpart of Lem. 29 in terms of the approximate Bregman distance
> instead of the exact one, which we do not know how to do at the moment.
> We will clarify this.
>
> 1. [Experiments] We will take your point into account and add more experiments.
>
>    Regarding the mini-batch size, we are not sure if you are correct. Consider, e.g., SGD with mini-batch size $b$ as applied to minimizing an $L$-smooth convex function. To reach accuracy $\epsilon$, it needs $N_b = \frac{L D^2}{\epsilon} + \frac{\sigma^2 D^2}{b \epsilon^2}$ oracle calls, each such a call requires $b$ stochastic gradients. If the computation is sequential, the total SFO complexity is then $b N_b = \frac{b L D^2}{\epsilon} + \frac{\sigma^2 D^2}{\epsilon^2}$, which is minimized for $b = 1$. To our knowledge, mini-batching is provably efficient only when we allow for parallel computations of stochastic gradients; then, the complexity becomes $N_b$ which indeed decreases with $b$ (however, $b$ cannot exceed the limits of our parallelism, e.g., the number of computing nodes). See also Sec. 6 in [54] for a similar point of view.
>
> 1. [W1] Please note that [2] considers another variance assumption, namely,
> $\\| g(x, \xi) - \nabla f(x) \\|^2 \leq \sigma_0^2 + \sigma_1^2 \\| \nabla f(x) \\|^2$,
> which is completely different from ours and is not guaranteed to be satisfied
> (with good constants) even for finite-sum problems with smooth components.
> Our assumption is actually much weaker and have never been explored in the
> literature on AdaGrad. Furthermore, for the accelerated method with AdaGrad
> step sizes, nothing has been known except for the classical case when the
> variance is uniformly bounded. We therefore kindly disagree that our
> contributions are "fairly niche" / straightforward, including also the uniform
> extension of everything to Hölder-smooth problems.
>
> ## Minor remarks
>
> Unfortunately, we do not have enough space to answer every remark but we will
> consider all of them and make the appropriate modifications.
>
> ## Conclusion
>
> We hope we could answer your questions, and kindly ask you to consider
> increasing your score to support our work.

---

> > ### Comment · Reviewer_aWSb · 2024-08-08
> >
> > Thanks for your response.
> >
> > > 1. It seems there is a certain misunderstanding of Ass. 6, which we hope to clarify.
> >
> > What misunderstanding? Did I say something in my review which indicated I didn't understand Assumption 6? I just want you to give a concrete example in the paper.
> >
> > > 2.  First, let us stress that Thm. 10 covers the standard setting of finite-sum problems with $L$-smooth components.
> >
> > Does it say this somewhere in the manuscript? I appreciate your general presentation, but it's quite typical to include corollaries showcasing how the theorems behave in simplified settings. I think adding this type of statement would greatly improve your paper.
> >
> > > 4. Asm. 3 may be violated for this case (consider the quadratic function on the entire space)
> >
> > Don't you assume a finite diameter $D$ in your analysis? In this case, I think finite-sum quadratics should be fine for Assumption 3, right? I would like to you provide an example of a function *in your setting* which satisfies Assumption 6, but doesn't satisfy Assumption 3. Note that I'm not particularly concerned about assuming $D$ is finite (unlike some of the other reviewers), but you shouldn't assume finite $D$ and then make comparisons on the full space.
> >
> > Moreover, since you assert $\sigma^2$ may be very big for finite $D$, can you provide a comparison between $\sigma^2$ and $\delta_g, L_g$?
> >
> > > 5. [Asm. 6 vs 9] ... which we do not know how to do at the moment
> >
> > That's unfortunate. I don't have a lot of insight into this problem, but it does seem strange to need both assumptions.
> >
> > > 6. [Experiments] Regarding the mini-batch size, we are not sure if you are correct.
> >
> > There are quite a few papers on this subject which support my comments. Please see Gower et al. (1) and Kento et al. (2). I don't particularly like the second paper, but it get's the point across.
> >
> > > 7. Please note that [2] considers another variance assumption...
> >
> > Sorry, it looks like I missed including the additional references for my review. I've edited the review to add them now. By [2], I was referring to the work by Nguyen et al., which is [2] in my list. Their variance assumption is what I would consider to be "variance at the minimum". Of course you will not agree that your contributions are fairly niche, but that is my opinion.
> >
> > [1] Gower, Robert Mansel, et al. "SGD: General analysis and improved rates." International conference on machine learning. PMLR, 2019.
> >
> > [2] Imaizumi, Kento, and Hideaki Iiduka. "Iteration and stochastic first-order oracle complexities of stochastic gradient descent using constant and decaying learning rates." Optimization (2024): 1-24.

---

> > > ### Author Response · Authors · 2024-08-09
> > >
> > > Dear Reviewer aWSb,
> > >
> > > Thanks for the discussion.
> > >
> > > > adding this type of statement would greatly improve your paper.
> > >
> > > We agree and will add it.
> > >
> > > > I would like to you provide an example ...
> > >
> > > This is exactly the example we mentioned the last time, namely, the quadratic
> > > function with the variance growing with $\\| x \\|$. In this case, $\sigma$
> > > grows with the diameter $D$ and therefore may be arbitrarily large,
> > > in contrast to $\sigma_*$ which may not grow with $D$ at all.
> > >
> > > To be more precise, **here is one simple specific example**:
> > > $f(x) = \mathbb{E}\_L[F(x, L)]$, where $L$ is the random variable uniformly
> > > distributed on $[0, L_{\max}]$ and $F(x, L) = \frac{L}{2} \\| x \\|^2$.
> > > We want to minimize this function over the ball of radius $R$ centered at the origin,
> > > and use the standard oracle $g(x, L) = \nabla_x F(x, L) = L x$.
> > > Denoting $\bar{L} = \mathbb{E}[L] = \frac{1}{2} L_{\max}$ and
> > > $L_V^2 = \mathbb{E}[(L - \bar{L})^2] = \frac{1}{12} L_{\max}^2$,
> > > we get $f(x) = \frac{\bar{L}}{2} \\| x \\|^2$, and the variance is
> > > $\sigma^2(x) = \mathbb{E}\_L[\\| g(x, L) - \nabla f(x) \\|^2] = L_V^2 \\| x \\|^2$.
> > > Its maximal value over the feasible set is then
> > > $\sigma = \max \\{ \sigma(x) : \\| x \\| \leq R \\} = L_V R$, while
> > > $\sigma_* = \sigma(x^*) = 0$. For this example, our Asm. 1 is satisfied
> > > with $L_f = \bar{L}$, $\delta_f = 0$, while Asm. 6 is satisfied with
> > > $L_g = L_{\max}$, $\delta_g = 0$. The "classical" convergence rate (Th. 4) is thus
> > > $
> > >   O(\frac{L_f R^2}{k} + \frac{\sigma R}{\sqrt{k}})
> > >   = O(\frac{\bar{L} R^2}{k} + \frac{L_V R^2}{\sqrt{k}})
> > >   = O(\frac{L_{\max} R^2}{\sqrt{k}})
> > > $.
> > > In contrast, the $\sigma_*$-rate (Th. 7) is
> > > $
> > >   O(\frac{(L_f + L_g) R^2}{k} + \frac{\sigma_* R}{\sqrt{k}})
> > >   = O(\frac{L_{\max} R^2}{k})
> > > $,
> > > which is much faster.
> > >
> > > Please note that we are actually discussing now the general question why
> > > $\sigma_*$-bounds are better than the corresponding $\sigma$-bounds for
> > > smooth stochastic optimization problems. This question has already been
> > > addressed in many previous works and is not directly related to the specific
> > > adaptive methods we propose. The only important detail is that our "new"
> > > Asm. 6 indeed covers smooth stochastic optimization problems.
> > >
> > > > Please see Gower et al. (1) ...
> > >
> > > Thanks for the references. We do not mind adding extra experiments with another
> > > way of counting mini-batch computations (as you suggested in the first post) and
> > > will do that in the revised version.
> > >
> > > Note, however, that, from the theoretical point of view, the "optimal"
> > > mini-batch size suggested in [Gower et al., 1] does not give any nontrivial
> > > results: it is always comparable to either $b = 1$ or $b = n$. Indeed,
> > > consider, the finite-sum minimization of $f(x) = \frac{1}{n} \sum_{i = 1}^n f_i(x)$
> > > with $L$-smooth components $f_i$ and assume that
> > > $\frac{1}{n} \sum_{i = 1}^n \\| \nabla f_i(x) - \nabla f(x) \\|^2 \leq \sigma^2$
> > > for any $x$. One of the standard examples considered in [Gower et al., 1]
> > > is the so-called $b$-nice sampling meaning that the oracle uses mini-batching
> > > $g_b(x, \xi) = \frac{1}{b} \sum_{j = 1}^b \nabla f_{\xi_j}(x)$ with indices $\xi_j$
> > > chosen uniformly at random from $\\{ 1, \ldots, n \\}$ without replacement.
> > > The variance of such an oracle is $\sigma_b^2 = \frac{n - b}{n - 1} \frac{\sigma^2}{b}$,
> > > so SGD needs
> > > $
> > >   N_b
> > >   = O(\frac{L D^2}{\epsilon} + \frac{\sigma_b^2 D^2}{\epsilon^2})
> > >   = O(\frac{L D^2}{\epsilon} + \frac{n - b}{n} \frac{\sigma^2 D^2}{b \epsilon^2})
> > > $
> > > oracle calls to reach accuracy $\epsilon$. The total SFO complexity is then
> > > $b N_b = O(\frac{b L D^2}{\epsilon} + \frac{n - b}{n} \frac{\sigma^2 D^2}{\epsilon^2})$,
> > > which is exactly the same expression as we wrote in our previous reply, up to
> > > the factor $\frac{n - b}{n}$ which comes because we now use sampling
> > > *without replacement*. As we can see, the resulting expression is a linear
> > > function of $b$, so its minimum value is attained at one of the boundary points:
> > > either $b = 1$ (if $n \geq \frac{\sigma^2}{L \epsilon}$) or $b = n$ (otherwise).
> > > The "optimal" $b$ in [Gower et al., 1] appears to be more complicated simply because
> > > they first rewrite
> > > $b N_b = O(\max\\{\frac{b L D^2}{\epsilon}, \frac{n - b}{n} \frac{\sigma^2 D^2}{\epsilon^2}\\})$
> > > and then minimize the maximum by solving
> > > $\frac{b L D^2}{\epsilon} = \frac{n - b}{n} \frac{\sigma^2 D^2}{\epsilon^2}$.
> > > Although the resulting solution could potentially improve some *absolute* constants,
> > > it will still result in the same SFO complexity of
> > > $O(\min\\{ \frac{L D^2}{\epsilon} + \frac{\sigma^2 D^2}{\epsilon^2}, \frac{n L D^2}{\epsilon} \\})$
> > > as the naive approach selecting either $b = 1$ or $b = n$.
> > >
> > > > By [2], I was referring to the work by Nguyen et al. ... Their variance assumption ...
> > >
> > > Indeed, they work with the variance at the minimizer. However, our objection
> > > was that there are no other results in the literature showing that AdaGrad does
> > > adapt to the variance at the minimizer for convex finite-sum problems. Note that
> > > the work of Nguyen et al. studies only **non-adaptive** methods.

---

> > > > ### Comment · Reviewer_aWSb · 2024-08-09
> > > >
> > > > > Please note that we are actually discussing now the general question why $\sigma_*$-bounds are better than the corresponding $\sigma$-bounds for smooth stochastic optimization problems.
> > > >
> > > > Yes, but that's only because you have changed the direction of this conversation. The paper claims that "the assumption of uniformly bounded variance may not hold for some problems", but we seem to be in agreement that this claim is not really accurate for finite-sum functions under the bounded diameter assumption.
> > > >
> > > > > Although the resulting solution could potentially improve some absolute constants...
> > > >
> > > > Yes, and the improved constants usually make quite a difference in the experiment results since they are large.
> > > >
> > > > ---
> > > >
> > > > To be frank, I don't appreciate that the authors are changing the goal posts so that valid criticism of their paper appears invalid. Moreover, I don't think that carefully litigating each of my comments/criticisms is a good use for the discussion period. I suggest the authors focus on improving their paper and addressing constructive feedback from the reviewers.

---

> > > > > ### Author Response · Authors · 2024-08-09
> > > > >
> > > > > Dear Reviewer aWSb,
> > > > >
> > > > > Please kindly note that we have already agreed to clarify several things and add
> > > > > extra experiments as you suggested. It would therefore be unfair to say that we
> > > > > are not "addressing constructive feedback from the reviewers".
> > > > >
> > > > > Our discussion about the mini-batch size was a constructive reply to your
> > > > > original criticism that our experiments "do not show the actual relationship
> > > > > between oracle calls and optimization error" and that you "strongly dislike the
> > > > > choice to treat one mini-batch gradient computation as a single oracle call
> > > > > instead of $b$ calls". We have explained in detail that there is actually
> > > > > nothing wrong with it and our viewpoint corresponds to parallel mini-batch
> > > > > computations. This is a realistic computational model for which the optimal
> > > > > mini-batch size is the number of available computing nodes.
> > > > >
> > > > > We would also like to return to the following two major points of criticism from
> > > > > your original review:
> > > > >
> > > > > 1. "In comparison to this work, Nguyen et al. [2] give results for
> > > > >    (strongly-convex) SGD that depend only on the stochastic gradient variance at
> > > > >    the minimizer and without any variance assumption."
> > > > >
> > > > > 2. "I'm somewhat disturbed that SVRG with Adagrad step-sizes doesn't converge
> > > > >    exactly to a minimizer $x^*$ under your assumptions...  In comparison,
> > > > >    standard SVRG, which does not require any variance bound
> > > > >    (Allen-Zhu and Yuan, [1]), converges exactly."
> > > > >
> > > > > As we have already clarified:
> > > > >
> > > > > - We do **not** need any extra assumptions compared to (Nguyen et al., [2])
> > > > >   and (Allen-Zhu and Yuan, [1]). Their smoothness assumption implies our
> > > > >   Assumptions 1 and 6, and our results are **not** weaker.
> > > > > - Under the same setting as in (Allen-Zhu and Yuan, [1]), our SVRG method
> > > > >   **does** converge to a minimizer. There are no problems with $\delta_g$.
> > > > >
> > > > > Could you please confirm if you agree with these statements and your concerns
> > > > > are now resolved?

---

> > > > > > ### Comment · Reviewer_aWSb · 2024-08-12
> > > > > >
> > > > > > > It would therefore be unfair to say that we not "addressing constructive feedback from the reviewers"
> > > > > >
> > > > > > I did not say that. To be clear, I think that highly technical arguments are quite difficult to hold over OpenReview, place significant burdens on both reviewers and authors, and are often better replaced by high-level clarifications.
> > > > > >
> > > > > > >  We have explained in detail that there is actually nothing wrong with it and our viewpoint corresponds to parallel mini-batch computations.
> > > > > >
> > > > > > I still don't agree with this. I think that our discussion so-far has shown that (i) there is an optimal mini-batch size, which in the setting of Gower et al. [1] improves by constants factors and in the setting of Imaizumi [2] improves the convergence order (note this requires a general stochastic setting). These facts are hidden by choice to treat a single mini batch as one oracle call. If you wish to consider highly parallel stochastic oracles which permit multiple, simultaneous oracle calls, then you should state that in the paper. Note that this literature admits different algorithms and different lower-bounds from standard sequential oracles [B2019].
> > > > > >
> > > > > > > Could you please confirm if you agree with these statements and your concerns are now resolved?
> > > > > >
> > > > > > Yes, I agree with these two statements. I will update my review accordingly.
> > > > > >
> > > > > > ----
> > > > > >
> > > > > > Overall, I agree with the remarks by Reviewer PtE8 that the extension to Holder-smooth functions is the main contribution of this paper. As I said in my original review, the contributions of this paper fit into small gap between existing works and, as such, I don't think it will have a large impact on either theory or practice. Since the paper is sound and the contributions are likely interesting to some of the NeurIPS community, I think 6 is an accurate score.
> > > > > >
> > > > > > [B2019] Bubeck, Sébastien, et al. "Complexity of highly parallel non-smooth convex optimization." Advances in neural information processing systems 32 (2019).

---

> > > > > > > ### Author Response · Authors · 2024-08-14
> > > > > > >
> > > > > > > Dear Reviewer aWSb,
> > > > > > >
> > > > > > > Regarding the mini-batch size, we have already explained to you (with two
> > > > > > > examples) why the "optimal" mini-batch size you propose will not improve
> > > > > > > anything besides maybe (already quite reasonable) absolute constants in the
> > > > > > > theoretical worst-case complexity estimates. Such an improvement is rather
> > > > > > > insignificant, especially considering the fact that the expression for the
> > > > > > > "optimal" mini-batch size involves a number of unknown constants and is
> > > > > > > therefore completely impractical; tuning the mini-batch size in the experiments
> > > > > > > is possible but will incur an additional overhead defeating the potential
> > > > > > > gain in improving the absolute constants.
> > > > > > >
> > > > > > > Note also that Imaizumi [2] does not improve any order of convergence. They
> > > > > > > simply say that, for any stepsize $\alpha \leq \frac{1}{L}$, it is possible to
> > > > > > > choose a sufficiently large mini-batch size $b \sim \frac{L \alpha \sigma^2}{\epsilon^2}$
> > > > > > > such that, after $N \sim \frac{F_0}{\alpha \epsilon^2}$ iterations, the method
> > > > > > > will find a point with $\epsilon$-suboptimality in terms of the gradient norm.
> > > > > > > However, the total SFO complexity is
> > > > > > > $b N \sim \frac{L F_0}{\epsilon^2} + \frac{L F_0 \sigma^2}{\epsilon^4}$,
> > > > > > > which is exactly the same result as one gets by using mini-batch size $b = 1$
> > > > > > > and choosing the step size
> > > > > > > $\alpha \sim \min\\{ \frac{1}{L}, \sqrt{\frac{F_0}{k L \sigma^2}} \\}$
> > > > > > > (see Corollary 2.2 in [GL13]).
> > > > > > >
> > > > > > > Therefore, we still disagree with your comment that treating the mini-batch
> > > > > > > computation as a single oracle call (assuming the corresponding computation can
> > > > > > > be efficiently parallelized) is wrong. In contrast to what you are proposing, in
> > > > > > > such a framework, increasing the mini-batch size does provably lead to the
> > > > > > > improvement in the total running time, and such an improvement is not just a
> > > > > > > (small) absolute constant. Furthermore, the corresponding computational model is
> > > > > > > completely realistic (distributed optimization) and does not require tuning the
> > > > > > > mini-batch size which is instead simply set to the number of available computing
> > > > > > > nodes. Again, see, e.g., [WS21] from our main reply for more details on the
> > > > > > > "minibatch parallelization speedup". Note that our viewpoint is still reasonable
> > > > > > > even if our methods might not attain the lower complexity bounds from [B2019]
> > > > > > > (which are nevertheless for a completely different setting of computing in
> > > > > > > parallel full gradients and when the number of such computations is growing
> > > > > > > sufficiently fast with the dimension of the space).
> > > > > > >
> > > > > > > **References:**
> > > > > > >
> > > > > > > - [GL13] S. Ghadimi, G. Lan. Stochastic First- and Zeroth-Order Methods for
> > > > > > >   Nonconvex Stochastic Programming. SIOPT, 2013.
> > > > > > >
> > > > > > > ---
> > > > > > >
> > > > > > > As for the contributions, we kindly disagree that extension to Hölder-smooth
> > > > > > > functions is the only main contribution of our work.
> > > > > > >
> > > > > > > As we have already mentioned in our general reply and also during our discussion
> > > > > > > with Reviewer PtE8, **another important contribution of our work is establishing
> > > > > > > $\sigma_*$-bounds** for SGD methods with AdaGrad step sizes (Theorems 7 and 8),
> > > > > > > **including the nontrivial accelerated method**. **These bounds are novel and
> > > > > > > have not been proved in any of the previous works.** Neither of the works [AK23]
> > > > > > > or (Nguyen et al., [2]) which you mentioned before contains any comparable
> > > > > > > results.
> > > > > > >
> > > > > > > **Another significant contribution is a novel version of the adaptive
> > > > > > > accelerated SVRG method (Algorithm 4).** This method and its analysis are both
> > > > > > > much simpler compared to the only other existing option which is the AdaVRAE
> > > > > > > method from [LNEN22], while our complexity results are stronger. See our
> > > > > > > discussion with Reviewers PtE8 and soXZ for more details.
> > > > > > >
> > > > > > > Compared to the variety of other papers, we also **allow the stochastic oracle
> > > > > > > to be biased** and **present concise proofs which cover, in a unified way, many
> > > > > > > different methods (basic and accelerated, with and without variance
> > > > > > > reduction)**. For example, the whole proof for the accelerated SVRG takes only
> > > > > > > 3.5 pages (see Section A.4.2) once we reuse the general result on one step of
> > > > > > > the accelerated SGD (which does not use any variance reduction at all). This is
> > > > > > > also an important aspect of our work which should not be neglected.

---

### Official Review · Reviewer_soXZ · 2024-07-11

**Soundness:** 3
**Presentation:** 3
**Contribution:** 3
**Rating:** 6
**Confidence:** 4

**Summary:**

This paper demonstrates the universality of AdaGrad in stochastic optimization, presenting adaptive algorithms that converge efficiently without prior knowledge of problem-specific constants.  The research contributes novel variance reduction techniques, theoretical proofs, and empirical evidence, showcasing robust performance across optimization scenarios.  It advances the field by offering a versatile approach to stochastic optimization, hinting at potential extensions to more complex problem sets.

**Strengths:**

The paper's strength lies in its theoretical depth, the algorithms are designed to be universally applicable to a wide range of optimization problems, including those with Hölder smooth components, showcasing a high level of adaptability. They also illustrate the impact of the mini-batch size on the convergence of proposed methods, which further helps to verify the theorem.

**Weaknesses:**

Rather than pointing out shortcomings, I would like to discuss some issues with the authors.

**Questions:**

The prior knowledge is also not required in UniXGrad[28] and AcceleGrad[32]. What is the difference between them and the proposed algorithm?

**Limitations:**

It is mentioned on line 246 of page 7 that "Alternative accelerated SVRG schemes with AdaGrad stepsizes (3) were recently proposed in [34]; however, they seem to be much more complicated. " I think it would be beneficial if the author discussed the algorithm in [34] in detail to help readers understand the advantages of the algorithm in this paper more clearly.

---

> ### Author Rebuttal · Authors · 2024-08-07
>
> Thank you for the positive evaluation of our work. Below you can find the
> answers to your questions / comments.
>
> > The prior knowledge is also not required in UniXGrad[28] and AcceleGrad[32].
> > What is the difference between them and the proposed algorithm?
>
> From the algorithmic perspective, all the three methods (UniXGrad, AcceleGrad
> and our UniFastSgd) are different versions of Nesterov's accelerated gradient
> method. Our UniFastSgd is one of the standard versions known as the
> Method of Similar Triangles (see Section 6.1.3 in [44] and [31]). The name stems
> from the fact that the next point $x_{k + 1}$ is defined in such a way that the
> triangles $(x_k, v_k, v_{k + 1})$ and $(x_k, y_k, x_{k + 1})$ are similar (see
> the picture on page 1 in the PDF attached to the main rebuttal). In contrast,
> UniXGrad and AcceleGrad choose $x_{k + 1}$ as the result of the (projected)
> gradient step from $y_k$, which results in a slightly more complicated iteration
> with no guarantee of similar triangles.
>
> From the theoretical perspective, the convergence rate guarantees for
> UniXGrad and AcceleGrad were proved only under the uniformly bounded variance
> and for functions with either bounded gradients or Lipschitz continuous
> gradients. In other words, it is not known whether UniXGrad and AcceleGrad can
> provably adapt to the more general assumptions from our paper (such as Hölder
> smoothness, variance at the minimizer, etc.). It might be possible to extend our
> techniques to those methods as well, but we have not investigated this direction
> as we personally find the Method of Similar Triangles more elegant and simpler
> to work with.
>
> > It is mentioned on line 246 of page 7 that "Alternative accelerated SVRG
> > schemes with AdaGrad stepsizes (3) were recently proposed in [34]; however,
> > they seem to be much more complicated. " I think it would be beneficial if the
> > author discussed the algorithm in [34] in detail to help readers understand
> > the advantages of the algorithm in this paper more clearly.
>
> Thank you for the suggestion. We agree and will elaborate on this in the revised
> version of the manuscript. Here are some of the reasons why our method is
> simpler and more elegant than the AdaVRAE algorithm from [34]:
>
> 1. Each epoch of our algorithm is essentially the standard Method of Similar
>    Triangles with the only difference that one of the vertices ($\tilde{x}$)
>    is fixed during the epoch (we mention this in lines 240-244; see also the
>    picture on page 2 in the PDF attached to the main rebuttal). As a result, its
>    geometry is much easier to understand than that of AdaVRAE, and we can
>    readily apply the classical results for each Triangle Step (such as Lemma 24)
>    without proving everything "from scratch".
> 2. Our algorithm uses a simple recurrent formula for the sequence $A_t$,
>    namely, $A_{t + 1} = A_t + \sqrt{A_t}$, which can be easily analyzed using
>    standard techniques (Lemma 32). In contrast, AdaVRAE uses quite complicated
>    ad-hoc formulas which are impossible to even write on a single line.
>
> We hope we could answer your questions, and kindly ask you to consider
> increasing your score to support our work.

---

> > ### Author Response · Authors · 2024-08-13
> >
> > Dear Reviewer soXZ,
> >
> > Thanks again for your feedback. Please let us know if you are satisfied with our
> > reply and if your concerns are now resolved. We would be happy to provide more
> > explanations if needed.

---

### Official Review · Reviewer_PtE8 · 2024-07-13

**Soundness:** 2
**Presentation:** 2
**Contribution:** 2
**Rating:** 5
**Confidence:** 4

**Summary:**

This paper studied how to apply AdaGrad type stepsize for stochastic convex optimization in a unified way. It proposed different algorithms and a general rule for the stepsize. Later, the authors presented different convergence rates under different settings, all of which match the existing best results. Finally, numerical experiments are given to show the better performance of these new algorithms.

**Strengths:**

The paper also extends the result to the Hölder smooth case.

**Weaknesses:**

1. The major weakness is still Assumption 2. Though I understand this condition is widely used in the prior works with AdaGrad stepsize, this doesn't mean we should always accept it.

    **i.** From the practical perspective, a bounded domain means one either has some prior information on the solution or puts some artificial constraints on the problem. However, both of them may not be realistic. A simple example is logistic regression with separable data (but this is unknown in advance). Then clearly, putting a bounded domain can guarantee the existence of $x^*$. However, one can notice that a better solution always exists but not in the domain.

    **ii.** From the analysis perspective, with a bounded domain, one can immediately bound $\frac{M_k\\|x_k-x^*\\|^2-M_k\\|x_{k+1}-x^\*\\|^2}{2}\leq\frac{M_k\\|x_k-x^*\\|^2-M_{k+1}\\|x_{k+1}-x^\*\\|^2}{2}+O((M_{k+1}-M_{k})D^2)$ to get a telescoping sum, which significantly simplify the analysis. In my experience, this is the major hardness. Without a bounded domain, one has to pay more effort to control this term or may meet other difficulties if one chooses to divide $M_k$ in the analysis.

    **iii.** More importantly, I think different works with AdaGrad-like stepsize are trying to relax it recently as mentioned by the authors.

    **iv**. To summarize, under Assumption 2, the paper is more like a combination of different existing results. Nothing particularly new I can find in both algorithms and proofs. Even under Assumption 1 (or any inexact oracle), the proofs still follow the classical steps and one only needs to put $\delta$ in the R.H.S. of every inequality.

2. If I am not wrong, all results (except Theorems 7 and 8) in the paper hold for any $F(u)$ but not only $F^*$. If so, the authors can add a remark to state it to improve the paper. But feel free to correct me if I missed something.

3. For some plots, I cannot find the confidence interval. Please add it according to the checklist.

**Questions:**

See **Weaknesses**.

**Limitations:**

Not applicable.

---

> ### Author Rebuttal · Authors · 2024-08-07
>
> Thanks for your time and efforts spent on reviewing this manuscript. We
> appreciate the feedback and would like to make some comments on the points you
> raised in your review.
>
> ## Major remarks
>
> 1. Our assumption on the bounded feasible set is satisfied whenever one knows
>    some upper bound $R$ on the distance $\\| x_0 - x^* \\|$ from the initial point
>    to the solution. In this case, we can easily transform our initial problem
>    into the one with the bounded feasible set of diameter $D = 2 R$ by adding
>    the additional simple ball constraint $\\| x - x_0 \\| \leq R$. This is
>    essentially the setting we consider.
>
>    We agree that it would be highly desirable to develop a completely
>    parameter-free method for stochastic optimization, which would not require
>    the knowledge of any problem-dependent parameters and would automatically and
>    efficiently adapt to all of them. However, according to the several recent
>    works (see [CH24,AK24,KJ24]), this seems too much to ask for. In
>    contrast to the standard deterministic optimization, where one can compute
>    exact function values and use line-search techniques by paying only an
>    additive logarithmic term for not knowing the parameters of the problem, for
>    stochastic optimization, one really needs some nontrivial knowledge of either
>    the distance to the solution $R$, or the smoothness constant / oracle's
>    variance, etc. Without this, the method would not be able to achieve a nearly
>    optimal complexity.
>
>    In other words, without assuming the knowledge of $R$, we would need to
>    impose other restrictive assumptions such as the knowledge of the Lipschitz
>    constant and/or oracle's variance. But then the algorithm will be tailored a
>    particular function class and will not be as universal as our methods
>    (working **simultaneously** for each Hölder class and different variance
>    assumptions). Therefore, it is quite debatable which assumption is actually
>    better.
>
> 2. Note that problems with bounded domains arise quite commonly in machine
>    learning problems. In fact, any $\ell_2$-regularized convex problem $\min_x
>    [f(x) + \frac{\lambda}{2} \\| x \\|^2]$ ($P_{\lambda}$) (e.g., logistic
>    regression) is equivalent to $\min_x \\{ f(x) : \\| x \\| \leq D \\}$ ($P'\_D$)
>    for a certain $D$, so that there is a one-to-one correspondence between them.
>    In practice, one usually selects the right regularization coefficient
>    $\lambda$ by using the grid search, i.e., solving $(P\_{\lambda})$ for
>    multiple values of $\lambda$ and checking the quality of the resulting
>    solution $x_{\lambda}$. But this is exactly the same as doing the grid search
>    over $D$, solving $(P'\_D)$ and checking the quality of the corresponding
>    solution $x'\_D$. For an additional discussion of bounded domains, see also
>    Ch. 5 in [SNW11] and p. 125 in particular.
>
> 3. We kindly disagree with your assessment that, under Assumption 2, our paper
>    is just a trivial combination of different existing results. Consider, for
>    example, the following two particular cases of (only a part of) our results:
>    - Adaptive accelerated SGD for smooth problems with the bound via $\sigma_*$
>      (Theorem 8 for $\delta_f = \delta_{\hat{g}} = 0$).
>    - Adaptive accelerated SVRG method for functions with Hölder-smooth
>      components (last row in Table 2 or Corollary 42).
>
>    In our opinion, both results are quite nontrivial:
>
>    1) The first (nonadaptive)
>    accelerated method with the bound via $\sigma_*$ was suggested only recently,
>    in [WS21]; the adaptive method was mentioned there as an open question. The
>    algorithm was later revisited and improved in [IJLL23] (by properly
>    separating the smoothness constants), however, the method was still
>    non-adaptive.
>
>    2) We are aware of only one adaptive accelerated SVRG method, namely, AdaVRAE
>    from [LNEN22], which was proven to work only for functions with Lipschitz
>    gradient. However, as we explained in our reply to Reviewer soXZ, that
>    algorithm and the corresponding proofs are rather complicated and it is
>    extremely difficult to check if the method provably works for the more
>    general class of Hölder-smooth problems.
>
>    Could you please be more specific and explain in detail why you believe that
>    the above two results are not particularly new and easily follow from the
>    existing results in the literature (assuming our Assumption 2 holds)?
>    Please provide some references.
>
> References:
> - [CH24] Y. Carmon, O. Hinder. The Price of Adaptivity in Stochastic Convex
>   Optimization. arXiv:2402.10898, 2024.
> - [AK24] A. Attia, T. Koren. How Free is Parameter-Free Stochastic Optimization?
>   ICML, 2024.
> - [KJ24] A. Khaled, C. Jin. Tuning-Free Stochastic Optimization. ICML, 2024.
> - [WS21] B. Woodworth, N. Srebro. An Even More Optimal Stochastic Optimization
>   Algorithm: Minibatching and Interpolation Learning. NeurIPS, 2021.
> - [LNEN22] Z. Liu, T. Nguyen, A. Ene, H. Nguyen. Adaptive Accelerated
>   (Extra-)Gradient Methods with Variance Reduction. ICML, 2022.
> - [IJLL23] S. Ilandarideva, A. Juditsky, G. Lan, T. Li. Accelerated stochastic
>   approximation with state-dependent noise. arXiv:2307.01497, 2023.
> - [SNW11] S. Sra, S. Nowozin, S. Wright. Optimization for Machine Learning. MIT
>   Press, 2011.
>
> ## Minor remarks
>
> - [W2] ($F(u)$ instead of $F^*$): Yes, this is indeed true, but we do not see
>   how this could be useful for improving our results. We would appreciate it if
>   you could elaborate.
> - [W3] (confidence interval): Thank you, we will add it.
>
> ## Conclusion
>
> We hope we were able to address your concerns and kindly ask you to reconsider
> your score.

---

> > ### Comment · Reviewer_PtE8 · 2024-08-10
> >
> > Thanks for your reply. My comments are as follows:
> >
> > **To Major remarks 1.** This response is not convincing.
> >
> > 1. I understand the setting you are considering. But as I stated and commented by the authors, it is undesirable since it may not be practical in many cases.
> >
> > 2.  The description of the lower bound is highly inaccurate. Note the recent lower bounds developed for parameter-free algorithms only apply to classical stochastic optimization. Instead, the current paper considers the finite-sum case. I remark that a key assumption in these lower bounds for general stochastic optimization is the finite variance condition. However, this condition is not usually assumed in the finite-sum case. Hence, the results mentioned by the authors don't make too much sense in the current setting.
> >
> >     As a simple example, one cannot apply the famous $O(1/\epsilon^2)$ lower bound in stochastic convex optimization to the finite sum case since the latter admits a well-known faster upper bound $O(n+\sqrt{n/\epsilon})$.
> >
> >
> > **To Major remarks 2.** Though theoretically speaking, the authors' discussion holds, I am confused about what the authors want to convey. The problem in the paper is to solve $P'_D$ in your notation. However, it seems like the authors want to tell me that grid search on $\lambda$ for $P\_\lambda$ is reasonable, which I think is unrelated to the paper. Could you elaborate more on this point?
> >
> > **To Major remarks 3.**
> >
> > 1. As I stated earlier, the telescoping sum appears easily under a bounded domain (see the inequality mentioned in the review) and one only needs to keep every $\delta$ in your notation simply on the R.H.S. of every inequality. If the authors think my statement is not accurate, could you please let me know at which step in the proof requires additional analysis on $\delta$?
> >
> > 2. Moreover, I would like to clarify that I only said the algorithm and the analysis are not new. I kindly remind the authors that I also pointed out that extending Hölder smooth case is a strength. In other words, the key issue in my opinion is that the analysis doesn't bring any new insights and hence is a trivial combination.
> >
> > 3. The description is also inaccurate for [LNEN22] as it contains two algorithms: AdaVRAE and AdaVRAG. Moreover, whether the proof is difficult to check or not is a subjective view. I think for a general reader who is not familiar with the optimization literature, your paper and [LNEN22] are in the same difficulty when going through the proof. Acutally, as far as I can see, once one puts $\delta$ in every R.H.S. inequality in the proof of [LNEN22], I cannot see any obvious obstacles to prevent their proof work. If the authors think there are some barriers in their proofs, please explicitly write it down.
> >
> > **To Minor remark [W2].** Note that one reason that people study optimization errors for the finite-sum case is to understand the excess risk of ML algorithms, which can always be decomposed as follows: $\mathbb{E}\_{S,A}[F(x(A))-F(x^*)]=\mathbb{E}\_{S,A}[F(x(A))-F\_S(x(A))]+\mathbb{E}_{S,A}[F\_S(x(A))-F\_S(x^*)]$ where $F(x)=\mathbb{E}\_{z\sim P}[f(x,z)]$, $P$ is an unknown distribution, $x^*\in\mathrm{argmin}F(x)$, $S$ is a set of independent samples of $z$, $F\_S(x)=\frac{1}{\|S\|}\sum\_{z\in S}f(x,z)$, and $x(A)$ is the output by an algorithm $A$. Note that $x^*$ may not be the optimal solution of $F_S(x)$. As such, a bound on the optimization error for any reference point $u$ is more useful than $F_S(x(A))-F\_S^*$ where $F\_S^*=\inf F\_S(x)$.

---

> > > ### Author Response · Authors · 2024-08-10
> > >
> > > **Remark 1:**
> > >
> > > 1. Please note that **our paper does not consider only the finite-sum case**.
> > >    Instead, we consider a general stochastic optimization problem in which the
> > >    objective function can be of any form, as long as we are able to compute its
> > >    stochastic gradients. In particular, it could be the "classical stochastic
> > >    optimization" problem $\min_x \\{ f(x) = \mathbb{E}_{\xi}[F(x, \xi)] \\}$ for
> > >    which the recent lower bounds developed for parameter-free algorithms do
> > >    apply. We mentioned the finite-sum case simply because it is an important
> > >    example (but not the only one).
> > >
> > > 1. The **lower bounds we were discussing do make sense even for finite-sum problems**
> > >    in the important situation when $n$ is very large ($n \to \infty$). In this
> > >    case, the $O(n + \sqrt{n / \epsilon})$ bound you mentioned is of no use.
> > >
> > > **Remark 2:**
> > >
> > > We were simply providing an example of an important family of applied problems
> > > which have bounded domain with known diameter. Essentially, the point was that,
> > > instead of the commonly used additive regularization for model selection, one
> > > may use the equivalent "ball regularization".
> > >
> > > **Remark 3:**
> > >
> > > 1. Even when $\delta_f = \delta_g = 0$, some of our results are still new, e.g.,
> > >    Th. 7 and 8. As we indicated in our previous reply, the adaptive
> > >    accelerated SGD method with the $\sigma_*$-bound was considered an open
> > >    question. **If you believe that Th. 8 and its proof containing the
> > >    analysis of Alg. 2 are not new, could you please indicate the
> > >    corresponding works with the same result?**
> > >
> > > 2. Regarding the addition of $\delta$ in the right-hand side of most
> > >    inequalities, this is largely true. However, **this is not a drawback of our
> > >    approach but instead a confirmation of its elegance**: we offer a
> > >    reasonably simple analysis leading, in particular, to the state-of-the-art
> > >    convergence rates for the Hölder-smooth problems. In contrast, the only other
> > >    existing convergence analysis of AdaGrad methods for Hölder-smooth stochastic
> > >    optimization problems from [RKWAC24] is completely different and more
> > >    complicated: it does not use any $\delta$ and requires several rather
> > >    technical lemmas to handle recurrent sequences involving the combination of
> > >    several terms in different powers depending on $\nu$ (see Lemmas E.6-E.9 in
> > >    their paper).
> > >
> > >    Nevertheless, some care should be taken with "simply adding $\delta$ everywhere".
> > >    One interesting example is Th. 8 which establishes the convergence rate of
> > >    $O(\frac{L_f D^2}{k^2} + \frac{L_g D^2}{k} + k \delta_f + \delta_g + \frac{\sigma_* D}{\sqrt{k}})$.
> > >    To get, from this result, the correct
> > >    $
> > >      O(\frac{H_f(\nu) D^{1 + \nu}}{k^{\frac{1 + 3 \nu}{2}}} +
> > >      \frac{H_{\max}(\nu) D^{1 + \nu}}{(b k)^{\frac{1 + \nu}{2}}} +
> > >      \frac{\sigma_* D}{\sqrt{k}})
> > >    $
> > >    convergence rate (Cor. 40), it is very important to allow $\delta_f$
> > >    and $\delta_g$ (defined by our Asms. 1 and 6) be different. If we were
> > >    not careful and treated them as the same $\delta$ everywhere, we would end up
> > >    with the slower rate of
> > >    $
> > >      O(\frac{H_f(\nu) D^{1 + \nu}}{k^{\frac{1 + 3 \nu}{2}}} +
> > >      \frac{H_{\max}(\nu) D^{1 + \nu}}{b^{\frac{1 + \nu}{2}} k^{\nu}} +
> > >      \frac{\sigma_* D}{\sqrt{k}})
> > >    $
> > >    (where the second term does not even go to zero when $\nu = 0$).
> > >    **One of the "nontrivial" insights of our work is the realization that such a
> > >    separation is important.**
> > >
> > > 3. We did not consider AdaVRAG from [LNEN22] because its complexity is worse by
> > >    an extra logarithmic factor.
> > >
> > >    As we have already explained in our reply to Reviewer soXZ, **our UniFastSvrg
> > >    algorithm is considerably simpler than AdaVRAE** and is based on the Method
> > >    of Similar Triangles which is well-known in the optimization community.
> > >    Consequently, its convergence analysis is much easier to follow: after
> > >    applying the standard results on one triangle step (which is unrelated to
> > >    SVRG methods at all), it only requires 3.5 pages to finish the proof for the
> > >    accelerated SVRG (see Sec. A.4.2). Of course, we are not speaking here
> > >    about "a general reader who is not familiar with the optimization literature"
> > >    for whom any optimization paper would be very difficult.
> > >
> > >    > Acutally, as far as I can see, once one puts $\delta$ in every R.H.S.
> > >    > inequality in the proof of [LNEN22], I cannot see any obvious obstacles to
> > >    > prevent their proof work
> > >
> > >    With all due respect, it is impossible to verify such a claim without
> > >    carefully checking every line of their 15+ page long proof.
> > >
> > > **Remark [W2]:**
> > >
> > > Thanks for the clarification. Note, however, that, for any point $u$, we have
> > > $F(x_k) - F(u) \leq F(x_k) - F^*$. Thus, proving $F(x_k) - F(u) \leq \epsilon$
> > > for any $u$ is exactly the same as proving $F(x_k) - F^* \leq \epsilon$.
> > >
> > > **References:**
> > >
> > > - [RKWAC24] A. Rodomanov, A. Kavis, Y. Wu, K. Antonakopoulos, V. Cevher.
> > >   Universal Gradient Methods for Stochastic Convex Optimization. ICML, 2024.

---

> > > > ### Comment · Reviewer_PtE8 · 2024-08-12
> > > >
> > > > I thank the author's detailed reply.
> > > >
> > > > **To Remark 1.** I first apologize for my previous wrong claim of your paper only considering the finite-sum case. However, I should mention that the general lower bound doesn't apply to the finite-sum case as it was proved under the finite variance condition as clearly stated in the previous discussion. $n\to \infty$ is another case that is not included in our previous discussion.
> > > >
> > > > **To Remark 2.** After the clarification, the authors seem to want to say that, instead of solving $P_\lambda$, solving $P'_D$ is acceptable. Again, theoretically speaking, this holds. But no one did this in practice as far as I know. Moreover, what $D$ people should use is still unknown. I remind the author that your setting is a problem with a **known** diameter. Say I want to take $\lambda = 1$ (or $\lambda$ in a set of values), then what $D$ should I use for $P'_D$? More importantly, with $\lambda >0$, $P\_\lambda$ is a strongly convex problem, which can be solved faster than $P'\_D$ if $f$ is not strongly convex, which is the case studied in the paper.
> > > >
> > > > **To Remark 3.** Thanks for the clarification. I understand that the separation between $\delta_f$ and $\delta_g$ is more critical, which I suggest to point out in the paper. Based on this insight, I would like to increase my score to 5.

---

> > > > > ### Author Response · Authors · 2024-08-12
> > > > >
> > > > > Dear Reviewer PtE8,
> > > > >
> > > > > Thanks for the interesting discussion.
> > > > >
> > > > > **Remark 2:**
> > > > >
> > > > > As we have indicated in our initial reply, for model selection, one usually
> > > > > needs to solve $(P_{\lambda})$ not for one specific value of $\lambda$ but for
> > > > > all of them, and then choose the best one (in terms of accuracy on the
> > > > > validation set). This is typically done by using grid search over multiple
> > > > > values of $\lambda$. What we were saying before is that this entire procedure is
> > > > > equivalent to doing grid search over $D$, solving $(P'_D)$ and choosing the best
> > > > > solution. Thus, we always know $D$ in the latter procedure.
> > > > >
> > > > > Regarding strong convexity, it is not entirely true that a strongly convex
> > > > > problem is always faster to solve than a (non-strongly) convex one. Everything
> > > > > depends on the specific constants. Indeed, consider the convex problem
> > > > > $\min_x f(x)$ with $L$-smooth objective and assume that the size of its solution
> > > > > is $D = \\| x^* \\|$. Solving this problem to $\epsilon$-accuracy by the
> > > > > Gradient Method (GM) started from the origin takes $O(\frac{L D^2}{\epsilon})$
> > > > > oracle calls. Now consider the strongly convex $\epsilon$-approximation of the
> > > > > original objective: $f\_{\lambda}(x) = f(x) + \frac{\lambda}{2} \\| x \\|^2$ for
> > > > > $\lambda = \frac{\epsilon}{R^2}$. To get an $\epsilon$-solution to the
> > > > > original problem, we need to minimize the regularized objective up to accuracy
> > > > > $\frac{\epsilon}{2}$. The function $f_{\lambda}$ is $\lambda$-strongly convex,
> > > > > so GM needs $\tilde{\Theta}(\frac{L}{\lambda}) = \tilde{\Theta}(\frac{L R^2}{\epsilon})$
> > > > > oracle calls, which is the same complexity as solving the original (non-strongly)
> > > > > convex problem. (What we have described here is, in fact, a well-known general
> > > > > reduction of a convex problem to a strongly convex one, which does not help
> > > > > to reduce the complexity.) For our example with problems $(P_{\lambda})$ and
> > > > > $(P'_D)$, we have exactly the same effect: $(P\_{\lambda})$ is easy to solve
> > > > > only when $\lambda$ is sufficiently large, but then the equivalent problem
> > > > > $(P'_D)$ has small $D$ and is therefore easy to solve as well. Again, please
> > > > > see Ch. 5 in [SNW11] (from our initial reply) and p. 125 in particular for an
> > > > > independent argument why solving $(P'_D)$ instead of $(P\_{\lambda})$ is indeed
> > > > > a reasonable idea.
> > > > >
> > > > > **$\sigma_*$-bounds:**
> > > > >
> > > > > We would also like to repeat our previous comment that establishing complexities
> > > > > for Hölder-smooth problems is only one of the main contributions of our work.
> > > > > Another significant one is proving $\sigma_*$-bounds for the basic and
> > > > > accelerated adaptive methods. To our knowledge, such bounds are novel (even in
> > > > > the case when $\delta = 0$) and the corresponding proofs are nontrivial
> > > > > (especially for the accelerated method, see our previous comments on this
> > > > > matter). Would you perhaps be willing to agree with this and raise your score
> > > > > further?

---

> > > > > > ### Comment · Reviewer_PtE8 · 2024-08-14
> > > > > >
> > > > > > I thank the authors' discussion. I finally decided to maintain my current score after careful consideration.

---

### Author Rebuttal · Authors · 2024-08-07

We thank all the reviewers for their valuable comments. We did our best to
answer all the questions and will be happy to continue the discussion if
needed.

After reading the reviews, we had the impression that several aspects of our
work were probably unnoticed or underappreciated (perhaps, due to the high level
of generality of our presentation). Therefore, we would like to draw the
reviewers' attention to the following few of our contributions which are
particular cases of our general results but are nonetheless quite important in
themselves:

1. Adaptive methods for minimizing finite-sum objectives
$f(x) = \frac{1}{n} \sum_{i = 1}^n f_i(x)$ with $L$-smooth components $f_i$,
whose convergence rate is expressed in terms of the variance $\sigma_*$ at the
minimizer (Theorems 7 and 8 for $\delta_f = \delta_g = 0$).

1. Extensions of these algorithms to Hölder-smooth problems (first two rows in
Table 2 or Corollaries 39 and 40).

1. Variance reduction methods for finite-sum problems with Hölder-smooth
components.

We are not aware of any other works that provably solve any of the
aforementioned problems, even under the assumption that the distance to the
solution is known, or the feasible set is bounded.

Regrading the first contribution or, more generally, relaxing the assumption of
uniformly bounded variance, we know only the works [FTCMSW22,AK23]. However,
they use a different variance assumption, which is weaker than ours and is not
guaranteed to hold for finite-sum problems with smooth components. Furthermore,
the corresponding methods are not accelerated. The first non-adaptive
accelerated method capable of solving finite-sum problems while enjoying the
variance bound only at the minimizer was developed only recently in [WS21];
making this method adaptive was left as an open question.

As for the Hölder smoothness, it is important to mention that the very fact that
stochastic AdaGrad methods, including the accelerated algorithm, provably work
for the entire Hölder class (and not just for its extreme subclasses&mdash;those
with bounded or Lipschitz continuous gradients&mdash;which was known before from
[LYC18,KLBC19]) is very recent and was proved only in [RKWAC24]. The extension
of the corresponding results to $\sigma_*$-bounds and explicit variance
reduction are both highly nontrivial tasks.

We would also like to point out that, for each of the above three contributions,
we do not only present the basic methods but also the accelerated ones, which is
always rather challenging.

With that being said, we hope the reviewers could take another look at our work
and increase their scores.

Finally, we attach the PDF containing the two pictures clarifying the geometry
of our accelerated methods, as discussed with Reviewer soXZ.

References:
- [RKWAC24] A. Rodomanov, A. Kavis, Y. Wu, K. Antonakopoulos, V. Cevher. ICML, 2024.
- [AK23] A. Attia and T. Koren. SGD with AdaGrad Stepsizes: Full Adaptivity with
  High Probability to Unknown Parameters, Unbounded Gradients and Affine
  Variance. ICML, 2023.
- [FTCMSW22] M. Faw, I. Tziotis, C. Caramanis, A. Mokhtari, S. Shakkottai, R.
  Ward. The Power of Adaptivity in SGD: Self-Tuning Step Sizes with Unbounded
  Gradients and Affine Variance, COLT, 2022.
- [WS21] B. Woodworth, N. Srebro. An Even More Optimal Stochastic Optimization
  Algorithm: Minibatching and Interpolation Learning. NeurIPS, 2021.
- [KLBC19] A. Kavis, K. Levy, F. Bach, V. Cevher. UniXGrad: A Universal,
  Adaptive Algorithm with Optimal Guarantees for Constrained Optimization.
  NeurIPS, 2019.
- [LYC18] K. Y. Levy, A. Yurtsever, V. Cevher. Online Adaptive Methods,
  Universality and Acceleration. NeurIPS, 2018.

---

> ### Author Response · Authors · 2024-08-14
>
> Dear All,
>
> We have noticed that the [RKWAC24] reference in our original reply was missing
> a title. The correct reference is the following:
>
> - [RKWAC24] A. Rodomanov, A. Kavis, Y. Wu, K. Antonakopoulos, V. Cevher.
>   Universal Gradient Methods for Stochastic Convex Optimization. ICML, 2024.
>
> We would also like to highlight a few other important contributions of our
> work, in addition to those mentioned previously.
>
> - We propose a novel version of the adaptive accelerated SVRG method (Algorithm 4).
>   This method and its analysis are both much simpler compared to the only other
>   existing option which is the AdaVRAE method from [LNEN22], while our
>   complexity results are stronger. See our discussion with Reviewers PtE8 and
>   soXZ for more details.
>
> - Even in the special case of Hölder-smooth problems with uniformly bounded
>   variance, for which the complexity results of our methods coincide with those
>   from [RKWAC24], our work still offers some novelty and improvements compared
>   to [RKWAC24]. First, our convergence analysis works not only for the modified
>   AdaGrad step size from [RKWAC24] but also for the more classical one
>   (based on the summation of squared gradient norms, see eq. (3) in our paper).
>   Further, our convergence analysis is considerably different and, in fact,
>   simpler. In particular, compared to [RKWAC24], we do not need to use several
>   rather technical lemmas to handle recurrent sequences involving the
>   combination of several terms in different powers depending on $\nu$
>   (see Lemmas E.6-E.9 in their paper). Such a simplification is quite
>   significant and was the first crucial step allowing us to concisely extend
>   the corresponding results to $\sigma_*$-bounds and variance-reduced methods.
>
> - Compared to the variety of other papers, we also allow the stochastic oracle
>   to be biased and present concise proofs which cover, in a unified way, many
>   different methods (basic and accelerated, with and without variance
>   reduction). For example, the whole proof for the accelerated SVRG takes only
>   3.5 pages (see Section A.4.2) once we reuse the general result on one step of
>   the accelerated SGD (which does not use any variance reduction at all). This
>   is also an important aspect of our work which should not be neglected.
>
> We hope the Reviewers and the Area Chair will keep in mind the contributions
> mentioned above and in the previous reply during the final evaluation of our
> work.

---

### Decision · Program_Chairs · 2024-09-25

**Decision:**

Accept (poster)

**Comment:**

After a length author response period, all reviewers ultimately recommended acceptance of the paper.